# Nearly Optimal VC-Dimension and Pseudo-Dimension Bounds for Deep Neural Network Derivatives

**Yahong YANG**
Department of Mathematics
The Pennsylvania State University, University Park,
State College, PA, USA
and
Department of Mathematics
Hong Kong University of Science and Technology
Clear Water Bay, Hong Kong SAR, China
yxy5498@psu.edu

**Haizhao YANG**[*]
Department of Mathematics and Department of Computer Science
University of Maryland College Park
College Park, MD, USA
hzyang@umd.edu

**Yang XIANG**
Department of Mathematics
Hong Kong University of Science and Technology
Clear Water Bay, Hong Kong SAR, China
and
Algorithms of Machine Learning and Autonomous Driving Research Lab
HKUST Shenzhen-Hong Kong Collaborative Innovation Research Institute
Futian, Shenzhen, China
maxiang@ust.hk

## Abstract

This paper addresses the problem of nearly optimal Vapnik–Chervonenkis dimension (VC-dimension) and pseudo-dimension estimations of the derivative functions of deep neural networks (DNNs). Two important applications of these estimations include: 1) Establishing a nearly tight approximation result of DNNs in the Sobolev space; 2) Characterizing the generalization error of machine learning methods with loss functions involving function derivatives. This theoretical investigation fills the gap of learning error estimations for a wide range of physics-informed machine learning models and applications including generative models, solving partial differential equations, operator learning, network compression, distillation, regularization, etc.

## 1   Introduction

The Sobolev training [8, 40, 46, 45, 22, 42] of deep neural networks (DNNs) has had a significant impact on scientific and engineering fields, including solving partial differential equations [25, 12,

---

[*]Corresponding author.

37th Conference on Neural Information Processing Systems (NeurIPS 2023).

33, 10], operator learning [29, 26], network compression [35], distillation [21, 34], regularization [8], and dynamic programming [15, 48], etc. For example, Sobolev (semi) norms have been applied to penalize function gradients in loss functions [2, 17, 15, 30] to control the Lipschitz constant of DNNs. Moreover, Sobolev norms and equivalent formulas are commonly used to define loss functions in various applications such as dynamic programming [15, 48], solving partial differential equations [25, 12, 33], and distillation [21, 34, 35]. These loss functions enable models to learn DNNs that can approximate the target function with small discrepancies in both magnitude and derivative. For example, when utilizing the Deep Ritz method [12] to solve PDEs such as the following one:

$$\begin{cases} -\Delta u = f & \text{in } \Omega, \\ \frac{\partial u}{\partial \nu} = 0 & \text{on } \partial\Omega, \end{cases} \tag{1}$$

the corresponding loss function can be expressed as:

$$\mathcal{E}_D(\boldsymbol{\theta}) := \frac{1}{2} \int_\Omega |\nabla\phi(\boldsymbol{x};\boldsymbol{\theta})|^2 \mathrm{d}\boldsymbol{x} + \frac{1}{2} \left( \int_\Omega \phi(\boldsymbol{x};\boldsymbol{\theta})\mathrm{d}\boldsymbol{x} \right)^2 - \int_\Omega f\phi(\boldsymbol{x};\boldsymbol{\theta})\mathrm{d}\boldsymbol{x},$$

where $\boldsymbol{\theta}$ represents all the parameters in the neural network. Here, $\Omega$ denotes the domain $(0,1)^d$. Proposition 1 in [27] establishes the equivalence between the loss function $\mathcal{E}_D(\boldsymbol{\theta})$ and $\|\phi(\boldsymbol{x};\boldsymbol{\theta}) - u^*(\boldsymbol{x})\|_{H^1((0,1)^d)}$, where $u^*(\boldsymbol{x})$ denotes the exact solution of the PDEs in equation (1), and $\|f\|_{H^1((0,1)^d)} := \left( \sum_{0 \le |\alpha| \le 1} \|D^{\boldsymbol{\alpha}}f\|_{L^2((0,1)^d)}^p \right)^{1/2}$. Thus, the Sobolev norm $H^1((0,1)^d)$ serves as a measure of the loss function, and Sobolev training is employed to solve PDEs within the Deep Ritz method.

Two natural questions that arise are: 1) What is the optimal approximation error of DNNs described by a Sobolev norm? 2) What is the generalization error of the loss function defined by a Sobolev norm? The key step to address these questions is to estimate the optimal Vapnik–Chervonenkis dimension (VC-dimension) and pseudo-dimension [3, 46, 1, 32] of DNNs and their derivatives. Intuitively, these concepts characterize the complexity or richness of a function set and, hence, they can be applied to establish the best possible approximation and generalization power of DNNs.

**Definition 1** (VC-dimension [1]). *Let $H$ denote a class of functions from $\mathcal{X}$ to $\{0, 1\}$. For any non-negative integer $m$, define the growth function of $H$ as*

$$\Pi_H(m) := \max_{x_1, x_2, \dots, x_m \in \mathcal{X}} |\{(h(x_1), h(x_2), \dots, h(x_m)) : h \in H\}| .$$

*The Vapnik–Chervonenkis dimension (VC-dimension) of $H$, denoted by VCdim($H$), is the largest $m$ such that $\Pi_H(m) = 2^m$. For a class $\mathcal{G}$ of real-valued functions, define VCdim($\mathcal{G}$) := VCdim(sgn($\mathcal{G}$)), where sgn($\mathcal{G}$) := $\{\text{sgn}(f) : f \in \mathcal{G}\}$ and sgn($x$) = $1[x > 0]$.*

**Definition 2** (pseudo-dimension [32]). *Let $\mathcal{F}$ be a class of functions from $\mathcal{X}$ to $\mathbb{R}$. The pseudo-dimension of $\mathcal{F}$, denoted by Pdim($\mathcal{F}$), is the largest integer $m$ for which there exists $(x_1, x_2, \dots, x_m, y_1, y_2, \dots, y_m) \in \mathcal{X}^m \times \mathbb{R}^m$ such that for any $(b_1, \dots, b_m) \in \{0, 1\}^m$ there is $f \in \mathcal{F}$ such that $\forall i : f(x_i) > y_i \iff b_i = 1$.*

The main contribution of this paper is to estimate nearly optimal bounds of the VC-dimension and pseudo-dimension of DNN derivatives. Based on these bounds, we can prove the optimality of our DNN approximation, as measured by Sobolev norms (Theorem 3), and obtain a tighter generalization error of loss functions defined by Sobolev norms. Our results facilitate the understanding of Sobolev training and the performance of DNNs in Sobolev spaces.

Bounds for the VC-dimension and pseudo-dimension of DNNs have been established in [16, 5, 3, 4, 6, 41, 20]. However, these approaches and findings cannot be applied to Sobolev training, as they do not account for the derivatives of DNNs, which represent a key difference between Sobolev training and classical methods. Obtaining such bounds for DNN derivatives is much more difficult due to their complex compositional structures. DNN derivatives consist of a series of interdependent parts that are multiplied together via the chain rule, rendering existing methods for estimating bounds inapplicable. Estimating the VC-dimension and pseudo-dimension of DNN derivatives is the most crucial and challenging problem addressed in this paper. In [11], the VC-dimension and pseudo-dimension of DNN derivatives were analyzed, but the results were suboptimal due to a lack of consideration for the relationships between the multiplied terms in a DNN derivative. As a result, their findings do not provide the optimal approximation of DNNs in Sobolev spaces and can only give a generalization

error that is much larger than the actual error that may arise from Sobolev training. In this paper, we introduce a novel method that investigates these relationships, resulting in a simplified complexity of DNN derivatives. This, in turn, allows us to obtain nearly optimal bounds on their VC-dimension and pseudo-dimension.

The paper is divided into two parts. In the first part, we establish a nearly optimal bound on the VC-dimension of DNN derivatives with the ReLU activation function $\sigma_1(x) := \max\{0, x\}$:

**Theorem 1.** *For any $N, L, d \in \mathbb{N}_+$, there exists a constant $\bar{C}$ independent with $N, L$ such that*

$$VCdim(D\Phi) \leq \bar{C} N^2 L^2 \log_2 L \log_2 N, \tag{2}$$

*for*

$$D\Phi := \{\psi = D_i\phi : \phi \in \Phi, \ i = 1, 2, \ldots, d\}, \tag{3}$$

*where $\Phi := \{\phi : \phi \text{ is a } \sigma_1\text{-NN in } \mathbb{R}^d \text{ with width} \leq N \text{ and depth} \leq L\}$, and $D_i$ is the weak derivative in the $i$-th variable.*

By utilizing Theorem 1, we prove that our DNN approximation rate for approximating functions in Sobolev spaces $W^{n,\infty}((0,1)^d)$ using Sobolev norms in $W^{1,\infty}((0,1)^d)$ is nearly optimal. We present our construction of DNNs for this approximation in Theorem 3, and we demonstrate the optimality of such approximation in Theorem 4. Furthermore, we generalize our method to approximate DNNs in Sobolev spaces measured by Sobolev norms $W^{m,\infty}((0,1)^d)$ for $m \geq 2$. The details of this generalization are presented in Corollaries 1 and 2. The Sobolev spaces, equipped with Sobolev (semi) norms, are defined as follows:

**Definition 3** (Sobolev Spaces [13]). *Denote $\Omega$ as $(0,1)^d$, $D$ as the weak derivative of a single variable function and $D^{\boldsymbol{\alpha}} = D_1^{\alpha_1} D_2^{\alpha_2} \ldots D_d^{\alpha_d}$ as the partial derivative where $\boldsymbol{\alpha} = [\alpha_1, \alpha_2, \ldots, \alpha_d]^T$ and $D_i$ is the derivative in the $i$-th variable. Let $n \in \mathbb{N}$ and $1 \leq p \leq \infty$. Then we define Sobolev spaces*

$$W^{n,p}(\Omega) := \{f \in L^p(\Omega) : D^{\boldsymbol{\alpha}} f \in L^p(\Omega) \text{ for all } \boldsymbol{\alpha} \in \mathbb{N}^d \text{ with } |\boldsymbol{\alpha}| \leq n\}$$

*with a norm $\|f\|_{W^{n,p}(\Omega)} := \left(\sum_{0 \leq |\boldsymbol{\alpha}| \leq n} \|D^{\boldsymbol{\alpha}} f\|_{L^p(\Omega)}^p\right)^{1/p}$, if $p < \infty$, and $\|f\|_{W^{n,\infty}(\Omega)} := \max_{0 \leq |\boldsymbol{\alpha}| \leq n} \|D^{\boldsymbol{\alpha}} f\|_{L^\infty(\Omega)}$. Furthermore, for $\boldsymbol{f} = (f_1, f_2, \ldots, f_d)$, $\boldsymbol{f} \in W^{1,\infty}(\Omega, \mathbb{R}^d)$ if and only if $f_i \in W^{1,\infty}(\Omega)$ for each $i = 1, 2, \ldots, d$ and $\|\boldsymbol{f}\|_{W^{1,\infty}(\Omega, \mathbb{R}^d)} := \max_{i=1,\ldots,d}\{\|f_i\|_{W^{1,\infty}(\Omega)}\}$.*

**Definition 4** (Sobolev semi-norm [13]). *Let $n \in \mathbb{N}_+$ and $1 \leq p \leq \infty$. Then we define Sobolev semi-norm $|f|_{W^{n,p}(\Omega)} := \left(\sum_{|\boldsymbol{\alpha}|=n} \|D^{\boldsymbol{\alpha}} f\|_{L^p(\Omega)}^p\right)^{1/p}$, if $p < \infty$, and $|f|_{W^{n,\infty}(\Omega)} := \max_{|\boldsymbol{\alpha}|=n} \|D^{\boldsymbol{\alpha}} f\|_{L^\infty(\Omega)}$. Furthermore, for $\boldsymbol{f} \in W^{1,\infty}(\Omega, \mathbb{R}^d)$, we define $|\boldsymbol{f}|_{W^{1,\infty}(\Omega, \mathbb{R}^d)} := \max_{i=1,\ldots,d}\{|f_i|_{W^{1,\infty}(\Omega)}\}$.*

In the second part of our paper, we utilize our previous work on estimating the VC-dimension of DNN derivatives to obtain an upper bound on the pseudo-dimension of DNN derivatives:

**Theorem 2.** *For any $N, L, d \in \mathbb{N}_+$, there exists a constant $\widehat{C}$ independent with $N, L$ such that*

$$Pdim(D\Phi) \leq \widehat{C} N^2 L^2 \log_2 L \log_2 N, \tag{4}$$

*where $D\Phi$ is defined in Theorem 1.*

Based on Theorem 2, we can estimate the generalization error of loss functions defined by Sobolev norms, as demonstrated in Theorem 5. Specifically, the error is bounded by $O(NL(\log_2 N \log_2 L)^{1/2})$ with respect to the width $N$ and depth $L$ of DNNs. This bound is significantly smaller than the previously reported bound of $O(NL^{5/2}(\log_2 N \log_2 L)^{1/2})$ in [11]. We attribute this improvement to our more accurate estimation of the pseudo-dimension of DNN derivatives. Our findings indicate that learning target functions with loss functions defined by Sobolev norms does not require substantially more sample points than those defined by $L^2$-norms [14], as their generalization error orders are equivalent with respect to the width $N$ and depth $L$ of DNNs.

Our main contributions are:

• We propose a method to achieve nearly optimal estimations of the VC-dimension and pseudo-dimension of DNN derivatives.

● By utilizing our estimation of the VC-dimension of DNN derivatives, we demonstrate the optimality of our DNN approximation, as measured by Sobolev norms.

● By applying our estimation of the pseudo-dimension of DNN derivatives, we obtain a bound for the generalization error measured by the Sobolev norm. Importantly, our results demonstrate that the degree of generalization error defined by Sobolev norms is equivalent to that defined by $L^2$-norms, corresponding to the width $N$ and depth $L$ of DNNs.

## 2 Preliminaries

Let us summarize all basic notations used in the DNNs as follows:

**1**. Matrices are denoted by bold uppercase letters. For example, $\boldsymbol{A} \in \mathbb{R}^{m \times n}$ is a real matrix of size $m \times n$ and $\boldsymbol{A}^{\intercal}$ denotes the transpose of $\boldsymbol{A}$.

**2**. Vectors are denoted by bold lowercase letters. For example, $\boldsymbol{v} \in \mathbb{R}^n$ is a column vector of size $n$. Furthermore, denote $\boldsymbol{v}(i)$ as the $i$-th elements of $\boldsymbol{v}$.

**3**. For a $d$-dimensional multi-index $\boldsymbol{\alpha} = [\alpha_1, \alpha_2, \cdots \alpha_d] \in \mathbb{N}^d$, we denote several related notations as follows: $(a)$ $|\boldsymbol{\alpha}| = |\alpha_1| + |\alpha_2| + \cdots + |\alpha_d|$; $(b)$ $\boldsymbol{x}^{\boldsymbol{\alpha}} = x_1^{\alpha_1} x_2^{\alpha_2} \cdots x_d^{\alpha_d}$, $\boldsymbol{x} = [x_1, x_2, \cdots, x_d]^{\intercal}$; $(c)$ $\boldsymbol{\alpha}! = \alpha_1! \alpha_2! \cdots \alpha_d!$.

**4**. Let $B_{r,|\cdot|}(\boldsymbol{x}) \subset \mathbb{R}^d$ be the closed ball with a center $\boldsymbol{x} \in \mathbb{R}^d$ and a radius $r$ measured by the Euclidean distance. Similarly, $B_{r,\|\cdot\|_{\ell_\infty}}(\boldsymbol{x}) \subset \mathbb{R}^d$ be the closed ball with a center $\boldsymbol{x} \in \mathbb{R}^d$ and a radius $r$ measured by the $\ell_\infty$-norm.

**5**. Assume $\boldsymbol{n} \in \mathbb{N}_+^n$, then $f(\boldsymbol{n}) = \boldsymbol{O}(g(\boldsymbol{n}))$ means that there exists positive $C$ independent of $\boldsymbol{n}, f, g$ such that $f(\boldsymbol{n}) \leq Cg(\boldsymbol{n})$ when all entries of $\boldsymbol{n}$ go to $+\infty$.

**6**. Define $\sigma_1(x) := \sigma(x) = \max\{0, x\}$ and $\sigma_2 := \sigma^2(x)$. We call the neural networks with activation function $\sigma_t$ with $t \leq i$ as $\sigma_i$ neural networks ($\sigma_i$-NNs). With the abuse of notations, we

define $\sigma_i : \mathbb{R}^d \to \mathbb{R}^d$ as $\sigma_i(\boldsymbol{x}) = \begin{bmatrix} \sigma_i(x_1) \\ \vdots \\ \sigma_i(x_d) \end{bmatrix}$ for any $\boldsymbol{x} = [x_1, \cdots, x_d]^T \in \mathbb{R}^d$.

**7**. Define $L, N \in \mathbb{N}_+$, $N_0 = d$ and $N_{L+1} = 1$, $N_i \in \mathbb{N}_+$ for $i = 1, 2, \ldots, L$, then a $\sigma_i$-NN $\phi$ with the width $N$ and depth $L$ can be described as follows:

$$\boldsymbol{x} = \tilde{\boldsymbol{h}}_0 \overset{W_1, b_1}{\longrightarrow} \boldsymbol{h}_1 \overset{\sigma_i}{\longrightarrow} \tilde{\boldsymbol{h}}_1 \ldots \overset{W_L, b_L}{\longrightarrow} \boldsymbol{h}_L \overset{\sigma_i}{\longrightarrow} \tilde{\boldsymbol{h}}_L \overset{W_{L+1}, b_{L+1}}{\longrightarrow} \phi(\boldsymbol{x}) = \boldsymbol{h}_{L+1},$$

where $\boldsymbol{W}_i \in \mathbb{R}^{N_i \times N_{i-1}}$ and $\boldsymbol{b}_i \in \mathbb{R}^{N_i}$ are the weight matrix and the bias vector in the $i$-th linear transform in $\phi$, respectively, i.e., $\boldsymbol{h}_i := \boldsymbol{W}_i \tilde{\boldsymbol{h}}_{i-1} + \boldsymbol{b}_i$, for $i = 1, \ldots, L+1$ and $\tilde{\boldsymbol{h}}_i = \sigma_i(\boldsymbol{h}_i)$, for $i = 1, \ldots, L$. In this paper, an DNN with the width $N$ and depth $L$, means (a) The maximum width of this DNN for all hidden layers less than or equal to $N$. (b) The number of hidden layers of this DNN less than or equal to $L$.

## 3 Nearly Optimal Approximation Results of DNNs in Sobolev Spaces Measured by Sobolev Norms

### 3.1 Approximation of functions in $W^{n,\infty}$ with $W^{1,\infty}$ norm by ReLU neural networks

In this subsection, we construct deep neural networks (DNNs) with a width of $\boldsymbol{O}(N \log N)$ and a depth of $\boldsymbol{O}(L \log L)$ to approximate functions in the Sobolev space $W^{n,\infty}$, as measured by Sobolev norms in $W^{1,\infty}$. The approximation rate achieved by these networks is $\boldsymbol{O}(N^{-2(n-1)/d} L^{-2(n-1)/d})$.

**Theorem 3.** *For any $f \in W^{n,\infty}((0,1)^d)$ with $n \geq 2$ and $\|f\|_{W^{n,\infty}((0,1)^d)} \leq 1$, any $N, L \in \mathbb{N}_+$, there is a $\sigma_1$-NN $\phi$ with the width $(34 + d)2^d n^{d+1}(N + 1) \log_2(8N)$ and depth $56d^2 n^2(L + 1) \log_2(4L)$ such that*

$$\|f(\boldsymbol{x}) - \phi(\boldsymbol{x})\|_{W^{1,\infty}((0,1)^d)} \leq C_9(n,d) N^{-2(n-1)/d} L^{-2(n-1)/d},$$

*where $C_9$ is the constant independent with $N, L$.*

The proof of Theorem 3 can be outlined in five parts, and the complete proof is provided in Appendix 7.2:

**(i)**: First of all, define a sequence of subsets of $\Omega$:

**Definition 5.** *Given $K, d \in \mathbb{N}^+$, and for any $\boldsymbol{m} = (m_1, m_2, \ldots, m_d) \in \{1, 2\}^d$, we define $\Omega_{\boldsymbol{m}} := \prod_{j=1}^d \Omega_{m_j}$, where $\Omega_1 := \bigcup_{i=0}^{K-1} \left[\frac{i}{K}, \frac{i}{K} + \frac{3}{4K}\right]$, $\Omega_2 := \bigcup_{i=0}^{K} \left[\frac{i}{K} - \frac{1}{2K}, \frac{i}{K} + \frac{1}{4K}\right] \cap [0, 1]$.*

Then we define a partition of unity $\{g_{\boldsymbol{m}}\}_{\boldsymbol{m} \in \{1,2\}^d}$ on $(0, 1)^d$ with supp $g_{\boldsymbol{m}} \cap (0, 1)^d \subset \Omega_{\boldsymbol{m}}$ for each $\boldsymbol{m} \in \{1, 2\}^d$:

**Definition 6.** *Given $K, d \in \mathbb{N}_+$, we define*

$$
g_1(x) := \begin{cases} 1, & x \in \left[\frac{i}{K} + \frac{1}{4K}, \frac{i}{K} + \frac{1}{2K}\right] \\ 0, & x \in \left[\frac{i}{K} + \frac{3}{4K}, \frac{i+1}{K}\right] \\ 4K\left(x - \frac{i}{K}\right), & x \in \left[\frac{i}{K}, \frac{i}{K} + \frac{1}{4K}\right] \\ -4K\left(x - \frac{i}{K} - \frac{3}{4K}\right), & x \in \left[\frac{i}{K} + \frac{1}{2K}, \frac{i}{K} + \frac{3}{4K}\right] \end{cases}, \quad g_2(x) := g_1\left(x + \frac{1}{2K}\right), \quad (5)
$$

*for $i \in \mathbb{Z}$. For any $\boldsymbol{m} = (m_1, m_2, \ldots, m_d) \in \{1, 2\}^d$, define $g_{\boldsymbol{m}}(\boldsymbol{x}) = \prod_{j=1}^d g_{m_j}(x_j)$, $\boldsymbol{x} = (x_1, x_2, \ldots, x_d)$.*

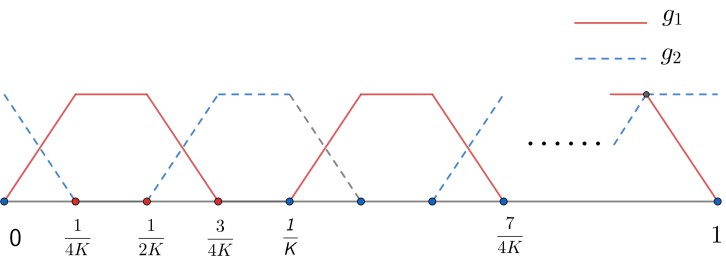

Figure 1: The schematic diagram of $g_i$ for $i = 1, 2$.

**(ii)**: Then we use the following proposition to approximate $\{g_{\boldsymbol{m}}\}_{\boldsymbol{m} \in \{1,2\}^d}$ by $\sigma_1$-NNs and construct a sequence of $\sigma_1$-NNs $\{\phi_{\boldsymbol{m}}\}_{\boldsymbol{m} \in \{1,2\}^d}$:

**Proposition 1.** *Given any $N, L, n \in \mathbb{N}_+$ for $K = \lfloor N^{1/d} \rfloor^2 \lfloor L^{2/d} \rfloor$, then for any $\boldsymbol{m} = (m_1, m_2, \ldots, m_d) \in \{1, 2\}^d$, there is a $\sigma_1$-NN with the width smaller than $(9+d)(N+1)+d-1$ and depth smaller than $15d(d-1)nL$ such as $\|\phi_{\boldsymbol{m}}(\boldsymbol{x}) - g_{\boldsymbol{m}}(\boldsymbol{x})\|_{W^{1,\infty}((0,1)^d)} \leq 50d^{\frac{5}{2}}(N+1)^{-4dnL}$.*

The proof of Proposition 1 is presented in Appendix 7.2.1.

**(iii)**: For each $\Omega_{\boldsymbol{m}} \subset [0, 1]^d$, where $\boldsymbol{m} \in \{1, 2\}^d$, we find a function $f_{K, \boldsymbol{m}}$ satisfying

$$
\begin{aligned}
\|f - f_{K, \boldsymbol{m}}\|_{W^{1,\infty}(\Omega_{\boldsymbol{m}})} &\leq C_1(n, d) K^{-(n-1)}, \\
\|f - f_{K, \boldsymbol{m}}\|_{L^\infty(\Omega_{\boldsymbol{m}})} &\leq C_1(n, d) K^{-n},
\end{aligned} \quad (6)
$$

where $C_1$ is a constant independent of $K$. Moreover, each $f_{K, \boldsymbol{m}}$ can be expressed as $f_{K, \boldsymbol{m}} = \sum_{|\boldsymbol{\alpha}| \leq n-1} g_{f, \boldsymbol{\alpha}, \boldsymbol{m}}(\boldsymbol{x}) \boldsymbol{x}^{\boldsymbol{\alpha}}$, where $g_{f, \boldsymbol{\alpha}, \boldsymbol{m}}(\boldsymbol{x})$ is a piecewise constant function on $\Omega_{\boldsymbol{m}}$. The proof of this result is based on the Bramble-Hilbert Lemma [7, Lemma 4.3.8], and the details are provided in Appendix 7.2.2.

**(iv)**: The fourth step involves approximating $f_{K, \boldsymbol{m}}$ using neural networks $\psi_{\boldsymbol{m}}$, following the approach outlined in [28]. This method is suitable for our work because $g_{f, \boldsymbol{\alpha}, \boldsymbol{m}}(\boldsymbol{x})$ is a piecewise constant function on $\Omega_{\boldsymbol{m}}$, and the weak derivative of $g_{f, \boldsymbol{\alpha}, \boldsymbol{m}}(\boldsymbol{x})$ on $\Omega_{\boldsymbol{m}}$ is zero. This property allows for the use of the $L^\infty$ norm approximation method presented in [28]. Thus, we obtain a neural network $\psi_{\boldsymbol{m}}$ with width $\boldsymbol{O}(N \log N)$ and depth $\boldsymbol{O}(L \log L)$ such that

$$
\begin{aligned}
\|f_{K, \boldsymbol{m}} - \psi_{\boldsymbol{m}}(\boldsymbol{x})\|_{W^{1,\infty}(\Omega_{\boldsymbol{m}})} &\leq C_5(n, d) N^{-2(n-1)/d} L^{-2(n-1)/d} \\
\|f_{K, \boldsymbol{m}} - \psi_{\boldsymbol{m}}(\boldsymbol{x})\|_{L^\infty(\Omega_{\boldsymbol{m}})} &\leq C_5(n, d) N^{-2n/d} L^{-2n/d},
\end{aligned} \quad (7)
$$

where $C_5$ is a constant independent of $N$ and $L$.

By combining (iii) and (iv) and setting $K = \lfloor N^{1/d} \rfloor^2 \lfloor L^{2/d} \rfloor$, we obtain that for each $\boldsymbol{m} \in \{1,2\}^d$, there exists a neural network $\psi_{\boldsymbol{m}}$ with width $\boldsymbol{O}(N \log N)$ and depth $\boldsymbol{O}(L \log L)$ such that

$$\|f(\boldsymbol{x}) - \psi_{\boldsymbol{m}}(\boldsymbol{x})\|_{W^{1,\infty}(\Omega_{\boldsymbol{m}})} \leq C_6(n,d) N^{-2(n-1)/d} L^{-2(n-1)/d}$$

$$\|f(\boldsymbol{x}) - \psi_{\boldsymbol{m}}(\boldsymbol{x})\|_{L^\infty(\Omega_{\boldsymbol{m}})} \leq C_6(n,d) N^{-2n/d} L^{-2n/d}, \tag{8}$$

where $C_6$ is a constant independent of $N$ and $L$. Further details are provided in Appendix 7.2.3.

**(v)**: The final step is to combine the sequences $\{\phi_{\boldsymbol{m}}\}_{\boldsymbol{m} \in \{1,2\}^d}$ and $\{\psi_{\boldsymbol{m}}\}_{\boldsymbol{m} \in \{1,2\}^d}$ to construct a network that can approximate $f$ over the entire space $[0,1]^d$. We define the sequence $\{\phi_{\boldsymbol{m}}\}_{\boldsymbol{m} \in \{1,2\}^d}$ because $\psi_{\boldsymbol{m}}$ may not accurately approximate $f$ on $[0,1]^d \backslash \Omega_{\boldsymbol{m}}$. The purpose of $\phi_{\boldsymbol{m}}$ is to remove this portion of the domain and allow other networks to approximate $f$ on $[0,1]^d \backslash \Omega_{\boldsymbol{m}}$. Further details on this step are provided in Appendix 7.2.4.

While recent works [28, 23, 39, 18, 31, 9, 19] have studied the approximation of smooth functions or functions in Sobolev spaces by DNNs measured in the norm of $L^p(\Omega)$ or $W^{s,p}(\Omega)$, they typically present results that are not optimal or are measured in $L^p$-norms. For example, in [28], they applies Taylor's expansion to approximate smooth functions but cannot be applied directly in Sobolev spaces. In [39], they improve on this by using the Bramble–Hilbert Lemma to approximate functions in Sobolev spaces, but their error is still measured in $L^p$-norms. In [18], the authors show that there exists a ReLU neural network that can approximate $f \in W^{1,p}(\Omega)$, but their approximation rate is not optimal and is the same as that in traditional methods such as the finite element theory. Our work provides a superior approximation rate. Later, a rigorous proof of optimality of Theorem 3 is discussed in Appendix 7.2.4 and Subsection 3.3.

## 3.2 Approximation of functions in $W^{n,\infty}$ measured by $W^{m,\infty}$ norm with $m > 1$ by neural networks (sketches of the proofs of the Corollaries 1 and 2)

In this subsection, we utilize neural networks to approximate functions in $W^{n,\infty}$ measured by $W^{m,\infty}$, where $m > 1$. The proof strategy is similar to the approximation measured in the norm of $W^{1,\infty}$. However, we cannot rely on ReLU neural networks alone to achieve this goal, as ReLU neural networks are piece-wise linear functions that do not belong to $W^{m,\infty}$ with $m > 1$. Note that the Bramble-Hilbert Lemma is still applicable in higher-order approximation. Therefore, we need to approximate the function $f_{K,m} = \sum_{|\boldsymbol{\alpha}| \leq n-1} g_{f,\boldsymbol{\alpha},\boldsymbol{m}}(\boldsymbol{x}) \boldsymbol{x}^{\boldsymbol{\alpha}}$ within each domain $\Omega_{\boldsymbol{m}}$ using DNNs, where $g_{f,\boldsymbol{\alpha},\boldsymbol{m}}(x)$ represents a piece-wise constant function within $\Omega_{\boldsymbol{m}}$. Consequently, ReLU-based DNNs can effectively approximate $g_{f,\boldsymbol{\alpha},\boldsymbol{m}}(x)$ within $\Omega_{\boldsymbol{m}}$ when measured by higher-order Sobolev spaces, since both the higher-order derivatives of $g_{f,\boldsymbol{\alpha},\boldsymbol{m}}(x)$ and ReLU-based DNNs are zero. The parts of ReLU-based DNNs that do not have high-order derivatives appear in the domain $\Omega \backslash \Omega_{\boldsymbol{m}}$, and we will use the partition of unity to ensure that this part disappears in the final presentation, this is the reason why it is still acceptable to have ReLU activations appearing in the network. However, when it comes to approximating $\boldsymbol{x}^{\boldsymbol{\alpha}}$ for $|\boldsymbol{\alpha}| > 1$ based on high-order Sobolev norms, ReLU-based DNNs fail to provide accurate results. Therefore, we require DNNs that utilize the square of ReLU activation in this specific scenario. This is how we construct our approach and the reason why we employ both ReLU and the square of ReLU activation functions.

Instead, we examine the use of $\sigma_2$ neural networks for approximating functions measured in the norm of $W^{2,\infty}$. As per Corollary 1, a neural network with $\boldsymbol{O}(N \log N)$ width and $\boldsymbol{O}(L \log L)$ depth can achieve a nonasymptotic approximation rate of $\boldsymbol{O}(N^{-2(n-2)/d} L^{-2(n-2)/d})$ with respect to the $W^{2,\infty}((0,1)^d)$ norm. Moreover, our method can be extended to approximations measured in the norm of $W^{m,\infty}$ with $m > 2$, as shown in Corollary 2. The proof strategy is similar to that used in Subsection 3.1, except that we need to construct a smoother partition of unity rather than $\{g_{\boldsymbol{m}}\}_{\boldsymbol{m} \in \{1,2\}^d}$. The corollaries are presented below, and further details are provided in Appendix 7.3.

**Corollary 1.** *For any $f \in W^{n,\infty}((0,1)^d)$ with $\|f\|_{W^{n,\infty}((0,1)^d)} \leq 1$, any $N, L \in \mathbb{N}_+$ with $NL + 2^{\lfloor \log_2 N \rfloor} \geq \max\{d,n\}$ and $L \geq \lceil \log_2 N \rceil$, there is a $\sigma_2$-NN $\gamma(\boldsymbol{x})$ with the width $2^{d+6} n^{d+1}(N + d) \log_2(8N)$ and depth $15n^2(L+2) \log_2(4L)$ such that*

$$\|f(\boldsymbol{x}) - \gamma(\boldsymbol{x})\|_{W^{2,\infty}((0,1)^d)} \leq 2^{d+7} C_{10}(n,d) N^{-2(n-2)/d} L^{-2(n-2)/d},$$

*where $C_{10}$ is the constant independent with $N, L$.*

**Corollary 2.** *For any $f \in W^{n,\infty}((0,1)^d)$ with $\|f\|_{W^{n,\infty}((0,1)^d)} \leq 1$, any $N, L, m \in \mathbb{N}_+$ with $NL + 2^{\lfloor \log_2 N \rfloor} \geq \max\{d, n\}$ and $L \geq \lceil \log_2 N \rceil$, there is a $\sigma_2$-NN $\varphi(\boldsymbol{x})$ with the width $\boldsymbol{O}(N \log N)$ and depth $\boldsymbol{O}(L \log L)$ such that*

$$\|f(\boldsymbol{x}) - \varphi(\boldsymbol{x})\|_{W^{m,\infty}((0,1)^d)} \leq C_{11}(n, d, m) N^{-2(n-m)/d} L^{-2(n-m)/d},$$

*where $C_{11}$ is the constant independent with $N, L$.*

### 3.3 Optimality of Theorem 3 via estimation of VC-dimension of DNN derivatives (Theorem 1)

In this section, we demonstrate that the approximation rate presented in Theorem 3 is nearly asymptotically optimal:

**Theorem 4.** *Given any $\rho, C_1, C_2, C_3, J_0 > 0$ and $n, d \in \mathbb{N}^+$, there exist $N, L \in \mathbb{N}$ with $NL \geq J_0$ and $f$ with $\|f\|_{W^{n,\infty}((0,1)^d)} \leq 1$, satisfying for any $\sigma_1$-NN $\phi$ with the width smaller than $C_1 N \log N$ and depth smaller than $C_2 L \log L$, we have*

$$|\phi - f|_{W^{1,\infty}((0,1)^d)} > C_3 L^{-2(n-1)/d - \rho} N^{-2(n-1)/d - \rho}. \tag{9}$$

In other words, the approximation rate of $\boldsymbol{O}(N^{-2(n-1)/d - \rho} K^{-2(n-1)/d - \rho})$ cannot be achieved asymptotically when ReLU $\sigma_1$-NNs with width $\boldsymbol{O}(N \log N)$ and depth $\boldsymbol{O}(L \log L)$ to approximate functions in $\mathcal{F}_{n,d} := \{ f \in W^{n,\infty}((0,1)^d) : \|f\|_{W^{n,\infty}((0,1)^d)} \leq 1 \}$. The proof of Theorem 4 is based on the estimation of the VC-dimension of DNN derivatives, which is provided in Theorem 1.

Theorem 1 plays a crucial role in our proof of Theorem 4, which is established through a proof by contradiction following the approach outlined in Ref. [28]. Further details on the proof can be found in Appendix 7.4. The main idea behind the proof is that Theorem 1 characterizes the complexity of DNN derivatives, which in turn limits the ability of DNNs to approximate functions in Sobolev spaces.

In this paper, we focus on the optimality of approximation rate with respect to width $N$ and depth $L$ of DNNs. The dimensionality $d$ is not the focus that we consider in our research. Addressing the question about mitigating the exponential dependence of width on dimensionality, we have observed that this arises from the utilization of methods like Taylor's expansion or average Taylor polynomials in our approximation techniques. It remains an open question for future research to explore alternative approaches to address the challenge of getting the dependence of $d$ in the lower bounds.

## 4 Generalization Analysis in Sobolev Spaces via Estimation of Pseudo-dimension of DNN Derivatives (Theorem 2)

In a typical supervised learning algorithm, the objective is to learn a high-dimensional target function $f(\boldsymbol{x})$ defined on $(0,1)^d$ with $\|f\|_{W^{n,\infty}((0,1)^d)} \leq 1$ from a finite set of data samples $\{(\boldsymbol{x}_i, f(\boldsymbol{x}_i))\}_{i=1}^M$. When training a DNN, we aim to identify a DNN $\phi(\boldsymbol{x}; \boldsymbol{\theta}_S)$ that approximates $f(\boldsymbol{x})$ based on random data samples $\{(\boldsymbol{x}_i, f(\boldsymbol{x}_i))\}_{i=1}^M$. We assume that $\{\boldsymbol{x}_i\}_{i=1}^M$ is an i.i.d. sequence of random variables uniformly distributed on $(0,1)^d$ in this section. Denote

$$\boldsymbol{\theta}_D := \arg\inf_{\boldsymbol{\theta}} \mathcal{R}_D(\boldsymbol{\theta}) := \arg\inf_{\boldsymbol{\theta}} \int_{(0,1)^d} |\nabla(f(\boldsymbol{x}) - \phi(\boldsymbol{x}; \boldsymbol{\theta}))|^2 + |f(\boldsymbol{x}) - \phi(\boldsymbol{x}; \boldsymbol{\theta})|^2 \, \mathrm{d}\boldsymbol{x}, \tag{10}$$

$$\boldsymbol{\theta}_S := \arg\inf_{\boldsymbol{\theta}} \mathcal{R}_S(\boldsymbol{\theta}) := \arg\inf_{\boldsymbol{\theta}} \frac{1}{M} \sum_{i=1}^M \left[ |\nabla(f(\boldsymbol{x}_i) - \phi(\boldsymbol{x}_i; \boldsymbol{\theta}))|^2 + |f(\boldsymbol{x}_i) - \phi(\boldsymbol{x}_i; \boldsymbol{\theta})|^2 \right]. \tag{11}$$

The overall inference error is $\mathbf{E}\mathcal{R}_D(\boldsymbol{\theta}_S)$, which can be divided into two parts:

$$\mathbf{E}\mathcal{R}_D(\boldsymbol{\theta}_S) = \mathcal{R}_D(\boldsymbol{\theta}_D) + \mathbf{E}\mathcal{R}_S(\boldsymbol{\theta}_D) - \mathcal{R}_D(\boldsymbol{\theta}_D) + \mathbf{E}\mathcal{R}_S(\boldsymbol{\theta}_S) - \mathbf{E}\mathcal{R}_S(\boldsymbol{\theta}_D) + \mathbf{E}\mathcal{R}_D(\boldsymbol{\theta}_S) - \mathbf{E}\mathcal{R}_S(\boldsymbol{\theta}_S)$$

$$\leq \underbrace{\mathcal{R}_D(\boldsymbol{\theta}_D)}_{\text{approximation error}} + \underbrace{\mathbf{E}\mathcal{R}_S(\boldsymbol{\theta}_D) - \mathcal{R}_D(\boldsymbol{\theta}_D) + \mathbf{E}\mathcal{R}_D(\boldsymbol{\theta}_S) - \mathbf{E}\mathcal{R}_S(\boldsymbol{\theta}_S)}_{\text{generalization error}}, \tag{12}$$

where the last inequality is due to $\mathbf{E}\mathcal{R}_S(\boldsymbol{\theta}_S) \leq \mathbf{E}\mathcal{R}_S(\boldsymbol{\theta}_D)$ by the definition of $\boldsymbol{\theta}_S$.

Due to Theorem 3, we know that the approximation error $\mathcal{R}_D(\boldsymbol{\theta}_D)$ is a $O(N^{-4(n-1)/d}L^{-4(n-1)/d})$ term since $\|f(\boldsymbol{x}) - \phi(\boldsymbol{x})\|_{H^1((0,1)^d)} \leq \|f(\boldsymbol{x}) - \phi(\boldsymbol{x})\|_{W^{1,\infty}((0,1)^d)}$. In this section, we bound generalization error in the $H^1((0,1)^d)$ sense:

**Theorem 5.** *For any $N, L, d, B, C_1, C_2$, if $\phi(\boldsymbol{x}; \boldsymbol{\theta}_D), \phi(\boldsymbol{x}; \boldsymbol{\theta}_S) \in \widetilde{\Phi}$, we will have that there are constants $C_5 = C_5(B, d, C_1, C_2)$ and $J = J(d, N, L, C_1, C_2)$ such that for any $M \geq J$, we have*

$$\mathbf{E}\mathcal{R}_S(\boldsymbol{\theta}_D) - \mathcal{R}_D(\boldsymbol{\theta}_D) + \mathbf{E}\mathcal{R}_D(\boldsymbol{\theta}_S) - \mathbf{E}\mathcal{R}_S(\boldsymbol{\theta}_S) \leq 2 \sup_{\boldsymbol{\theta}, \phi(\boldsymbol{x};\boldsymbol{\theta}) \in \widetilde{\Phi}} |\mathbf{E}(\mathcal{R}_S(\boldsymbol{\theta})) - \mathcal{R}_D(\boldsymbol{\theta})|$$

$$\leq C_5 \frac{NL(\log_2 L \log_2 N)^{\frac{1}{2}}}{\sqrt{M}} \log M. \qquad (13)$$

*where $\widetilde{\Phi} := \{\phi : \phi \text{ with the width} \leq C_1 N \log N \text{ and depth} \leq C_2 L \log L, \|\phi\|_{W^{1,\infty}((0,1)^d)} \leq B\}$, and $\mathcal{R}_S, \mathcal{R}_D, \boldsymbol{\theta}_S, \boldsymbol{\theta}_D$ are defined in Eqs. (10,11).*

The proof of Theorem 5 is based on the works of [3, 11, 26]. We begin by bounding the generalization error using the Rademacher Complexity and then bound the Rademacher Complexity by the uniform covering number. We further bound the uniform covering number by the pseudo-dimension. Finally, we estimate the pseudo-dimension by Theorem 2. The proof of Theorem 5 is presented in Appendix 7.5

Theorem 2 helps to control the degree of the generalization error with respect to $N$ and $L$ in Theorem 5. In [11], the generalization error is bounded by $O(NL^{\frac{5}{2}})$. In [24], the authors estimate the covering number using the Lipschitz condition of DNNs instead of the pseudo-dimension, leading to a generalization error that is exponentially dependent on the depth of the DNNs. Our result is much better than them due to the optimal estimation of pseudo-dimension of DNN derivatives (Theorem 2).

## 5 Proof Sketches for Theorems 1 and 2

As Theorems 1 and 2 address the estimation of VC-dimension and pseudo-dimension of DNN derivatives, which is the main contribution of this paper, we provide the proofs for these theorems in this section. The distinction in our approach compared to that of [4] lies in the fact that the application of the chain rule requires the consideration of correlations among distinct segments of the deep neural networks, as opposed to treating them as independent components multiplied together.

In the proof of Theorem 1, we use the following lemmas:

**Lemma 1** ([4, Lemma 17],[3, Theorem 8.3]). *Suppose $W \leq M$ and let $P_1, \ldots, P_M$ be polynomials of degree at most $D$ in $W$ variables. Define $K := \left|\{(\text{sgn}(P_1(a)), \ldots, \text{sgn}(P_M(a))) : a \in \mathbb{R}^W\}\right|$, then we have $K \leq 2(2eMD/W)^W$.*

**Lemma 2** ([4, Lemma 18]). *Suppose that $2^m \leq 2^t(mr/w)^w$ for some $r \geq 16$ and $m \geq w \geq t \geq 0$. Then, $m \leq t + w \log_2(2r \log_2 r)$.*

As the proof of Theorem 1 represents the most critical and challenging question in our work, we present a sketch of it below.

*Proof Sketch of Theorem 1.* An element in $\Phi$ can be represented as $\phi = \boldsymbol{W}_{L+1}\sigma_1(\boldsymbol{W}_L\sigma_1(\ldots\sigma_1(\boldsymbol{W}_1\boldsymbol{x} + \boldsymbol{b}_1)\ldots) + \boldsymbol{b}_L) + b_{L+1}$. Therefore, an element in $D\Phi$ can be represented as

$$\psi(\boldsymbol{x}) = D_i\phi(\boldsymbol{x}) = \boldsymbol{W}_{L+1}\sigma_0(\boldsymbol{W}_L\sigma_1(\ldots\sigma_1(\boldsymbol{W}_1\boldsymbol{x} + \boldsymbol{b}_1)\ldots) + \boldsymbol{b}_L)$$
$$\cdot \boldsymbol{W}_L\sigma_0(\ldots\sigma_1(\boldsymbol{W}_1\boldsymbol{x} + \boldsymbol{b}_1)\ldots)\ldots\boldsymbol{W}_2\sigma_0(\boldsymbol{W}_1\boldsymbol{x} + \boldsymbol{b}_1)(\boldsymbol{W}_1)_i, \qquad (14)$$

where $\boldsymbol{W}_i \in \mathbb{R}^{N_i \times N_{i-1}}$ $((\boldsymbol{W})_i$ is $i$-th column of $\boldsymbol{W})$ and $\boldsymbol{b}_i \in \mathbb{R}^{N_i}$ are the weight matrix and the bias vector in the $i$-th linear transform in $\phi$, and $\sigma_0(x) = \text{sgn}(x) = 1[x > 0]$, which is the derivative of the ReLU function and $\sigma_0(\boldsymbol{x}) = \text{diag}(\sigma_0(x_i))$.

Let $\boldsymbol{x} \in \mathbb{R}^d$ be an input and $\boldsymbol{\theta} \in \mathbb{R}^W$ be a parameter vector in $\psi$. We denote the output of $\psi$ with input $\boldsymbol{x}$ and parameter vector $\boldsymbol{\theta}$ as $f(\boldsymbol{x}, \boldsymbol{\theta})$. For fixed $\boldsymbol{x}_1, \boldsymbol{x}_2, \ldots, \boldsymbol{x}_m$ in $\mathbb{R}^d$, we aim to bound

$$K := \left|\{(\text{sgn}(f(\boldsymbol{x}_1, \boldsymbol{\theta})), \ldots, \text{sgn}(f(\boldsymbol{x}_m, \boldsymbol{\theta}))) : \boldsymbol{\theta} \in \mathbb{R}^W\}\right|. \qquad (15)$$

The proof is inspired by [4, Theorem 7]. For any partition $\mathcal{S} = \{P_1, P_2, \ldots, P_T\}$ of the parameter domain $\mathbb{R}^W$, we have $K \leq \sum_{i=1}^{T} |\{(\text{sgn}(f(\boldsymbol{x}_1, \boldsymbol{\theta})), \ldots, \text{sgn}(f(\boldsymbol{x}_m, \boldsymbol{\theta}))) : \boldsymbol{\theta} \in P_i\}|$. We choose the partition such that within each region $P_i$, the functions $f(\boldsymbol{x}_j, \cdot)$ are all fixed polynomials of bounded degree. This allows us to bound each term in the sum using Lemma 1.

The partition of $\mathbb{R}^W$ is constructed layer by layer through successive refinements denoted by $\mathcal{S}_0, \mathcal{S}_1, \ldots, \mathcal{S}_L$. These refinements possess the following properties:

**1**. We have $|\mathcal{S}_0| = 1$, and for each $n = 1, \ldots, L$, we have $\frac{|\mathcal{S}_n|}{|\mathcal{S}_{n-1}|} \leq 2 \left( \frac{2emnN_k}{\sum_{i=1}^{n} W_i} \right)^{\sum_{i=1}^{n} W_i}$.

**2**. For each $n = 0, \ldots, L-1$, each element $S$ of $\mathcal{S}_n$, when $\boldsymbol{\theta}$ varies in $S$, the output of each term in $\mathbb{F}_n$ is a fixed polynomial function in $\sum_{i=1}^{n} W_i$ variables of $\boldsymbol{\theta}$, with a total degree no more than $n + 1$.

**3**. For each element $S$ of $\mathcal{S}_L$, when $\boldsymbol{\theta}$ varies in $S$, the $h$-th term in $\mathbb{F}_L$ for $h \in \{1, 2, \ldots, L+1\}$ is a fixed polynomial function in $W_h$ variables of $\boldsymbol{\theta}$, with a total degree no more than 1. The sequence of sets of functions $\{\mathbb{F}_j\}_{j=0}^{L}$ with respect to parameters $\boldsymbol{\theta} \in \mathbb{R}^W$ is defined as:

$$\mathbb{F}_0 := \{(\boldsymbol{W}_1)_i, \boldsymbol{W}_1 \boldsymbol{x} + \boldsymbol{b}_1\}$$
$$\mathbb{F}_1 := \{(\boldsymbol{W}_1)_i, \boldsymbol{W}_2 \sigma_0(\boldsymbol{W}_1 \boldsymbol{x} + \boldsymbol{b}_1), \boldsymbol{W}_2 \sigma_1(\boldsymbol{W}_1 \boldsymbol{x} + \boldsymbol{b}_1) + \boldsymbol{b}_2\}$$
$$\mathbb{F}_2 := \{(\boldsymbol{W}_1)_i, \boldsymbol{W}_2 \sigma_0(\boldsymbol{W}_1 \boldsymbol{x} + \boldsymbol{b}_1), \boldsymbol{W}_3 \sigma_0(\boldsymbol{W}_2 \sigma_1(\boldsymbol{W}_1 \boldsymbol{x} + \boldsymbol{b}_1) + \boldsymbol{b}_2),$$
$$\boldsymbol{W}_3 \sigma_1(\boldsymbol{W}_2 \sigma_1(\boldsymbol{W}_1 \boldsymbol{x} + \boldsymbol{b}_1) + \boldsymbol{b}_2) + \boldsymbol{b}_3\}$$

$$\vdots$$

$$\mathbb{F}_L := \{(\boldsymbol{W}_1)_i, \boldsymbol{W}_2 \sigma_0(\boldsymbol{W}_1 \boldsymbol{x} + \boldsymbol{b}_1), \ldots, \boldsymbol{W}_{L+1} \sigma_0(\boldsymbol{W}_L \sigma_1(\ldots \sigma_1(\boldsymbol{W}_1 \boldsymbol{x} + \boldsymbol{b}_1) \ldots) + \boldsymbol{b}_L)\}. \quad (16)$$

The details of the refinements required to obtain the partitions are presented in the appendix. By leveraging Property 3 and Lemma 1, we can deduce that $K \leq 2^{L+1} \left( \frac{2em(L+2)(L+1)N}{2U} \right)^U$ where $U = \boldsymbol{O}(N^2 L^2)$, $N$ is the width of the network, and the last inequality is due to weighted AM-GM. For the definition of the VC-dimension, we have $2^{\text{VCdim}(D\Phi)} \leq 2^{L+1} \left( \frac{e\text{VCdim}(D\Phi)(L+1)(L+2)N}{U} \right)^U$. Due to Lemma 2, we obtain that $\text{VCdim}(D\Phi) \leq L + 1 + U \log_2[2(L+1)(L+2)\log_2(L+1)(L+2)] = \boldsymbol{O}(N^2 L^2 \log_2 L \log_2 N)$ since $U = \boldsymbol{O}(N^2 L^2)$. $\qquad\square$

Note that the VC-dimension estimation achieved in Theorem 1 is nearly optimal, as demonstrated in Corollary 3 combined with the upper bound in Theorem 1. If the polynomial degree in the VC-dimension bound as a function of $N$ and $L$ were any smaller, it would contradict Theorem 3, which is based on our proof of Theorem 4.

**Corollary 3.** *For any* $d \in \mathbb{N}_+$, $C, J_0, \varepsilon > 0$, *there exists* $N, L \in \mathbb{N}$ *with* $NL \geq J_0$ *such that*

$$VCdim(D\Phi) > CN^{2-\varepsilon}L^{2-\varepsilon}, \quad (17)$$

*where* $D\Phi$ *is defined in Theorem 1.*

We discuss the proof of Corollary 3 at the end of Section 7.4. Next we now present the proof for Theorem 2.

The proof of Theorem 2 can be found in the appendix. The key idea behind the proof is to reformulate the pseudo-dimension of DNN derivatives as the VC-dimension of a larger set. Based on the definition of pseudo-dimensions, it follows that $\text{Pdim}(D\Phi) \geq \text{VCdim}(D\Phi)$. Consequently, we can derive the following corollary, utilizing Corollary 3, to demonstrate the near optimality of the pseudo-dimension estimate presented in Theorem 2:

**Corollary 4.** *For any* $d \in \mathbb{N}_+$, $C, J_0, \varepsilon > 0$, *there exists* $N, L \in \mathbb{N}$ *with* $NL \geq J_0$ *such that*

$$Pdim(D\Phi) > CN^{2-\varepsilon}L^{2-\varepsilon}, \quad (18)$$

*where* $D\Phi$ *is defined in Theorem 1.*

## 6  Conclusions and Discussions

In this paper, we establish nearly optimal bounds for the VC-dimension and pseudo-dimension of DNN derivatives. Based on these bounds, two contributions to Sobolev training [8, 40, 46] are made in

this paper. Firstly, we show that the optimal approximation rate of DNNs with a width of $\boldsymbol{O}(N \log N)$ and a depth of $\boldsymbol{O}(L \log L)$ is $\boldsymbol{O}(N^{-2(n-1)/d} L^{-2(n-1)/d})$ in Sobolev spaces. This demonstrates the ability of DNNs to learn target functions well in Sobolev training. Secondly, we find that the degree of the pseudo-dimension of DNN derivatives is the same as that for DNNs corresponding to the width $N$ and depth $L$ of DNNs. This result suggests that despite the apparent complexity of DNN derivatives, the degree of generalization error of loss functions containing derivatives of DNNs is equivalent to that without derivatives, corresponding to the width $N$ and depth $L$ of DNNs. As a result, we do not need to use a significantly larger number of sample points to learn the target function in Sobolev training compared to regular training.

The estimations of the VC-dimension and pseudo-dimension of DNN derivatives have broad applications in deep learning research. For example, in classification tasks, the VC-dimension characterizes the uniform convergence of misclassification frequencies to probabilities and asymptotically determines the sample complexity of PAC learning [4, 44, 6]. These applications can be explored in the further work. Our focus in this paper is on the Sobolev training with loss functions containing first-order derivatives, and we also obtain the approximation rate of $\sigma_2$-NNs described by higher-order Sobolev norms (Corollaries 1 and 2). The optimality of these results and the generalization error of Sobolev training with loss functions containing higher-order derivatives of DNNs remain open problems, as estimating the VC-dimension and pseudo-dimension of higher-order derivatives of $\sigma_2$-NNs requires further investigation.

## Acknowledgments and Disclosure of Funding

This work was done during Y.Y.'s visit under the supervision of Prof. H.Y., in the Department of Mathematics, University of Maryland College Park. The work of H. Y. was partially supported by the US National Science Foundation under award DMS-2244988, DMS-2206333, and the Office of Naval Research Award N00014-23-1-2007. The work of Y.X. was supported by the Project of Hetao Shenzhen-HKUST Innovation Cooperation Zone HZQB-KCZYB-2020083.

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

# 7 Supplementary Material

## 7.1 Proofs of Theorems 1 and 2.

*Proof of Theorem 1.* An element in $\Phi$ can be represented as $\phi = \boldsymbol{W}_{L+1}\sigma_1(\boldsymbol{W}_L\sigma_1(\ldots\sigma_1(\boldsymbol{W}_1\boldsymbol{x} + \boldsymbol{b}_1)\ldots) + \boldsymbol{b}_L) + b_{L+1}$. Therefore, an element in $D\Phi$ can be represented as

$$\psi(\boldsymbol{x}) = D_i\phi(\boldsymbol{x}) = \boldsymbol{W}_{L+1}\sigma_0(\boldsymbol{W}_L\sigma_1(\ldots\sigma_1(\boldsymbol{W}_1\boldsymbol{x} + \boldsymbol{b}_1)\ldots) + \boldsymbol{b}_L)$$
$$\cdot \boldsymbol{W}_L\sigma_0(\ldots\sigma_1(\boldsymbol{W}_1\boldsymbol{x} + \boldsymbol{b}_1)\ldots)\ldots\boldsymbol{W}_2\sigma_0(\boldsymbol{W}_1\boldsymbol{x} + \boldsymbol{b}_1)(\boldsymbol{W}_1)_i, \quad (19)$$

where $\boldsymbol{W}_i \in \mathbb{R}^{N_i \times N_{i-1}}$ ($(\boldsymbol{W})_i$ is $i$-th column of $\boldsymbol{W}$) and $\boldsymbol{b}_i \in \mathbb{R}^{N_i}$ are the weight matrix and the bias vector in the $i$-th linear transform in $\phi$, and $\sigma_0(x) = \text{sgn}(x) = 1[x > 0]$, which is the derivative of the ReLU function and $\sigma_0(\boldsymbol{x}) = \text{diag}(\sigma_0(x_i))$. Denote $W_i$ as the number of parameters in $\boldsymbol{W}_i, \boldsymbol{b}_i$, i.e., $W_i = N_i N_{i-1} + N_i$.

Let $\boldsymbol{x} \in \mathbb{R}^d$ be an input and $\boldsymbol{\theta} \in \mathbb{R}^W$ be a parameter vector in $\psi$. We denote the output of $\psi$ with input $\boldsymbol{x}$ and parameter vector $\boldsymbol{\theta}$ as $f(\boldsymbol{x}, \boldsymbol{\theta})$. For fixed $\boldsymbol{x}_1, \boldsymbol{x}_2, \ldots, \boldsymbol{x}_m$ in $\mathbb{R}^d$, we aim to bound

$$K := \left|\{(\text{sgn}(f(\boldsymbol{x}_1, \boldsymbol{\theta})), \ldots, \text{sgn}(f(\boldsymbol{x}_m, \boldsymbol{\theta}))) : \boldsymbol{\theta} \in \mathbb{R}^W\}\right|. \quad (20)$$

The proof is inspired by [4, Theorem 7]. For any partition $\mathcal{S} = \{P_1, P_2, \ldots, P_T\}$ of the parameter domain $\mathbb{R}^W$, we have $K \le \sum_{i=1}^T |\{(\text{sgn}(f(\boldsymbol{x}_1, \boldsymbol{\theta})), \ldots, \text{sgn}(f(\boldsymbol{x}_m, \boldsymbol{\theta}))) : \boldsymbol{\theta} \in P_i\}|$. We choose the partition such that within each region $P_i$, the functions $f(\boldsymbol{x}_j, \cdot)$ are all fixed polynomials of bounded degree. This allows us to bound each term in the sum using Lemma 1.

We define a sequence of sets of functions $\{\mathbb{F}_j\}_{j=0}^L$ with respect to parameters $\boldsymbol{\theta} \in \mathbb{R}^W$:

$$\mathbb{F}_0 := \{(\boldsymbol{W}_1)_i, \boldsymbol{W}_1\boldsymbol{x} + \boldsymbol{b}_1\}$$
$$\mathbb{F}_1 := \{(\boldsymbol{W}_1)_i, \boldsymbol{W}_2\sigma_0(\boldsymbol{W}_1\boldsymbol{x} + \boldsymbol{b}_1), \boldsymbol{W}_2\sigma_1(\boldsymbol{W}_1\boldsymbol{x} + \boldsymbol{b}_1) + \boldsymbol{b}_2\}$$
$$\mathbb{F}_2 := \{(\boldsymbol{W}_1)_i, \boldsymbol{W}_2\sigma_0(\boldsymbol{W}_1\boldsymbol{x} + \boldsymbol{b}_1), \boldsymbol{W}_3\sigma_0(\boldsymbol{W}_2\sigma_1(\boldsymbol{W}_1\boldsymbol{x} + \boldsymbol{b}_1) + \boldsymbol{b}_2),$$
$$\boldsymbol{W}_3\sigma_1(\boldsymbol{W}_2\sigma_1(\boldsymbol{W}_1\boldsymbol{x} + \boldsymbol{b}_1) + \boldsymbol{b}_2) + \boldsymbol{b}_3\}$$
$$\vdots$$
$$\mathbb{F}_L := \{(\boldsymbol{W}_1)_i, \boldsymbol{W}_2\sigma_0(\boldsymbol{W}_1\boldsymbol{x} + \boldsymbol{b}_1), \ldots, \boldsymbol{W}_{L+1}\sigma_0(\boldsymbol{W}_L\sigma_1(\ldots\sigma_1(\boldsymbol{W}_1\boldsymbol{x} + \boldsymbol{b}_1)\ldots) + \boldsymbol{b}_L)\}. \quad (21)$$

The partition of $\mathbb{R}^W$ is constructed layer by layer through successive refinements denoted by $\mathcal{S}_0, \mathcal{S}_1, \ldots, \mathcal{S}_L$. These refinements possess the following properties:

**1**. We have $|\mathcal{S}_0| = 1$, and for each $n = 1, \ldots, L$, we have $\frac{|\mathcal{S}_n|}{|\mathcal{S}_{n-1}|} \le 2\left(\frac{2emnN_k}{\sum_{i=1}^n W_i}\right)^{\sum_{i=1}^n W_i}$.

**2**. For each $n = 0, \ldots, L-1$, each element $S$ of $\mathcal{S}_n$, when $\boldsymbol{\theta}$ varies in $S$, the output of each term in $\mathbb{F}_n$ is a fixed polynomial function in $\sum_{i=1}^n W_i$ variables of $\boldsymbol{\theta}$, with a total degree no more than $n + 1$.

**3**. For each element $S$ of $\mathcal{S}_L$, when $\boldsymbol{\theta}$ varies in $S$, the $h$-th term in $\mathbb{F}_L$ for $h \in \{1, 2, \ldots, L+1\}$ is a fixed polynomial function in $W_h$ variables of $\boldsymbol{\theta}$, with a total degree no more than 1.

We define $\mathcal{S}_0 = \{\mathbb{R}^W\}$, which satisfies properties 1,2 above, since $\boldsymbol{W}_1\boldsymbol{x}_j + \boldsymbol{b}_1$ and $(\boldsymbol{W}_1)_i$ are affine functions of $\boldsymbol{W}_1, \boldsymbol{b}_1$.

To define $\mathcal{S}_n$, we use the last term of $\mathbb{F}_{n-1}$ as inputs for the last two terms in $\mathbb{F}_n$. Assuming that $\mathcal{S}_0, \mathcal{S}_1, \ldots, \mathcal{S}_{n-1}$ have already been defined, we observe that the last two terms are new additions to $\mathbb{F}_n$ when comparing it to $\mathbb{F}_{n-1}$. Therefore, all elements in $\mathbb{F}_n$ except the last two are fixed polynomial functions in $W_n$ variables of $\boldsymbol{\theta}$, with a total degree no greater than $n$ when $\boldsymbol{\theta}$ varies in $S \in \mathcal{S}_n$. This is because $\mathcal{S}_n$ is a finer partition than $\mathcal{S}_{n-1}$.

We denote $p_{\boldsymbol{x}_j, n-1, S, k}(\boldsymbol{\theta})$ as the output of the $k$-th node in the last term of $\mathbb{F}_{n-1}$ in response to $\boldsymbol{x}_j$ when $\boldsymbol{\theta} \in S$. The collection of polynomials

$$\{p_{\boldsymbol{x}_j, n-1, S, k}(\boldsymbol{\theta}) : j = 1, \ldots, m, \ k = 1, \ldots, N_n\}$$

can attain at most $2\left(\frac{2emnN_n}{\sum_{i=1}^n W_i}\right)^{\sum_{i=1}^n W_i}$ distinct sign patterns when $\boldsymbol{\theta} \in S$ due to Lemma 1 for sufficiently large $m$. Therefore, we can divide $S$ into $2\left(\frac{2emnN_n}{\sum_{i=1}^n W_i}\right)^{\sum_{i=1}^n W_i}$ parts, each having the

property that $p_{\boldsymbol{x}_j, n-1, S, k}(\boldsymbol{\theta})$ does not change sign within the subregion. By performing this for all $S \in \mathcal{S}_{n-1}$, we obtain the desired partition $\mathcal{S}_n$. This division ensures that the required property 1 is satisfied.

Additionally, since the input to the last two terms in $\mathbb{F}_n$ is $p_{\boldsymbol{x}_j, n-1, S, k}(\boldsymbol{\theta})$, and we have shown that the sign of this input will not change in each region of $\mathcal{S}_n$, it follows that the output of the last two terms in $\mathbb{F}_n$ is also a polynomial without breakpoints in each element of $\mathcal{S}_n$. Therefore, the required property 2 is satisfied.

In the context of DNNs, the last layer is characterized by all terms containing the activation function $\sigma_0$. Consequently, for any element $S$ of the partition $\mathcal{S}_L$, when the vector of parameters $\boldsymbol{\theta}$ varies within $S$, the $h$-th term in $\mathbb{F}_L$ for $h \in \{1, 2, \ldots, L+1\}$ can be expressed as a polynomial function of at most degree 1, which depends on at most $W_h$ variables of $\boldsymbol{\theta}$. Hence, the required property 3 is satisfied.

Due to property 3, we multiply all the terms in $\mathbb{F}_L$ and obtain a term in $D\Phi$. Hence, the output of each term in $D\Phi$ is a polynomial function in $\sum_{i=1}^{L+1} W_i$ variables of $\boldsymbol{\theta} \in S \in \mathcal{S}_L$, of total degree no more than $L+1$. Therefore, for each $S \in \mathcal{S}_L$ we have $|\{(\mathrm{sgn}(f(\boldsymbol{x}_1, \boldsymbol{\theta})), \ldots, \mathrm{sgn}(f(\boldsymbol{x}_m, \boldsymbol{\theta}))) : \boldsymbol{\theta} \in S\}| \leq 2\left(2em(L+1)/\sum_{i=1}^{L+1} W_i\right)^{\sum_{i=1}^{L+1} W_i}$. Then

$$
K \leq 2\left(2em(L+1)/\sum_{i=1}^{L+1} W_i\right)^{\sum_{i=1}^{L+1} W_i} \cdot \prod_{n=1}^{L} 2\left(\frac{2emnN_n}{\sum_{i=1}^{n} W_i}\right)^{\sum_{i=1}^{n} W_i} \leq \prod_{n=1}^{L+1} 2\left(\frac{2emnN_n}{\sum_{i=1}^{n} W_i}\right)^{\sum_{i=1}^{n} W_i}
$$

$$
\leq 2^{L+1}\left(\frac{2em(L+2)(L+1)N}{2U}\right)^{U} \tag{22}
$$

where $U := \sum_{n=1}^{L+1} \sum_{i=1}^{n} W_i = \boldsymbol{O}(N^2 L^2)$, $N$ is the width of the network, and the last inequality is due to weighted AM-GM. For the definition of the VC-dimension, we have

$$
2^{\mathrm{VCdim}(D\Phi)} \leq 2^{L+1}\left(\frac{e\mathrm{VCdim}(D\Phi)(L+1)(L+2)N}{U}\right)^{U}. \tag{23}
$$

Due to Lemma 2, we obtain that

$$
\mathrm{VCdim}(D\Phi) \leq L + 1 + U \log_2[2(L+1)(L+2)\log_2(L+1)(L+2)] = \boldsymbol{O}(N^2 L^2 \log_2 L \log_2 N) \tag{24}
$$

since $U = \boldsymbol{O}(N^2 L^2)$. $\qquad\square$

*Proof of Theorem 2.* Denote $D\Phi_{\mathcal{N}} := \{\eta(\boldsymbol{x}, y) : \eta(\boldsymbol{x}, y) = \psi(\boldsymbol{x}) - y, \psi \in D\Phi, (\boldsymbol{x}, y) \in \mathbb{R}^{d+1}\}$. Based on the definition of VC-dimension and pseudo-dimension, we have that

$$
\mathrm{Pdim}(D\Phi) \leq \mathrm{VCdim}(D\Phi_{\mathcal{N}}). \tag{25}
$$

For the $\mathrm{VCdim}(D\Phi_{\mathcal{N}})$, it can be bounded by $O(N^2 L^2 \log_2 L \log_2 N)$. The proof is similar to that for the estimate of $\mathrm{VCdim}(D\Phi)$ as given in Theorem 1. $\qquad\square$

We establish that $\mathrm{Pdim}(D\Phi) \leq \mathrm{VCdim}(D\Phi_{\mathcal{N}})$, where $\Phi_{\mathcal{N}}$ represents DNNs with $N+1$ width and $L+1$ depth. This implies that $\mathrm{Pdim}(D\Phi)$ is upper bounded by $\bar{C}(N+1)^2(L+1)^2 \log_2(L+1) \log_2(N+1) \leq 64\bar{C}N^2 L^2 \log_2 L \log_2 N$. Therefore, we conclude that $64\bar{C} \geq \hat{C}$.

## 7.2 Proof of Theorem 3

### 7.2.1 Propositions of Sobolev spaces and ReLU neural networks

The following two lemmas estimate the Sobolev norms and Sobolev semi-norms for the composition and product, which will be used in later proof.

**Lemma 3** ([18, Corollary B.5]). *Let $d, m \in \mathbb{N}_+$ and $\Omega_1 \subset \mathbb{R}^d$ and $\Omega_2 \subset \mathbb{R}^m$ both be open, bounded, and convex. Then for $\boldsymbol{f} \in W^{1,\infty}(\Omega_1, \mathbb{R}^m)$ and $g \in W^{1,\infty}(\Omega_2)$ with $\mathrm{ran}\boldsymbol{f} \subset \Omega_2$, we have*

$$
\|g \circ \boldsymbol{f}\|_{W^{1,\infty}(\Omega_2)} \leq \sqrt{d m} \max\{\|g\|_{L^\infty(\Omega_2)}, |g|_{W^{1,\infty}(\Omega_2)} |\boldsymbol{f}|_{W^{1,\infty}(\Omega_1, \mathbb{R}^m)}\}.
$$

**Lemma 4** ([18, Corollary B.6]). *Let $d \in \mathbb{N}_+$ and $\Omega \subset \mathbb{R}^d$. Then for $f, g \in W^{1,\infty}(\Omega)$, we have*

$$\|gf\|_{W^{1,\infty}(\Omega)} \leq \|g\|_{L^\infty(\Omega)}|f|_{W^{1,\infty}(\Omega)} + \|f\|_{L^\infty(\Omega)}|g|_{W^{1,\infty}(\Omega)}.$$

Then we collect and establish some propositions for ReLU neural networks.

**Proposition 2** ([28, Proposition 4.3]). *Given any $N, L \in \mathbb{N}_+$ and $\delta \in \left(0, \frac{1}{3K}\right]$ for $K = \lfloor N^{1/d} \rfloor^2 \lfloor L^{2/d} \rfloor$, there exists a $\sigma_1$-NN $\phi$ with the width $4N + 5$ and depth $4L + 4$ such that*

$$\phi(x) = k, x \in \left[\frac{k}{K}, \frac{k+1}{K} - \delta \cdot 1_{k<K-1}\right], \ k = 0, 1, \ldots, K - 1.$$

**Proposition 3.** *[28, Proposition 4.4] Given any $N, L, s \in \mathbb{N}_+$ and $\xi_i \in [0,1]$ for $i = 0, 1, \ldots N^2 L^2 - 1$, there exists a $\sigma_1$-NN $\phi$ with the width $16s(N+1)\log_2(8N)$ and depth $(5L+2)\log_2(4L)$ such that*

*1. $|\phi(i) - \xi_i| \leq N^{-2s}L^{-2s}$ for $i = 0, 1, \ldots N^2 L^2 - 1$.*

*2. $0 \leq \phi(x) \leq 1$, $x \in \mathbb{R}$.*

**Proposition 4.** *For any $N, L \in \mathbb{N}_+$ and $a > 0$, there is a $\sigma_1$-NN $\phi$ with the width $15N$ and depth $2L$ such that $\|\phi\|_{W^{1,\infty}((-a,a)^2)} \leq 12a^2$ and*

$$\|\phi(x,y) - xy\|_{W^{1,\infty}((-a,a)^2)} \leq 6a^2 N^{-L}. \tag{26}$$

*Furthermore,*

$$\phi(0,y) = \frac{\partial \phi(0,y)}{\partial y} = 0, \ y \in (-a,a). \tag{27}$$

*Proof.* We first need to construct a neural network to approximate $x^2$ on $(-1,1)$, and the idea is similar with [23, Lemma 3.2] and [28, Lemma 5.1]. The reason we do not use [23, Lemma 3.4] and [28, Lemma 4.2] directly is that constructing $\phi(x,y)$ by translating a neural network in $W^{1,\infty}[0,1]$ will lose the proposition of $\phi(0.y) = 0$. Here we need to define teeth functions $T_i$ on $\widetilde{x} \in [-1,1]$:

$$T_1(\widetilde{x}) = \begin{cases} 2|\widetilde{x}|, & |\widetilde{x}| \leq \frac{1}{2}, \\ 2(1 - |\widetilde{x}|), & |\widetilde{x}| > \frac{1}{2}, \end{cases}$$

and

$$T_i = T_{i-1} \circ T_1, \quad \text{for } i = 2, 3, \cdots.$$

Define

$$\widetilde{\psi}(\widetilde{x}) = \widetilde{x} - \sum_{i=1}^{s} \frac{T_i(\widetilde{x})}{2^{2i}},$$

According to [23, Lemma 3.2] and [28, Lemma 5.1], we know $\psi$ is a neural network with the width $5N$ and depth $2L$ such that $\|\widetilde{\psi}(\widetilde{x})\|_{W^{1,\infty}((-1,1))} \leq 2$, $\|\widetilde{\psi}(\widetilde{x}) - \widetilde{x}^2\|_{W^{1,\infty}((-1,1))} \leq N^{-L}$ and $\psi(0) = 0$.

By setting $x = a\widetilde{x} \in (-a,a)$ for $\widetilde{x} \in (-1,1)$, we define

$$\psi(x) = a^2 \widetilde{\psi}\left(\frac{x}{a}\right).$$

Note that $x^2 = a^2 \left(\frac{x}{a}\right)^2$, we have

$$\|\psi(x) - x^2\|_{\mathcal{W}^{1,\infty}(-a,a)} = a^2 \left\|\widetilde{\psi}\left(\frac{x}{a}\right) - \left(\frac{x}{a}\right)^2\right\|_{\mathcal{W}^{1,\infty}((-a,a))}$$
$$\leq a^2 N^{-L},$$

and $\psi(0) = 0$, which will be used to prove Eq. (27).

Then we can construct $\phi(x,y)$ as

$$\phi(x,y) = 2\left[\psi\left(\frac{|x+y|}{2}\right) - \psi\left(\frac{|x|}{2}\right) - \psi\left(\frac{|y|}{2}\right)\right] \tag{28}$$

where $\phi(x)$ is a neural network with the width $15N$ and depth $2L$ such that $\|\phi\|_{W^{1,\infty}((-a,a)^2)} \leq 12a^2$ and

$$\|\phi(x,y) - xy\|_{W^{1,\infty}((-a,a)^2)} \leq 6a^2 N^{-L}. \tag{29}$$

For the last equation Eq. (27) is due to $\phi(x,y)$ in the proof can be read as Eq. (28) with $\psi(0) = 0$. $\quad\square$

**Proposition 5.** *For any $N, L, s \in \mathbb{N}_+$ with $s \geq 2$, there exists a $\sigma_1$-NN $\phi$ with the width $9(N+1) + s - 1$ and depth $14s(s-1)L$ such that $\|\phi\|_{\mathcal{W}^{1,\infty}((0,1)^s)} \leq 18$ and*

$$\|\phi(\boldsymbol{x}) - x_1 x_2 \cdots x_s\|_{\mathcal{W}^{1,\infty}((0,1)^s)} \leq 10(s-1)(N+1)^{-7sL}. \tag{30}$$

*Furthermore, for any $i = 1, 2, \ldots, s$, if $x_i = 0$, we will have*

$$\phi(x_1, x_2, \ldots, x_{i-1}, 0, x_{i+1}, \ldots, x_s) = \frac{\partial \phi(x_1, x_2, \ldots, x_{i-1}, 0, x_{i+1}, \ldots, x_s)}{\partial x_j} = 0, \; i \neq j. \tag{31}$$

*Proof.* The proof of the first inequality Eq. (30) can be found in [23, Lemma 3.5]. The proof of Eq. (31) can be obtained via induction. For $s = 2$, based on Proposition 4, we know there is a neural network $\phi_2$ satisfied Eq. (31).

Now assume that for any $i \leq n-1$, there is a neural network $\phi_i$ satisfied Eq. (31). $\phi_n$ in [23] is constructed as

$$\phi_n(x_1, x_2, \ldots, x_n) = \phi_2(\phi_{n-1}(x_1, x_2, \ldots, x_{n-1}), \sigma(x_n)), \tag{32}$$

which satisfies Eq. (30). Then $\phi_n(x_1, x_2, \ldots, x_{i-1}, 0, x_{i+1}, \ldots, x_n) = 0$ for any $i = 1, 2, \ldots, n$. For $i = n$, we have

$$\frac{\phi(x_1, x_2, \ldots, 0)}{\partial x_j} = \underbrace{\frac{\partial \phi_2(\phi_{n-1}(x_1, x_2, \ldots, x_{n-1}), 0)}{\partial \phi_{n-1}(x_1, x_2, \ldots, x_{n-1})}}_{=0,\text{ by the property of } \phi_2.} \cdot \frac{\partial \phi_{n-1}(x_1, x_2, \ldots, x_{n-1})}{\partial x_j} = 0. \tag{33}$$

For $i < n$ and $j < n$, we have

$$\frac{\phi(x_1, x_2, \ldots, x_{i-1}, 0, x_{i+1}, \ldots, x_n)}{\partial x_j}$$
$$= \frac{\partial \phi_2(\phi_{n-1}(x_1, x_2, \ldots, x_{i-1}, 0, x_{i+1}, \ldots, x_{n-1}), \sigma(x_n))}{\partial \phi_{n-1}(x_1, \ldots, 0, x_{i+1}, \ldots, x_{n-1})} \cdot \underbrace{\frac{\partial \phi_{n-1}(x_1, \ldots, 0, x_{i+1}, \ldots, x_{n-1})}{\partial x_j}}_{=0,\text{ via induction.}} = 0.$$
$$\tag{34}$$

For $i < n$ and $j = n$, we have

$$\frac{\phi(x_1, x_2, \ldots, x_{i-1}, 0, x_{i+1}, \ldots, x_n)}{\partial x_n}$$
$$= \underbrace{\frac{\partial \phi_2(\phi_{n-1}(x_1, x_2, \ldots, x_{i-1}, 0, x_{i+1}, \ldots, x_{n-1}), \sigma(x_n))}{\partial \sigma(x_n)}}_{=0,\text{ by the property of } \phi_2.} \cdot \frac{d\sigma(x_n)}{dx_n} = 0. \tag{35}$$

Therefore, Eq. (31) is valid. $\quad\square$

**Proposition 6** ([23, Propositiion 3.6]). *For any $N, L, s \in \mathbb{N}_+$ and $|\boldsymbol{\alpha}| \leq s$, there is a $\sigma_1$-NN $\phi$ with the width $9(N+1) + s - 1$ and depth $14s^2 L$ such that $\|\phi\|_{W^{1,\infty}((0,1)^d)} \leq 18$ and*

$$\|\phi(\boldsymbol{x}) - \boldsymbol{x}^{\boldsymbol{\alpha}}\|_{W^{1,\infty}((0,1)^d)} \leq 10s(N+1)^{-7sL}. \tag{36}$$

**Proposition 7** ([39, Proposition 1]). *Given a sequence of the neural network $\{p_i\}_{i=1}^M$, and each $p_i$ is a $\sigma_1$-NN from $\mathbb{R} \to \mathbb{R}$ with the width $N$ and depth $L_i$, then $\sum_{i=1}^M p_i$ is a $\sigma_1$-NN with the width $N+4$ and depth $\sum_{i=1}^M L_i$.*

We present the proof of Proposition 1 below.

*Proof of Proposition 1.* First, we construct $g_1$ and $g_2$ by neural networks in $[0, 1]$. Note that $\lfloor L^{2/d} \rfloor \leq L^{2/d} \leq \left( \lfloor L^{1/d} \rfloor + 1 \right)^2$. We first construct a $\sigma_1$-NN in the small set $\left[ 0, \lfloor N^{1/d} \rfloor \lfloor L^{2/d} \rfloor \right]$. It is easy to check there is a neural network $\hat{\psi}$ with the width 4 and one layer such as

$$\hat{\psi}(x) := \begin{cases} 1, & x \in \left[ \frac{1}{8K}, \frac{3}{8K} \right] \\ 4K \left( x - \frac{1}{8K} \right), & x \in \left[ \frac{1}{8K}, \frac{3}{8K} \right] \\ -4K \left( x - \frac{7}{8K} \right), & x \in \left[ \frac{5}{8K}, \frac{7}{8K} \right] \\ 0, & \text{Otherwise.} \end{cases} \tag{37}$$

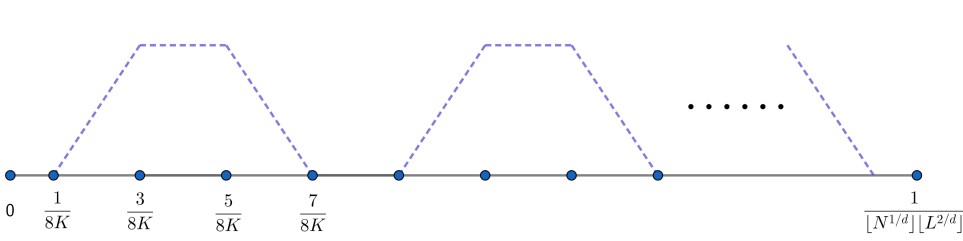

Figure 2: $\psi_1$

Hence, we have a network $\psi_1$ with the width $4 \lfloor N^{1/d} \rfloor$ and one layer such as

$$\psi_1(x) := \sum_{i=0}^{\lfloor N^{1/d} \rfloor - 1} \hat{\psi} \left( x - \frac{i}{K} \right).$$

Next, we construct $\psi_i$ for $i = 2, 3, 4$ based on the symmetry and periodicity of $g_i$. $\psi_2$ is the function with period $\frac{2}{\lfloor N^{1/d} \rfloor \lfloor L^{2/d} \rfloor}$ in $\left[ 0, \frac{1}{\lfloor L^{2/d} \rfloor} \right]$, and each period is a hat function with gradient 1. $\psi_3$ is the function with period $\frac{2}{\lfloor L^{2/d} \rfloor}$ in $\left[ 0, \frac{\lfloor L^{1/d} \rfloor + 1}{\lfloor L^{2/d} \rfloor} \right]$, and each period is a hat function with gradient 1. $\psi_4$ is the function with period $\frac{2(\lfloor L^{1/d} \rfloor + 1)}{\lfloor L^{2/d} \rfloor}$ in $\left[ 0, \frac{(\lfloor L^{1/d} \rfloor + 1)^2}{\lfloor L^{2/d} \rfloor} \right]$, and each period is a hat function with gradient 1. The schematic diagram is in Fig. 3 (The diagram is shown the case for $\lfloor N^{1/d} \rfloor$ and $\lfloor L^{1/d} \rfloor + 1$ is a even integer.).

Note that $\psi_2 \circ \psi_3 \circ \psi_4(x)$ is the function with period $\frac{2}{\lfloor N^{1/d} \rfloor \lfloor L^{2/d} \rfloor}$ in $[0, 1] \subset \left[ 0, \frac{(\lfloor L^{1/d} \rfloor + 1)^2}{\lfloor L^{2/d} \rfloor} \right]$, and each period is a hat function with gradient 1. Then function $\psi_1 \circ \psi_2 \circ \psi_3 \circ \psi_4(x)$ is obtained by repeating reflection $\psi_1$ in $\left[ 0, \frac{(\lfloor L^{1/d} \rfloor + 1)^2}{\lfloor L^{2/d} \rfloor} \right]$, which is the function we want.

Similar with $\psi_1$, $\psi_2$ is a network with $4 \lfloor N^{1/d} \rfloor$ width and one layer. Due to Proposition 7, we know that $\psi_3$ and $\psi_4$ is a network with 7 width and $\lfloor L^{1/d} \rfloor + 1$ depth. Hence

$$\psi(x) := \psi_1 \circ \psi_2 \circ \psi_3 \circ \psi_4(x) \tag{38}$$

is a network with $4 \lfloor N^{1/d} \rfloor$ width and $2 \lfloor L^{1/d} \rfloor + 4$ depth and $g_1 = \psi \left( x + \frac{1}{8K} \right)$ and $g_1 = \psi \left( x + \frac{5}{8K} \right)$.

Now we can construct $g_{\boldsymbol{m}}$ for $\boldsymbol{m} \in \{1, 2\}^d$ based on Proposition 5: There is a neural network $\phi_{\text{prod}}$ with the width $9(N + 1) + d - 1$ and depth $14d(d - 1)nL$ such that $\| \phi_{\text{prod}} \|_{\mathcal{W}^{1,\infty}((0,1)^d)} \leq 18$ and

$$\| \phi_{\text{prod}}(\boldsymbol{x}) - x_1 x_2 \cdots x_d \|_{\mathcal{W}^{1,\infty}((0,1)^d)} \leq 10(d - 1)(N + 1)^{-7dnL}.$$

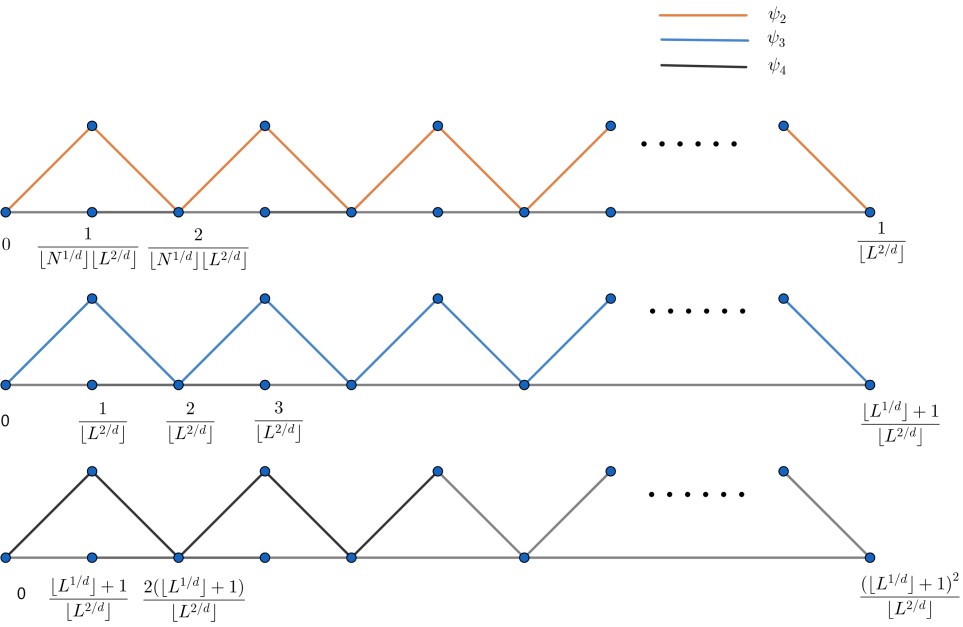

Figure 3: $\psi_i$ for $i = 2, 3, 4$

Then denote $\phi_{\boldsymbol{m}}(\boldsymbol{x}) := \phi_{\text{prod}}(g_{m_1}, g_{m_2}, \ldots, g_{m_d})$ which is a neural network with the width smaller than $(9 + d)(N + 1) + d - 1$ and depth smaller than $15d(d-1)nL$. Furthermore, due to Lemma 3, we have

$$
\begin{aligned}
\|\phi_{\boldsymbol{m}}(\boldsymbol{x}) - g_{\boldsymbol{m}}(\boldsymbol{x})\|_{\mathcal{W}^{1,\infty}((0,1)^d)} \leq & d^{\frac{3}{2}} \left\|\phi_{\text{prod}}(\boldsymbol{x}) - x_1 x_2 \cdots x_d\right\|_{L^\infty((0,1)^d)} \\
& + d^{\frac{3}{2}} \left\|\phi_{\text{prod}}(\boldsymbol{x}) - x_1 x_2 \cdots x_d\right\|_{\mathcal{W}^{1,\infty}((0,1)^d)} |\psi|_{W^{1,\infty}(0,1)} \\
\leq & d^{\frac{3}{2}} 10(d-1)(N+1)^{-7ndL} \left(1 + 4\lfloor N^{1/d}\rfloor^2 \lfloor L^{2/d}\rfloor\right) \\
\leq & 50 d^{\frac{5}{2}} (N+1)^{-4dnL}, \tag{39}
\end{aligned}
$$

where the last inequality is due to

$$
\frac{\lfloor N^{1/d}\rfloor^2 \lfloor L^{2/d}\rfloor}{(N+1)^{3dnL}} \leq \frac{N^2 L^2}{(N+1)^{3dnL}} \leq \frac{L^2}{(N+1)^{3dnL-2}} \leq \frac{L^2}{2^{dnL}} \leq 1.
$$

$\square$

In the final of this subsection, we establish three lemmas for $\{\Omega_{\boldsymbol{m}}\}_{\boldsymbol{m} \in \{1,2\}^d}$, $\{g_{\boldsymbol{m}}\}_{\boldsymbol{m} \in \{1,2\}^d}$ and $\{\phi_{\boldsymbol{m}}\}_{\boldsymbol{m} \in \{1,2\}^d}$ defined in Subsection 3.1.

**Lemma 5.** *For $\{\Omega_{\boldsymbol{m}}\}_{\boldsymbol{m} \in \{1,2\}^d}$ defined in Definition 5, we have*

$$
\bigcup_{\boldsymbol{m} \in \{1,2\}^d} \Omega_{\boldsymbol{m}} = [0, 1]^d.
$$

*Proof.* We prove this lemma via induction. $d = 1$ is valid due to $\Omega_1 \cup \Omega_2 = [0, 1]$. Assume that the lemma is true for $d - 1$, then

$$
\begin{aligned}
\bigcup_{\boldsymbol{m} \in \{1,2\}^d} \Omega_{\boldsymbol{m}} = [0,1]^d = & \bigcup_{\boldsymbol{m} \in \{1,2\}^{d-1}} \Omega_{\boldsymbol{m}} \times \Omega_1 + \bigcup_{\boldsymbol{m} \in \{1,2\}^{d-1}} \Omega_{\boldsymbol{m}} \times \Omega_2 \\
= & \left([0,1]^{d-1} \times \Omega_1\right) \bigcup \left([0,1]^{d-1} \times \Omega_2\right) = [0,1]^d, \tag{40}
\end{aligned}
$$

hence the case of $d$ is valid, and we finish the proof of the lemma. $\square$

**Lemma 6.** $\{g_{\boldsymbol{m}}\}_{\boldsymbol{m}\in\{1,2\}^d}$ *defined in Definition 6 satisfies:*

*(i):* $\sum_{\boldsymbol{m}\in\{1,2\}^d} g_{\boldsymbol{m}}(x) = 1$ *for every* $x \in [0,1]^d$.

*(ii):* $\operatorname{supp} g_{\boldsymbol{m}} \cap [0,1]^d \subset \Omega_{\boldsymbol{m}}$, *where* $\Omega_{\boldsymbol{m}}$ *is defined in Definition 5.*

*(ii): For any* $\boldsymbol{m} = (m_1, m_2, \ldots, m_d) \in \{1,2\}^d$ *and* $\boldsymbol{x} = (x_1, x_2, \ldots, x_d) \in [0,1]^d \backslash \Omega_{\boldsymbol{m}}$, *there exists* $j$ *such as* $g_{m_j}(x_j) = 0$ *and* $\frac{\mathrm{d}g_{m_j}(x_j)}{\mathrm{d}x_j} = 0$.

*Proof.* (i) can be proved via induction as Lemma 5, and we leave it to readers.

As for (ii) and (iii), without loss of generality, we show the proof for $\boldsymbol{m}_* := (1,1,\ldots,1)$. For any $\boldsymbol{x} \in [0,1]^d\backslash\Omega_{\boldsymbol{m}_*}$, there is $x_j \in [0,1]\backslash\Omega_1$. Then $g_1(x_j) = 0$ and $g_{\boldsymbol{m}_*}(\boldsymbol{x}) = \prod_{j=1}^d g_1(x_j) = 0$, therefore $\operatorname{supp} g_{\boldsymbol{m}_*} \cap [0,1]^d \subset \Omega_{\boldsymbol{m}_*}$. Furthermore, $\frac{\mathrm{d}g_{m_j}(x_j)}{\mathrm{d}x_j} = 0$ for $x_j \in [0,1] \in \Omega_1$ due to the definition of $g_1$ (Definition 6), then we finish this proof. $\square$

The following lemma demonstrates that $\phi_{\boldsymbol{m}}$, as defined in Proposition 1, can restrict the Sobolev norm of the entire space to $\Omega_{\boldsymbol{m}}$.

**Lemma 7.** *For any* $\chi(\boldsymbol{x}) \in W^{1,\infty}((0,1)^d)$, *denote*

$$M = \max\{\|\chi\|_{W^{1,\infty}((0,1)^d)}, \|\phi_{\boldsymbol{m}}\|_{W^{1,\infty}((0,1)^d)}\},$$

*then we have*

$$\|\phi_{\boldsymbol{m}}(\boldsymbol{x}) \cdot \chi(\boldsymbol{x})\|_{W^{1,\infty}((0,1)^d)} = \|\phi_{\boldsymbol{m}}(\boldsymbol{x}) \cdot \chi(\boldsymbol{x})\|_{W^{1,\infty}(\Omega_{\boldsymbol{m}})}$$
$$\|\phi_{\boldsymbol{m}}(\boldsymbol{x}) \cdot \chi(\boldsymbol{x}) - \phi_M(\phi_{\boldsymbol{m}}(\boldsymbol{x}), \chi(\boldsymbol{x}))\|_{W^{1,\infty}((0,1)^d)} = \|\phi_{\boldsymbol{m}}(\boldsymbol{x}) \cdot \chi(\boldsymbol{x}) - \phi_M(\phi_{\boldsymbol{m}}(\boldsymbol{x}), \chi(\boldsymbol{x}))\|_{W^{1,\infty}(\Omega_{\boldsymbol{m}})}$$
$$(41)$$

*for any* $\boldsymbol{m} \in \{1,2\}^d$, *where* $\phi_{\boldsymbol{m}}(\boldsymbol{x})$ *and* $\Omega_{\boldsymbol{m}}$ *is defined in Proposition 1 and Definition. 5, and* $\phi_M$ *is from Proposition 4 (choosing* $a = M$ *in the proposition).*

*Proof.* For the first equality, we only need to show that

$$\|\phi_{\boldsymbol{m}}(\boldsymbol{x}) \cdot \chi(\boldsymbol{x})\|_{W^{1,\infty}((0,1)^d\backslash\Omega_{\boldsymbol{m}})} = 0. \tag{42}$$

According to the Proposition 1, we have $\phi_{\boldsymbol{m}}(\boldsymbol{x}) = \phi_{\mathrm{prod}}(g_{m_1}, g_{m_2}, \ldots, g_{m_d})$, and for any $\boldsymbol{x} = (x_1, x_2, \ldots, x_d) \in (0,1)^d\backslash\Omega_{\boldsymbol{m}}$, there is $m_j$ such as $g_{m_j}(x_j) = 0$ and $\frac{\mathrm{d}g_{m_j}(x_j)}{\mathrm{d}x_j} = 0$ due to Lemma 6. Based on Eq. (31) in Proposition 5, we have

$$\phi_{\boldsymbol{m}}(\boldsymbol{x}) = \frac{\partial\phi_{\boldsymbol{m}}(\boldsymbol{x})}{\partial x_s} = 0, \ x \in (0,1)^d\backslash\Omega_{\boldsymbol{m}}, s \neq j.$$

Furthermore,

$$\frac{\partial\phi_{\boldsymbol{m}}(\boldsymbol{x})}{\partial x_j} = \frac{\partial\phi_{\mathrm{prod}}(g_{m_1}, g_{m_2}, \ldots, g_{m_d})}{\partial g_{m_j}} \frac{\mathrm{d}g_{m_j}(x_j)}{\mathrm{d}x_j} = 0. \tag{43}$$

Hence we have

$$|\phi_{\boldsymbol{m}}(\boldsymbol{x}) \cdot \chi(\boldsymbol{x})| + \sum_{q=1}^d \left|\frac{\partial[\phi_{\boldsymbol{m}}(\boldsymbol{x}) \cdot \chi(\boldsymbol{x})]}{\partial x_q}\right| = 0 \tag{44}$$

for all $\boldsymbol{x} \in (0,1)^d\backslash\Omega_{\boldsymbol{m}}$.

Similarly, for the second equality in this lemma, we have

$$|\phi_M(\phi_{\boldsymbol{m}}(\boldsymbol{x}), \chi(\boldsymbol{x}))| + \sum_{q=1}^d \left|\frac{\partial[\phi_M(\phi_{\boldsymbol{m}}(\boldsymbol{x}), \chi(\boldsymbol{x}))]}{\partial x_q}\right|$$

$$= |\phi_M(0, \chi(\boldsymbol{x}))| + \sum_{q=1}^d \left[\left|\frac{\partial[\phi_M(0, \chi(\boldsymbol{x}))]}{\partial\chi(\boldsymbol{x})} \cdot \frac{\partial\chi(\boldsymbol{x})}{\partial x_q}\right| + \left|\frac{\partial[\phi_M(\phi_{\boldsymbol{m}}(\boldsymbol{x}), \chi(\boldsymbol{x}))]}{\partial\phi_{\boldsymbol{m}}(\boldsymbol{x})} \cdot \frac{\partial\phi_{\boldsymbol{m}}(\boldsymbol{x})}{\partial x_q}\right|\right]$$

$$= 0, \tag{45}$$

for all $\boldsymbol{x} \in (0,1)^d \backslash \Omega_{\boldsymbol{m}}$ based on

$$\phi_M(0,y) = \frac{\partial \phi_M(0,y)}{\partial y} = 0, \ y \in (-M, M),$$

and $\frac{\partial \phi_{\boldsymbol{m}}(\boldsymbol{x})}{\partial x_q} = 0$. Hence we finish our proof. $\qquad \square$

### 7.2.2 An approximation of functions in Sobolev spaces based on the Bramble–Hilbert Lemma [7, Lemma 4.3.8]

In this subsection, we establish $\{f_{K,\boldsymbol{m}}\}_{\boldsymbol{m} \in \{1,2\}^d}$ as mentioned in Subsection 3.1, which is presented in Theorem 6. To prove this result, we build upon the work of [18], which leverages the average Taylor polynomials and the Bramble-Hilbert Lemma to approximate functions in Sobolev spaces.

Before we show Theorem 6, we define subsets of $\Omega_{\boldsymbol{m}}$ for simplicity notations.

For any $\boldsymbol{m} \in \{1,2\}^d$, we define

$$\Omega_{\boldsymbol{m},\boldsymbol{i}} := [0,1]^d \cap \prod_{j=1}^d \left[ \frac{2i_j - 1_{m_j \leq 2}}{2K}, \frac{3 + 4i_j - 2 \cdot 1_{m_j \leq 2}}{4K} \right] \tag{46}$$

$\boldsymbol{i} = (i_1, i_2, \ldots, i_d) \in \{0, 1 \ldots, K\}^d$, and it is easy to check $\bigcup_{\boldsymbol{i} \in \{0,1\ldots,K\}^d} \Omega_{\boldsymbol{m},\boldsymbol{i}} = \Omega_{\boldsymbol{m}}$.

**Theorem 6.** *Let $K \in \mathbb{N}_+$ and $n \geq 2$. Then for any $f \in W^{n,\infty}((0,1)^d)$ with $\|f\|_{W^{n,\infty}((0,1)^d)} \leq 1$ and $\boldsymbol{m} \in \{1,2\}^d$, there exist piece-wise polynomials function $f_{K,\boldsymbol{m}} = \sum_{|\boldsymbol{\alpha}| \leq n-1} g_{f,\boldsymbol{\alpha},\boldsymbol{m}}(\boldsymbol{x}) \boldsymbol{x}^{\boldsymbol{\alpha}}$ on $\Omega_{\boldsymbol{m}}$ (Definition 5) with the following properties:*

$$\|f - f_{K,\boldsymbol{m}}\|_{W^{1,\infty}(\Omega_{\boldsymbol{m}})} \leq C_1(n,d) K^{-(n-1)},$$
$$\|f - f_{K,\boldsymbol{m}}\|_{L^\infty(\Omega_{\boldsymbol{m}})} \leq C_1(n,d) K^{-n}. \tag{47}$$

*Furthermore, $g_{f,\boldsymbol{\alpha},\boldsymbol{m}}(\boldsymbol{x}) : \Omega_{\boldsymbol{m}} \to \mathbb{R}$ is a constant function with on each $\Omega_{\boldsymbol{m},\boldsymbol{i}}$ for $\boldsymbol{i} \in \{0, 1 \ldots, K\}^d$. And*

$$|g_{f,\boldsymbol{\alpha},\boldsymbol{m}}(\boldsymbol{x})| \leq C_2(n,d) \tag{48}$$

*for all $\boldsymbol{x} \in \Omega_{\boldsymbol{m}}$, where $C_1$ and $C_2$ are constants independent with $K$.*

This proof is similar to that of [18, Lemma C.4.], but we provide detailed proof as follows for readability. Before the proof, we must introduce the partition of unity, average Taylor polynomials, and a lemma.

**Definition 7** (The partition of unity). *Let $d, K \in \mathbb{N}_+$, then*

$$\Psi = \left\{ h_{\boldsymbol{i}} : \boldsymbol{i} \in \{0, 1, \ldots, K\}^d \right\}$$

*with $h_{\boldsymbol{i}} : \mathbb{R}^d \to \mathbb{R}$ for all $\boldsymbol{i} \in \{0, 1, \ldots, K\}^d$ is called the partition of unity $[0,1]^d$ if it satisfies*

*(i): $0 \leq h_{\boldsymbol{i}}(\boldsymbol{x}) \leq 1$ for every $h_{\boldsymbol{i}} \in \Psi$.*

*(ii): $\sum_{h_{\boldsymbol{i}} \in \Psi} h_{\boldsymbol{i}} = 1$ for every $x \in [0,1]^d$.*

**Definition 8.** *Let $n \geq 1$ and $f \in W^{n,\infty}((0,1)^d)$, $\boldsymbol{x}_0 \in ((0,1)^d)$ and $r > 0$ such that for the ball $B(\boldsymbol{x}_0) := B(\boldsymbol{x}_0)_{r,|\cdot|}$ which is a compact subset of $((0,1)^d)$. The corresponding Taylor polynomial of order $n$ of $f$ averaged over $B$ is defined for*

$$Q^n f(x) := \int_B T_{\boldsymbol{y}}^n f(\boldsymbol{x}) b_r(\boldsymbol{y}) \, d\boldsymbol{y} \tag{49}$$

*where*

$$T_{\boldsymbol{y}}^n f(\boldsymbol{x}) := \sum_{|\boldsymbol{\alpha}| \leq n-1} \frac{1}{\boldsymbol{\alpha}!} D^{\boldsymbol{\alpha}} f(\boldsymbol{y})(\boldsymbol{x} - \boldsymbol{y})^{\boldsymbol{\alpha}},$$

$$b_r(\boldsymbol{x}) := \begin{cases} \frac{1}{c_r} e^{-\left(1 - (|\boldsymbol{x} - \boldsymbol{x}_0|/r)^2\right)^{-1}}, & |\boldsymbol{x} - \boldsymbol{x}_0| < r, \\ 0, & |\boldsymbol{x} - \boldsymbol{x}_0| \leq r, \end{cases}$$

$$c_r = \int_{\mathbb{R}^d} e^{-\left(1 - (|\boldsymbol{x} - \boldsymbol{x}_0|/r)^2\right)^{-1}} \, dx. \tag{50}$$

**Lemma 8.** *Let $n \geq 1$ and $f \in W^{n,\infty}((0,1)^d)$, $\boldsymbol{x}_0 \in \Omega$ and $r > 0$ such that for the ball $B(\boldsymbol{x}_0) := B_{r,|\cdot|}(\boldsymbol{x}_0)$ which is a compact subset of $((0,1)^d)$. The corresponding Taylor polynomial of order $n$ of $f$ averaged over $B$ can be read as*

$$Q^n f(\boldsymbol{x}) = \sum_{|\boldsymbol{\alpha}| \leq n-1} c_{f,\boldsymbol{\alpha}} \boldsymbol{x}^{\boldsymbol{\alpha}}.$$

*Furthermore,*

$$|c_{f,\boldsymbol{\alpha}}| \leq C_2(n,d) \|f\|_{W^{n-1,\infty}(B)}. \tag{51}$$

*where $C_2(n,d) = \sum_{|\boldsymbol{\alpha}+\boldsymbol{\beta}| \leq n-1} \frac{1}{\boldsymbol{\alpha}! \boldsymbol{\beta}!}$.*

*Proof.* Based on [18, Lemma B.9.], $Q^n f(x)$ can be read as

$$Q^n f(\boldsymbol{x}) = \sum_{|\boldsymbol{\alpha}| \leq n-1} c_{f,\boldsymbol{\alpha}} \boldsymbol{x}^{\boldsymbol{\alpha}} \tag{52}$$

where

$$c_{f,\boldsymbol{\alpha}} = \sum_{|\boldsymbol{\alpha}+\boldsymbol{\beta}| \leq n-1} \frac{1}{(\boldsymbol{\beta}+\boldsymbol{\alpha})!} a_{\boldsymbol{\beta}+\boldsymbol{\alpha}} \int_B D^{\boldsymbol{\alpha}+\boldsymbol{\beta}} f(\boldsymbol{x}) \boldsymbol{y}^{\boldsymbol{\beta}} b_r(\boldsymbol{y}) \, \mathrm{d}\boldsymbol{y} \tag{53}$$

for $a_{\boldsymbol{\beta}+\boldsymbol{\alpha}} \leq \frac{(\boldsymbol{\alpha}+\boldsymbol{\beta})!}{\boldsymbol{\alpha}! \boldsymbol{\beta}!}$. Note that

$$\left| \int_B D^{\boldsymbol{\alpha}+\boldsymbol{\beta}} f(\boldsymbol{x}) \boldsymbol{y}^{\boldsymbol{\beta}} b_r(\boldsymbol{y}) \, \mathrm{d}\boldsymbol{y} \right| \leq \|f\|_{W^{n-1,\infty}(B)} \|b_r(x)\|_{L^1(B)} = \|f\|_{W^{n-1,\infty}(B)}. \tag{54}$$

Then

$$|c_{f,\boldsymbol{\alpha}}| \leq C_2(n,d) \|f\|_{W^{n-1,\infty}(B_{\boldsymbol{m},N})}. \tag{55}$$

where $C_2(n,d) = \sum_{|\boldsymbol{\alpha}+\boldsymbol{\beta}| \leq n-1} \frac{1}{\boldsymbol{\alpha}! \boldsymbol{\beta}!}$. $\qquad\square$

The proof of Theorem 6 is based on average Taylor polynomials and the Bramble–Hilbert Lemma [7, Lemma 4.3.8].

**Definition 9.** *Let $\Omega$, $B \in \mathbb{R}^d$. Then $\Omega$ is called stared-shaped with respect to $B$ if*

$$\overline{conv}\left(\{\boldsymbol{x}\} \cup B \subset \Omega\right), \textit{ for all } \boldsymbol{x} \in \Omega.$$

**Definition 10.** *Let $\Omega \in \mathbb{R}^d$ be bounded, and define*

$$\mathcal{R} := \left\{ r > 0 : \begin{array}{l} \text{there exists } \boldsymbol{x}_0 \in \Omega \text{ such that } \Omega \text{ is} \\ \text{star-shaped with respect to } B_{r,|\cdot|}(\boldsymbol{x}_0) \end{array} \right\}.$$

*Then we define*

$$r_{\max}^{\star} := \sup \mathcal{R} \quad \textit{and call} \quad \gamma := \frac{\mathrm{diam}(\Omega)}{r_{\max}^{\star}}$$

*the chunkiness parameter of $\Omega$ if $\mathcal{R} \neq \emptyset$.*

**Lemma 9** (Bramble–Hilbert Lemma [7, Lemma 4.3.8]). *Let $\Omega \in \mathbb{R}^d$ be open and bounded, $\boldsymbol{x}_0 \in \Omega$ and $r > 0$ such that $\Omega$ is the stared-shaped with respect to $B := B_{r,|\cdot|}(\boldsymbol{x}_0)$, and $r \geq \frac{1}{2} r_{\max}^{\star}$. Moreover, let $n \in \mathbb{N}_+$, $1 \leq p \leq \infty$ and denote by $\gamma$ by the chunkiness parameter of $\Omega$. Then there is a constant $C(n,d,\gamma) > 0$ such that for all $f \in W^{n,p}(\Omega)$*

$$|f - Q^n f|_{W^{k,p}(\Omega)} \leq C(n,d,\gamma) h^{n-k} |f|_{W^{n,p}(\Omega)} \quad \textit{for } k = 0,1,\ldots,n$$

*where $Q^n f$ denotes the Taylor polynomial of order $n$ of $f$ averaged over $B$ and $h = \mathrm{diam}(\Omega)$.*

*Proof of Theorem 6.* Without loss of generalization, we prove the case for $\boldsymbol{m} = (1,1,\ldots,1) =: \boldsymbol{m}_*$.

Denote $E : W^{n,\infty}((0,1)^d) \to W^{n,\infty}(\mathbb{R}^d)$ be an extension operator [43] and set $\tilde{f} := Ef$ and $C_E$ is the norm of the extension operator.

Define $p_{f,\boldsymbol{i}}$ as the average Taylor polynomial Definition 8 in $B_{\boldsymbol{i},K} := B_{\frac{1}{4K},|\cdot|}\left(\frac{8\boldsymbol{i}+3}{8K}\right)$ i.e.

$$p_{f,\boldsymbol{i}} := \int_{B_{\boldsymbol{i},K}} T_{\boldsymbol{y}}^n \tilde{f}(\boldsymbol{x}) b_{\frac{1}{4K}}(\boldsymbol{y})\, \mathrm{d}\boldsymbol{y}. \tag{56}$$

Based on Lemma 8, $p_{f,\boldsymbol{i}}$ can be read as

$$p_{f,\boldsymbol{i}} = \sum_{|\boldsymbol{\alpha}| \leq n-1} c_{f,\boldsymbol{i},\boldsymbol{\alpha}} \boldsymbol{x}^{\boldsymbol{\alpha}} \tag{57}$$

where

$$|c_{f,\boldsymbol{i},\boldsymbol{\alpha}}| \leq C_2(n,d). \tag{58}$$

The reason to define average Taylor polynomial on $B_{\boldsymbol{i},K}$ is to use the Bramble–Hilbert Lemma 9 on

$$\Omega_{\boldsymbol{m}_*,\boldsymbol{i}} = B_{\frac{3}{8K},\|\cdot\|_{\ell_\infty}}\left(\frac{8\boldsymbol{i}+3}{8K}\right) = \prod_{j=1}^d \left[\frac{i_j}{K}, \frac{3+4i_j}{4K}\right].$$

Note that

$$\frac{1}{4K} \geq \frac{1}{2}\cdot\frac{3}{8K} = \frac{1}{2}r_{\max}^\star(\Omega_{\boldsymbol{m}_*,\boldsymbol{i}}), \ \gamma(\Omega_{\boldsymbol{m}_*,\boldsymbol{i}}) = \frac{\mathrm{diam}(\Omega_{\boldsymbol{m}_*,\boldsymbol{i}})}{r_{\max}^\star(\Omega_{\boldsymbol{m}_*,\boldsymbol{i}})} = 2\sqrt{d}.$$

Therefore we can apply the Bramble–Hilbert Lemma 9 and have

$$\begin{aligned}
\|\tilde{f} - p_{f,\boldsymbol{i}}\|_{L^\infty(\Omega_{\boldsymbol{m}_*,\boldsymbol{i}})} &\leq C_{BH}(n,d)K^{-n}\\
|\tilde{f} - p_{f,\boldsymbol{i}}|_{W^{1,\infty}(\Omega_{\boldsymbol{m}_*,\boldsymbol{i}})} &\leq C_{BH}(n,d)K^{-(n-1)}
\end{aligned} \tag{59}$$

where $C_{BH}(n,d) = |\{|\boldsymbol{\alpha}| = n\}| \frac{1}{d\int_0^1 x^{d-1}e^{-(1-x^2)^{-1}}\,\mathrm{d}x}\left(2+4\sqrt{d}\right)^d C_E$ by following the proof of Lemma [7, Lemma 4.3.8]. Therefore,

$$\|\tilde{f} - p_{f,\boldsymbol{i}}\|_{W^{1,\infty}(\Omega_{\boldsymbol{m}_*,\boldsymbol{i}})} \leq C_1(n,d)K^{-(n-1)}$$

where $C_1(n,d) = 2C_{BH}(n,d)$.

Now we construct a partition of unity that we use in this theorem. First of all, given any integer $K$, define $\{h_i\}_{i=0}^K$ from $\mathbb{R} \to \mathbb{R}$:

$$h_i(x) := h\left(4K\left(x - \frac{8i+3}{8K}\right)\right), \ h(x) := \begin{cases} 1, & |x| < \frac{3}{2} \\ 0, & |x| > 2 \\ 4 - 2|x|, & \frac{3}{2} \leq |x| \leq 2. \end{cases} \tag{60}$$

It is easy to check that $\{h_i\}_{i=0}^K$ is a partition of unity of $[0,1]$ and $h_i(x) = 1$ for $x \in \left[\frac{i}{K}, \frac{3+4i}{4K}\right]$. Hence we can define $h_{\boldsymbol{i}}(\boldsymbol{x})$ for $\boldsymbol{i} = (i_1, i_2, \ldots, i_d) \in \{0, 1, \ldots, K\}^d$ and $\boldsymbol{x} = (x_1, x_2, \ldots, x_d) \in \mathbb{R}^d$:

$$h_{\boldsymbol{i}}(\boldsymbol{x}) = \prod_{j=1}^d h_{i_j}(x_j), \tag{61}$$

and $\left\{h_{\boldsymbol{i}} : \boldsymbol{i} \in \{0, 1, \ldots, K\}^d\right\}$ is a partition of unity of $[0,1]^d$ and $h_{\boldsymbol{i}}(\boldsymbol{x}) = 1$ for $\boldsymbol{x} \in \prod_{j=1}^d \left[\frac{i_j}{K}, \frac{3+4i_j}{4K}\right] = \Omega_{\boldsymbol{m}_*,\boldsymbol{i}}$ and $\boldsymbol{i} = (i_1, i_2, \ldots, i_d) \in \{0, 1, \ldots, K\}^d$.

Furthermore,

$$\|h_{\boldsymbol{i}}(\tilde{f} - p_{f,\boldsymbol{i}})\|_{L^\infty(\Omega_{\boldsymbol{m}_*,\boldsymbol{i}})} \leq \|\tilde{f} - p_{f,\boldsymbol{i}}\|_{L^\infty(\Omega_{\boldsymbol{m}_*,\boldsymbol{i}})} \leq C_{BH}(n,d)K^{-n} \tag{62}$$

and

$$|h_{\boldsymbol{i}}(\tilde{f} - p_{f,\boldsymbol{i}})|_{W^{1,\infty}(\Omega_{\boldsymbol{m}_*,\boldsymbol{i}})} \leq |\tilde{f} - p_{f,\boldsymbol{i}}|_{W^{1,\infty}(\Omega_{\boldsymbol{m}_*,\boldsymbol{i}})} \leq C_{BH}(n,d)K^{-(n-1)} \tag{63}$$

which is due to $h_{\boldsymbol{i}} = 1$ on $\Omega_{\boldsymbol{m}_*,\boldsymbol{i}}$.

Then

$$\|h_{\boldsymbol{i}}(\tilde{f} - p_{f,\boldsymbol{i}})\|_{W^{1,\infty}(\Omega_{\boldsymbol{m}_*,\boldsymbol{i}})} \leq C_1(n,d)K^{-(n-1)}.$$

Finally,

$$\left\| f - \sum_{\boldsymbol{i} \in \{0,1,\dots,K\}^d} h_{\boldsymbol{i}} p_{f,\boldsymbol{i}} \right\|_{W^{1,\infty}(\Omega_{m_*})} \leq \max_{\boldsymbol{i} \in \{0,1,\dots,K\}^d} \| h_{\boldsymbol{i}} (\tilde{f} - p_{f,\boldsymbol{i}}) \|_{W^{1,\infty}(\Omega_{m_*,\boldsymbol{i}})}$$
$$\leq C_1(n,d) K^{-(n-1)}, \tag{64}$$

which is due to $\cup_{\boldsymbol{i} \in \{0,1,\dots,K\}^d} \Omega_{\boldsymbol{m}_*,\boldsymbol{i}} = \Omega_{m_*}$ and supp $h_{\boldsymbol{i}} \cap \Omega_{m_*} = \Omega_{\boldsymbol{m}_*,\boldsymbol{i}}$.

Similarly,

$$\left\| f - \sum_{\boldsymbol{i} \in \{0,1,\dots,K\}^d} h_{\boldsymbol{i}} p_{f,\boldsymbol{i}} \right\|_{L^{\infty}(\Omega_{1,d})} \leq C_1(n,d) K^{-n}. \tag{65}$$

Last of all,

$$\begin{aligned} f_{k,\boldsymbol{m}_*}(\boldsymbol{x}) &:= \sum_{\boldsymbol{i} \in \{0,1,\dots,K\}^d} h_{\boldsymbol{i}} p_{f,\boldsymbol{i}} = \sum_{\boldsymbol{i} \in \{0,1,\dots,K\}^d} \sum_{|\boldsymbol{\alpha}| \leq n-1} h_{\boldsymbol{i}} c_{f,\boldsymbol{i},\boldsymbol{\alpha}} \boldsymbol{x}^{\boldsymbol{\alpha}} \\ &= \sum_{|\boldsymbol{\alpha}| \leq n-1} \sum_{\boldsymbol{i} \in \{0,1,\dots,K\}^d} h_{\boldsymbol{i}} c_{f,\boldsymbol{i},\boldsymbol{\alpha}} \boldsymbol{x}^{\boldsymbol{\alpha}} \\ &=: \sum_{|\boldsymbol{\alpha}| \leq n-1} g_{f,\boldsymbol{\alpha},\boldsymbol{m}_*}(\boldsymbol{x}) \boldsymbol{x}^{\boldsymbol{\alpha}} \end{aligned} \tag{66}$$

with $|g_{f,\boldsymbol{\alpha},\boldsymbol{m}_*}(\boldsymbol{x})| \leq C_2(n,d)$ for $x \in \Omega_{\boldsymbol{m}_*}$. Note that $g_{f,\boldsymbol{\alpha},\boldsymbol{m}_*}(\boldsymbol{x})$ is a step function from $\Omega_{\boldsymbol{m}_*} \to \mathbb{R}$:

$$g_{f,\boldsymbol{\alpha},\boldsymbol{m}_*}(\boldsymbol{x}) = c_{f,\boldsymbol{i},\boldsymbol{\alpha}} \tag{67}$$

for $\boldsymbol{x} \in \prod_{j=1}^d \left[ \frac{i_j}{K}, \frac{3+4i_j}{4K} \right]$ and $\boldsymbol{i} = (i_1, i_2, \dots, i_d)$ since $h_{\boldsymbol{i}}(\boldsymbol{x}) = 0$ for $\boldsymbol{x} \in \Omega_{\boldsymbol{m}_*} \setminus \prod_{j=1}^d \left[ \frac{i_j}{K}, \frac{3+4i_j}{4K} \right]$ and $h_{\boldsymbol{i}}(\boldsymbol{x}) = 1$ for $\boldsymbol{x} \in \prod_{j=1}^d \left[ \frac{i_j}{K}, \frac{3+4i_j}{4K} \right]$. $\qquad \square$

### 7.2.3 Approximation of functions in $W^{n,\infty}$ with $W^{1,\infty}$ norm by ReLU neural networks in the whole space except a small set

**Theorem 7.** *For any $f \in W^{n,\infty}((0,1)^d)$ with $\|f\|_{W^{n,\infty}((0,1)^d)} \leq 1$, any $N, L \in \mathbb{N}_+$, and $\boldsymbol{m} = (m_1, m_2, \dots, m_d) \in \{1,2\}^d$, there is a neural network $\psi_{\boldsymbol{m}}$ with the width $25n^{d+1}(N+1)\log_2(8N)$ and depth $27n^2(L+2)\log_2(4L)$ such that*

$$\|f(\boldsymbol{x}) - \psi_{\boldsymbol{m}}(\boldsymbol{x})\|_{W^{1,\infty}(\Omega_{\boldsymbol{m}})} \leq C_6(n,d) N^{-2(n-1)/d} L^{-2(n-1)/d}$$
$$\|f(\boldsymbol{x}) - \psi_{\boldsymbol{m}}(\boldsymbol{x})\|_{L^{\infty}(\Omega_{\boldsymbol{m}})} \leq C_6(n,d) N^{-2n/d} L^{-2n/d}, \tag{68}$$

*where $C_6$ is the constant independent with $N, L$.*

*Proof.* Without loss of the generalization, we consider the case for $\boldsymbol{m}_* = (1,1,\dots,1)$. Due to Theorem 6 and setting $K = \lfloor N^{1/d} \rfloor^2 \lfloor L^{2/d} \rfloor$, we have

$$\|f - f_{K,\boldsymbol{m}_*}\|_{W^{1,\infty}(\Omega_{\boldsymbol{m}_*})} \leq C_1(n,d) K^{-(n-1)} \leq C_1(n,d) N^{-2(n-1)/d} L^{-2(n-1)/d}$$
$$\|f - f_{K,\boldsymbol{m}_*}\|_{L^{\infty}(\Omega_{\boldsymbol{m}_*})} \leq C_1(n,d) K^{-n} \leq C_1(n,d) N^{-2n/d} L^{-2n/d}, \tag{69}$$

where $f_{K,\boldsymbol{m}_*} = \sum_{|\boldsymbol{\alpha}| \leq n-1} g_{f,\boldsymbol{\alpha},\boldsymbol{m}_*}(\boldsymbol{x}) \boldsymbol{x}^{\boldsymbol{\alpha}}$ for $x \in \Omega_{\boldsymbol{m}_*}$. Note that $g_{f,\boldsymbol{\alpha},\boldsymbol{m}_*}(\boldsymbol{x})$ is a constant function for $\boldsymbol{x} \in \prod_{j=1}^d \left[ \frac{i_j}{K}, \frac{3+4i_j}{4K} \right]$ and $\boldsymbol{i} = (i_1, i_2, \dots, i_d) \in \{0,1,\dots,K-1\}^d$. The remaining part is to approximate $f_{K,\boldsymbol{m}_*}$ by neural networks.

The way to approximate $g_{f,\boldsymbol{\alpha},\boldsymbol{m}_*}(\boldsymbol{x})$ is similar with [23, Theorem 3.1]. First of all, due to Proposition 2, there is a neural network $\phi_1(x)$ with the width $4N+5$ and depth $4L+4$ such that

$$\phi(x) = k, x \in \left[ \frac{k}{K}, \frac{k+1}{K} - \frac{1}{4K} \right], \ k = 0,1,\dots,K-1. \tag{70}$$

Note that we choose $\delta = \frac{1}{4K} \le \frac{1}{3K}$ in Proposition 2. Then define

$$\phi_2(\boldsymbol{x}) = \left[\frac{\phi_1(x_1)}{K}, \frac{\phi_1(x_2)}{K}, \ldots, \frac{\phi_1(x_d)}{K}\right]^{\mathsf{T}}.$$

For each $p = 0, 1, \ldots, K^d - 1$, there is a bijection

$$\boldsymbol{\eta}(p) = [\eta_1, \eta_2, \ldots, \eta_d] \in \{0, 1, \ldots, K-1\}^d$$

such that $\sum_{j=1}^{d} \eta_j K^{j-1} = p$. Then define

$$\xi_{\boldsymbol{\alpha}, p} = \frac{g_{f,\boldsymbol{\alpha},\boldsymbol{m}_*}\left(\frac{\boldsymbol{\eta}(p)}{K}\right) + C_2(n, d)}{2C_2(n, d)} \in [0, 1],$$

where $C_2(n, d)$ is the bounded of $g_{f,\boldsymbol{\alpha},\boldsymbol{m}_*}$ defined in Theorem 6. Therefore, based on Proposition 3, there is a neural network $\tilde{\phi}_{\boldsymbol{\alpha}}(x)$ with the width $16n(N+1)\log_2(8N)$ and depth $(5L+2)\log_2(4L)$ such that $|\tilde{\phi}_{\boldsymbol{\alpha}}(p) - \xi_{\boldsymbol{\alpha}, p}| \le N^{-2n}L^{-2n}$ for $p = 0, 1, \ldots K^d - 1$. Denote

$$\phi_{\boldsymbol{\alpha}}(\boldsymbol{x}) = 2C_2(n, d)\tilde{\phi}_{\boldsymbol{\alpha}}\left(\sum_{j=1}^{d} x_j K^j\right) - C_2(n, d)$$

and obtain that

$$\left|\phi_{\boldsymbol{\alpha}}\left(\frac{\boldsymbol{\eta}(p)}{K}\right) - g_{f,\boldsymbol{\alpha},\boldsymbol{m}_*}\left(\frac{\boldsymbol{\eta}(p)}{K}\right)\right| = 2C_2(n, d)|\tilde{\phi}_{\boldsymbol{\alpha}}(p) - \xi_{\boldsymbol{\alpha}, p}| \le 2C_2(n, d)N^{-2n}L^{-2n}.$$

Then we obtain that

$$\begin{aligned}\|\phi_{\boldsymbol{\alpha}}(\boldsymbol{\phi}_2(\boldsymbol{x})) - g_{f,\boldsymbol{\alpha},\boldsymbol{m}_*}(\boldsymbol{x})\|_{W^{1,\infty}(\Omega_{\boldsymbol{m}_*})} &= \|\phi_{\boldsymbol{\alpha}}(\boldsymbol{\phi}_2(\boldsymbol{x})) - g_{f,\boldsymbol{\alpha},\boldsymbol{m}_*}(\boldsymbol{x})\|_{L^{\infty}(\Omega_{\boldsymbol{m}_*})} \\ &\le 2C_2(n, d)N^{-2n}L^{-2n}\end{aligned} \quad (71)$$

which is due to $\phi_{\boldsymbol{\alpha}}(\boldsymbol{\phi}_2(\boldsymbol{x})) - g_{f,\boldsymbol{\alpha},\boldsymbol{m}_*}(\boldsymbol{x})$ is a step function, and the first order weak derivative is 0 in $\Omega_{\boldsymbol{m}_*}$.

Due to Proposition 6, there is a neural network $\phi_{3,\boldsymbol{\alpha}}$ with the width $9(N+1) + n - 1$ and depth $14n^2 L$ such that $\|\phi_{3,\boldsymbol{\alpha}}\|_{W^{1,\infty}((0,1)^d)} \le 18$ and

$$\|\phi_{3,\boldsymbol{\alpha}}(\boldsymbol{x}) - \boldsymbol{x}^{\boldsymbol{\alpha}}\|_{W^{1,\infty}((0,1)^d)} \le 10n(N+1)^{-7nL}. \quad (72)$$

Due to Proposition 4, there is a neural network $\phi_4$ with the width $15(N+1)$ and depth $4n(L+1)$ such that $\|\phi_4\|_{W^{1,\infty}(-C_3, C_3)^2} \le 12(C_2(n, d))^2$ and

$$\|\phi_4(x, y) - xy\|_{W^{1,\infty}((-C_3, C_3)^2)} \le 6(C_2(n, d))^2(N+1)^{-2n(L+1)}. \quad (73)$$

where $C_3(n, d) = \max\{3C_2(n, d), 18\}$.

Now we define the neural network $\phi_{\boldsymbol{m}_*}(\boldsymbol{x})$ to approximate $f_{K,\boldsymbol{m}_*}(\boldsymbol{x})$ in $\Omega_{\boldsymbol{m}_*}$:

$$\psi_{\boldsymbol{m}_*}(\boldsymbol{x}) = \sum_{|\boldsymbol{\alpha}| \le n-1} \phi_4\left[\phi_{\boldsymbol{\alpha}}(\boldsymbol{\phi}_2(\boldsymbol{x})), \phi_{3,\boldsymbol{\alpha}}(\boldsymbol{x})\right]. \quad (74)$$

The remaining question is to find the error $\mathcal{E}$:

$$\mathcal{E} := \left\| \sum_{|\boldsymbol{\alpha}| \leq n-1} \phi_4 \left[ \phi_{\boldsymbol{\alpha}}(\boldsymbol{\phi}_2(\boldsymbol{x})), \phi_{3,\boldsymbol{\alpha}}(\boldsymbol{x}) \right] - f_{K,\boldsymbol{m}_*}(\boldsymbol{x}) \right\|_{W^{1,\infty}(\Omega_{\boldsymbol{m}_*})}$$

$$\leq \sum_{|\boldsymbol{\alpha}| \leq n-1} \left\| \phi_4 \left[ \phi_{\boldsymbol{\alpha}}(\boldsymbol{\phi}_2(\boldsymbol{x})), \phi_{3,\boldsymbol{\alpha}}(\boldsymbol{x}) \right] - g_{f,\boldsymbol{\alpha},\boldsymbol{m}_*}(\boldsymbol{x}) \boldsymbol{x}^{\boldsymbol{\alpha}} \right\|_{W^{1,\infty}(\Omega_{\boldsymbol{m}_*})}$$

$$\leq \underbrace{\sum_{|\boldsymbol{\alpha}| \leq n-1} \left\| \phi_4 \left[ \phi_{\boldsymbol{\alpha}}(\boldsymbol{\phi}_2(\boldsymbol{x})), \phi_{3,\boldsymbol{\alpha}}(\boldsymbol{x}) \right] - \phi_{\boldsymbol{\alpha}}(\boldsymbol{\phi}_2(\boldsymbol{x})) \phi_{3,\boldsymbol{\alpha}}(\boldsymbol{x}) \right\|_{W^{1,\infty}(\Omega_{\boldsymbol{m}_*})}}_{=:\mathcal{E}_1}$$

$$+ \underbrace{\sum_{|\boldsymbol{\alpha}| \leq n-1} \left\| \phi_{\boldsymbol{\alpha}}(\boldsymbol{\phi}_2(\boldsymbol{x})) \phi_{3,\boldsymbol{\alpha}}(\boldsymbol{x}) - g_{f,\boldsymbol{\alpha},\boldsymbol{m}_*}(\boldsymbol{x}) \phi_{3,\boldsymbol{\alpha}}(\boldsymbol{x}) \right\|_{W^{1,\infty}(\Omega_{\boldsymbol{m}_*})}}_{=:\mathcal{E}_2}$$

$$+ \underbrace{\sum_{|\boldsymbol{\alpha}| \leq n-1} \left\| g_{f,\boldsymbol{\alpha},\boldsymbol{m}_*}(\boldsymbol{x}) \phi_{3,\boldsymbol{\alpha}}(\boldsymbol{x}) - g_{f,\boldsymbol{\alpha},\boldsymbol{m}_*}(\boldsymbol{x}) \boldsymbol{x}^{\boldsymbol{\alpha}} \right\|_{W^{1,\infty}(\Omega_{\boldsymbol{m}_*})}}_{=:\mathcal{E}_3}. \tag{75}$$

As for $\mathcal{E}_1$, due to Lemma 3, we have

$$\mathcal{E}_1 \leq \sum_{|\boldsymbol{\alpha}| \leq n-1} 2\sqrt{d} \max \left\{ \| \phi_4(x,y) - xy \|_{L^\infty((-C_3,C_3)^2)}, \| \phi_4(x,y) - xy \|_{W^{1,\infty}((-C_3,C_3)^2)} \right.$$

$$\left. \cdot \max\{ \| \phi_{\boldsymbol{\alpha}}(\boldsymbol{\phi}_2(\boldsymbol{x})) \|_{W^{1,\infty}(\Omega_{\boldsymbol{m}_*})}, \| \phi_{3,\boldsymbol{\alpha}}(\boldsymbol{x}) \|_{W^{1,\infty}(\Omega_{\boldsymbol{m}_*})} \} \right\}$$

$$\leq \sum_{|\boldsymbol{\alpha}| \leq n-1} 2\sqrt{d} \max \left\{ \| \phi_4(x,y) - xy \|_{L^\infty((-C_3,C_3)^2)}, C_3(n,d) \| \phi_4(x,y) - xy \|_{W^{1,\infty}((-C_3,C_3)^2)} \right\}$$

$$\leq \sum_{|\boldsymbol{\alpha}| \leq n-1} 12\sqrt{d} \left[ C_3(n,d) + 1 \right] (C_2(n,d))^2 (N+1)^{-2n(L+1)}$$

$$\leq C_4(n,d)(N+1)^{-2n(L+1)} \tag{76}$$

where $C_4(n,d) = 12\sqrt{d} n^d \left[ C_3(n,d) + 1 \right] (C_2(n,d))^2$.

As for $\mathcal{E}_2$, due to Lemma 4, we have

$$\mathcal{E}_2 \leq \sum_{|\boldsymbol{\alpha}| \leq n-1} 2 \| \phi_{\boldsymbol{\alpha}}(\boldsymbol{\phi}_2(\boldsymbol{x})) - g_{f,\boldsymbol{\alpha},\boldsymbol{m}_*}(\boldsymbol{x}) \|_{W^{1,\infty}(\Omega_{\boldsymbol{m}_*})} \cdot \| \phi_{3,\boldsymbol{\alpha}}(\boldsymbol{x}) \|_{W^{1,\infty}(\Omega_{\boldsymbol{m}_*})}$$

$$\leq 72 n^d C_2(n,d) N^{-2n} L^{-2n}. \tag{77}$$

The estimation of $\mathcal{E}_3$ is similar with that of $\mathcal{E}_2$ which is

$$\mathcal{E}_3 \leq \sum_{|\boldsymbol{\alpha}| \leq n-1} \| g_{f,\boldsymbol{\alpha},\boldsymbol{m}_*} \|_{W^{1,\infty}(\Omega_{\boldsymbol{m}_*})} \cdot \| \phi_{3,\boldsymbol{\alpha}}(\boldsymbol{x}) - \boldsymbol{x}^{\boldsymbol{\alpha}} \|_{W^{1,\infty}(\Omega_{\boldsymbol{m}_*})}$$

$$\leq 10 n^d C_2(n,d) n (N+1)^{-7nL}. \tag{78}$$

Therefore, using

$$(N+1)^{-7nL} \leq (N+1)^{-2n(L+1)} \leq N^{-2n} L^{-2n}$$

the total error is

$$\mathcal{E} \leq \mathcal{E}_1 + \mathcal{E}_2 + \mathcal{E}_3 \leq C_5(n,d) K^{-2n} L^{-2n}, \tag{79}$$

where $C_5(n,d) = C_4(n,d) + 72 n^d C_2(n,d) + 10 n^d C_2(n,d) n$.

At last, we finish the proof by estimating the network's width and depth, implementing $\psi_{\boldsymbol{m}_*}(\boldsymbol{x})$. From Eq. (74), we know that $\psi_{\boldsymbol{m}_*}(\boldsymbol{x})$ consists of the following subnetworks:

1. $\phi_{3,\boldsymbol{\alpha}}(\boldsymbol{x})$ with the width $9(N+1) + n - 1$ and depth $14 n^2 L$.

2. $\phi_2(\boldsymbol{x})$ with the width $4N + 5$ and depth $4L + 4$.

3. $\phi_{\boldsymbol{\alpha}}$ with the width $16n(N + 1)\log_2(8N)$ and depth $(5L + 2)\log_2(4L)$.

4. $\phi_4(x, y)$ with the width $15(N + 1)$ and depth $4n(L + 1)$.

Therefore $\phi(\boldsymbol{x})$ is a neural network with the width $25n^{d+1}(N + 1)\log_2(8N)$ and depth $27n^2(L + 2)\log_2(4L)$.

Combining Eqs. (69) and (79), we have that there is a neural network $\psi_{\boldsymbol{m}_*}$ with the width $25n^{d+1}(N + 1)\log_2(8N)$ and depth $27n^2(L + 2)\log_2(4L)$ such that

$$\|f(\boldsymbol{x}) - \psi_{\boldsymbol{m}_*}(\boldsymbol{x})\|_{W^{1,\infty}(\Omega_{\boldsymbol{m}_*})} \leq C_6(n, d)N^{-2(n-1)/d}L^{-2(n-1)/d}$$
$$\|f(\boldsymbol{x}) - \psi_{\boldsymbol{m}_*}(\boldsymbol{x})\|_{L^\infty(\Omega_{\boldsymbol{m}_*})} \leq C_6(n, d)N^{-2n/d}L^{-2n/d}, \tag{80}$$

where $C_6 = C_1 + C_5$ is the constant independent with $N, L$.

Similarly, we can construct a neural network $\psi_{\boldsymbol{m}}$ with the width $25n^{d+1}(N + 1)\log_2(8N)$ and depth $27n^2(L + 2)\log_2(4L)$ which can approximate $f$ on $\Omega_{\boldsymbol{m}}$ with same order of Eq. (80). □

### 7.2.4 Proof of Theorem 3

Now we can prove Theorem 3 based on Theorem 7 and Proposition 1.

*Proof of Theorem 3.* Based on Theorem 7, there is a sequence of the neural network $\{\psi_{\boldsymbol{m}}(\boldsymbol{x})\}_{\boldsymbol{m} \in \{1,2\}^d}$ such that

$$\|f(\boldsymbol{x}) - \psi_{\boldsymbol{m}}(\boldsymbol{x})\|_{W^{1,\infty}(\Omega_{\boldsymbol{m}})} \leq C_6(n, d)N^{-2(n-1)/d}L^{-2(n-1)/d}$$
$$\|f(\boldsymbol{x}) - \psi_{\boldsymbol{m}}(\boldsymbol{x})\|_{L^\infty(\Omega_{\boldsymbol{m}})} \leq C_6(n, d)N^{-2n/d}L^{-2n/d}, \tag{81}$$

where $C_6 = C_1 + C_5$ is the constant independent with $N, L$, and each $\psi_{\boldsymbol{m}}$ is a neural network with the width $25n^{d+1}(N + 1)\log_2(8N)$ and depth $27n^2(L + 2)\log_2(4L)$. According to Proposition 1, there is a sequence of the neural network $\{\phi_{\boldsymbol{m}}(\boldsymbol{x})\}_{\boldsymbol{m} \in \{1,2\}^d}$ such that

$$\|\phi_{\boldsymbol{m}}(\boldsymbol{x}) - g_{\boldsymbol{m}}(\boldsymbol{x})\|_{W^{1,\infty}((0,1)^d)} \leq 50d^{\frac{5}{2}}(N + 1)^{-4dnL},$$

where $\{g_{\boldsymbol{m}}\}_{\boldsymbol{m} \in \{1,2\}^d}$ is defined in Definition 6 with $\sum_{\boldsymbol{m} \in \{1,2\}^d} g_{\boldsymbol{m}}(\boldsymbol{x}) = 1$ and $\operatorname{supp} g_{\boldsymbol{m}} \cap [0, 1]^d = \Omega_{\boldsymbol{m}}$. For each $\phi_{\boldsymbol{m}}$, it is a neural network with the width smaller than $(9 + d)(N + 1) + d - 1$ and depth smaller than $15d(d - 1)nL$.

Due to Proposition 4, there is a neural network $\widehat{\phi}$ with the width $15(N + 1)$ and depth $14n^2L$ such that $\|\widehat{\phi}\|_{W^{1,\infty}(-C_7,C_7)^2} \leq 12(C_7(n, d))^2$ and

$$\left\|\widehat{\phi}(x, y) - xy\right\|_{W^{1,\infty}(-C_7,C_7)^2} \leq 6(C_7)^2(N + 1)^{-7n(L+1)}, \tag{82}$$

where $C_7 = C_6 + 50d^{\frac{5}{2}} + 1$.

Now we define

$$\phi(\boldsymbol{x}) = \sum_{\boldsymbol{m} \in \{1,2\}^d} \widehat{\phi}(\phi_{\boldsymbol{m}}(\boldsymbol{x}), \psi_{\boldsymbol{m}}(\boldsymbol{x})). \tag{83}$$

Note that

$$\mathcal{R} := \|f(\boldsymbol{x}) - \phi(\boldsymbol{x})\|_{W^{1,\infty}((0,1)^d)} = \left\|\sum_{\boldsymbol{m} \in \{1,2\}^d} g_{\boldsymbol{m}} \cdot f(\boldsymbol{x}) - \phi(\boldsymbol{x})\right\|_{W^{1,\infty}((0,1)^d)}$$

$$\leq \left\|\sum_{\boldsymbol{m} \in \{1,2\}^d} [g_{\boldsymbol{m}} \cdot f(\boldsymbol{x}) - \phi_{\boldsymbol{m}}(\boldsymbol{x}) \cdot \psi_{\boldsymbol{m}}(\boldsymbol{x})]\right\|_{W^{1,\infty}((0,1)^d)}$$

$$+ \left\|\sum_{\boldsymbol{m} \in \{1,2\}^d} \left[\phi_{\boldsymbol{m}}(\boldsymbol{x}) \cdot \psi_{\boldsymbol{m}}(\boldsymbol{x}) - \widehat{\phi}(\phi_{\boldsymbol{m}}(\boldsymbol{x}), \psi_{\boldsymbol{m}}(\boldsymbol{x}))\right]\right\|_{W^{1,\infty}((0,1)^d)}. \tag{84}$$

As for the first part,

$$
\left\| \sum_{\boldsymbol{m} \in \{1,2\}^d} [g_{\boldsymbol{m}} \cdot f(\boldsymbol{x}) - \phi_{\boldsymbol{m}}(\boldsymbol{x}) \cdot \psi_{\boldsymbol{m}}(\boldsymbol{x})] \right\|_{W^{1,\infty}((0,1)^d)}
$$

$$
\leq \sum_{\boldsymbol{m} \in \{1,2\}^d} \| g_{\boldsymbol{m}} \cdot f(\boldsymbol{x}) - \phi_{\boldsymbol{m}}(\boldsymbol{x}) \cdot \psi_{\boldsymbol{m}}(\boldsymbol{x}) \|_{W^{1,\infty}((0,1)^d)}
$$

$$
\leq \sum_{\boldsymbol{m} \in \{1,2\}^d} \left[ \| (g_{\boldsymbol{m}} - \phi_{\boldsymbol{m}}(\boldsymbol{x})) \cdot f(\boldsymbol{x}) \|_{W^{1,\infty}((0,1)^d)} + \| (f_{\boldsymbol{m}} - \psi_{\boldsymbol{m}}(\boldsymbol{x})) \cdot \phi_{\boldsymbol{m}}(\boldsymbol{x}) \|_{W^{1,\infty}((0,1)^d)} \right]
$$

$$
= \sum_{\boldsymbol{m} \in \{1,2\}^d} \left[ \| (g_{\boldsymbol{m}} - \phi_{\boldsymbol{m}}(\boldsymbol{x})) \cdot f(\boldsymbol{x}) \|_{W^{1,\infty}((0,1)^d)} + \| (f_{\boldsymbol{m}} - \psi_{\boldsymbol{m}}(\boldsymbol{x})) \cdot \phi_{\boldsymbol{m}}(\boldsymbol{x}) \|_{W^{1,\infty}(\Omega_{\boldsymbol{m}})} \right],
$$

$$(85)$$

where the last equality is due to Lemma 7. Based on Lemma 4 and $\|f\|_{W^{1,\infty}((0,1)^d)} \leq 1$, we have

$$
\| (g_{\boldsymbol{m}} - \phi_{\boldsymbol{m}}(\boldsymbol{x})) \cdot f(\boldsymbol{x}) \|_{W^{1,\infty}((0,1)^d)} \leq \| (g_{\boldsymbol{m}} - \phi_{\boldsymbol{m}}(\boldsymbol{x})) \|_{W^{1,\infty}((0,1)^d)} \leq 50 d^{\frac{5}{2}} (N+1)^{-4dnL}.
$$

$$(86)$$

And

$$
\| (f_{\boldsymbol{m}} - \psi_{\boldsymbol{m}}(\boldsymbol{x})) \cdot \phi_{\boldsymbol{m}}(\boldsymbol{x}) \|_{W^{1,\infty}(\Omega_{\boldsymbol{m}})}
$$

$$
\leq \| (f_{\boldsymbol{m}} - \psi_{\boldsymbol{m}}(\boldsymbol{x})) \|_{W^{1,\infty}(\Omega_{\boldsymbol{m}})} \cdot \| \phi_{\boldsymbol{m}} \|_{L^\infty(\Omega_{\boldsymbol{m}})} + \| (f_{\boldsymbol{m}} - \psi_{\boldsymbol{m}}(\boldsymbol{x})) \|_{L^\infty(\Omega_{\boldsymbol{m}})} \cdot \| \phi_{\boldsymbol{m}} \|_{W^{1,\infty}(\Omega_{\boldsymbol{m}})}
$$

$$
\leq C_6(n,d) N^{-2(n-1)/d} L^{-2(n-1)/d} \cdot \left( 1 + 50 d^{\frac{5}{2}} \right) + C_6(n,d) N^{-2n/d} L^{-2n/d} \cdot 54 d^{\frac{5}{2}} \lfloor N^{1/d} \rfloor^2 \lfloor L^{2/d} \rfloor
$$

$$
\leq C_7(n,d) N^{-2(n-1)/d} L^{-2(n-1)/d},
$$

$$(87)$$

where the second inequality is due to

$$
\| \phi_{\boldsymbol{m}} \|_{L^\infty(\Omega_{\boldsymbol{m}})} \leq \| \phi_{\boldsymbol{m}} \|_{L^\infty([0,1]^d)} \leq \| g_{\boldsymbol{m}} \|_{L^\infty([0,1]^d)} + \| \phi_{\boldsymbol{m}} - g_{\boldsymbol{m}} \|_{L^\infty([0,1]^d)} \leq 1 + 50 d^{\frac{5}{2}}
$$

$$
\| \phi_{\boldsymbol{m}} \|_{W^{1,\infty}(\Omega_{\boldsymbol{m}})} \leq \| \phi_{\boldsymbol{m}} \|_{W^{1,\infty}([0,1]^d)} \leq \| g_{\boldsymbol{m}} \|_{W^{1,\infty}([0,1]^d)} + \| \phi_{\boldsymbol{m}} - g_{\boldsymbol{m}} \|_{W^{1,\infty}([0,1]^d)}
$$

$$
\leq 4 \lfloor N^{1/d} \rfloor^2 \lfloor L^{2/d} \rfloor + 50 d^{\frac{5}{2}}.
$$

$$(88)$$

Therefore

$$
\left\| \sum_{\boldsymbol{m} \in \{1,2\}^d} [g_{\boldsymbol{m}} \cdot f(\boldsymbol{x}) - \phi_{\boldsymbol{m}}(\boldsymbol{x}) \cdot \psi_{\boldsymbol{m}}(\boldsymbol{x})] \right\|_{W^{1,\infty}((0,1)^d)} \leq 2^d (C_7(n,d) + 50 d^{\frac{5}{2}}) N^{-2(n-1)/d} L^{-2(n-1)/d}
$$

$$(89)$$

due to $(N+1)^{-4dnL} \leq N^{-2n} L^{-2n}$.

For the second part, due to Lemma 7, we have

$$
\left\| \sum_{\boldsymbol{m} \in \{1,2\}^d} \left[ \phi_{\boldsymbol{m}}(\boldsymbol{x}) \cdot \psi_{\boldsymbol{m}}(\boldsymbol{x}) - \widehat{\phi}(\phi_{\boldsymbol{m}}(\boldsymbol{x}), \psi_{\boldsymbol{m}}(\boldsymbol{x})) \right] \right\|_{W^{1,\infty}((0,1)^d)}
$$

$$
\leq \sum_{\boldsymbol{m} \in \{1,2\}^d} \left\| \phi_{\boldsymbol{m}}(\boldsymbol{x}) \cdot \psi_{\boldsymbol{m}}(\boldsymbol{x}) - \widehat{\phi}(\phi_{\boldsymbol{m}}(\boldsymbol{x}), \psi_{\boldsymbol{m}}(\boldsymbol{x})) \right\|_{W^{1,\infty}((0,1)^d)}
$$

$$
= \sum_{\boldsymbol{m} \in \{1,2\}^d} \left\| \phi_{\boldsymbol{m}}(\boldsymbol{x}) \cdot \psi_{\boldsymbol{m}}(\boldsymbol{x}) - \widehat{\phi}(\phi_{\boldsymbol{m}}(\boldsymbol{x}), \psi_{\boldsymbol{m}}(\boldsymbol{x})) \right\|_{W^{1,\infty}(\Omega_{\boldsymbol{m}})}.
$$

$$(90)$$

Similarly with the estimation of $\mathcal{E}_1$ (76), we have that

$$
\left\| \phi_{\boldsymbol{m}}(\boldsymbol{x}) \cdot \psi_{\boldsymbol{m}}(\boldsymbol{x}) - \widehat{\phi}(\phi_{\boldsymbol{m}}(\boldsymbol{x}), \psi_{\boldsymbol{m}}(\boldsymbol{x})) \right\|_{W^{1,\infty}(\Omega_{\boldsymbol{m}})}
$$

$$
\leq C_8(n,d)(N+1)^{-7n(L+1)} \leq C_8(n,d) N^{-2(n-1)/d} L^{-2(n-1)/d}.
$$

$$(91)$$

Combining (89) and (91), we have that there is a $\sigma_1$-NN $\phi$ with the width $(34 + d)2^d n^{d+1}(N + 1)\log_2(8N)$ and depth $56d^2 n^2(L + 1)\log_2(4L)$ such that

$$\|f(\boldsymbol{x}) - \phi(\boldsymbol{x})\|_{W^{1,\infty}((0,1)^d)} \leq C_9(n, d)N^{-2(n-1)/d}L^{-2(n-1)/d},$$

where $C_9$ is the constant independent with $N, L$.

$\square$

The method proposed in [28, 23, 39, 38, 37] may not be applied to prove Theorems 3. These works approximate the target function $f$ using a deep neural network $\phi$ in the unit cube except for an arbitrarily small region $\Omega_\delta$, as per [36, Lemma 2.2]. Since $\|\phi\|_{L^\infty(\Omega)}$ can be bounded and is independent of the size of $\Omega\delta$, $\|f - \phi\|_{L^p(\Omega)}$ can be well estimated across the entire space for $p \in [1, +\infty)$. For approximations measured in the $L^\infty(\Omega)$ norm, [28] translates the deep neural network $\phi$, while [39] constructs different neural networks in the unit cube away from various negligible regions. Both methods aim to find neural networks $\{\phi_i(\boldsymbol{x})\}i = 1^N$ that approximate the target function $f$ well in different regions. They then observe that the middle value of $\{\phi_i(\boldsymbol{x})\}_{i=1}^N$ is close to $f(\boldsymbol{x})$ for all $\boldsymbol{x}^* \in \Omega$, and construct the middle-value function using a ReLU neural network. However, these methods may not be generalized to prove the theorems presented in this paper.

Neither of the methods previously proposed can be applied to the approximation measured in Sobolev space. In the first method, $\|\phi\|_{W^{1,\infty}(\Omega)}$ depends on the length of $\Omega\delta$, and the derivative is substantial in the negligible region, as shown in [36, Lemma 2.2]. Thus, $\|f - \phi\|_{W^{1,p}(\Omega)}$ will be excessively large. In the second method, median value functions can only identify the median values, not the median values of functions and their derivatives simultaneously. In this paper, we overcome this difficulty using a partition of unity. We construct a partition of unity of $\Omega$ and approximate them using ReLU DNNs denoted as $\{\phi_{\boldsymbol{m}}\}_{\boldsymbol{m}\in\{1,2\}^d}$. For each $\phi\boldsymbol{m}$, its support set is the unit cube away from a small region, and we can construct a deep neural network $\psi_{\boldsymbol{m}}$ that approximates the target function $f$ well on supp $\phi_{\boldsymbol{m}}$. We then combine $\{\phi_{\boldsymbol{m}}\}_{\boldsymbol{m}\in\{1,2\}^d}$ and $\{\psi_{\boldsymbol{m}}\}_{\boldsymbol{m}\in\{1,2\}^d}$ to obtain a deep neural network that can approximate the target function $f$ well across the entire space. This approach resolves the issue of simultaneous approximation of both functions and their derivatives in Sobolev spaces.

### 7.3 Proofs of Corollaries 1 and 2

#### 7.3.1 Preliminaries

First, we list a few basic lemmas of $\sigma_2$ neural networks repeatedly applied in our main analysis.

**Lemma 10** ([23, Lemma 3.7]). *The following basic lemmas of $\sigma_2$ neural networks hold:*

*(i) $\sigma_1$ neural networks are $\sigma_2$ neural networks.*

*(ii) Any identity map in $\mathbb{R}^d$ can be realized exactly by a $\sigma_2$ neural network with one hidden layer and $2d$ neurons.*

*(iii) $f(x) = x^2$ can be realized exactly by a $\sigma_2$ neural network with one hidden layer and two neurons.*

*(iv) $f(x, y) = xy = \frac{(x+y)^2 - (x-y)^2}{4}$ can be realized exactly by a $\sigma_2$ neural network with one hidden layer and four neurons.*

*(v) Assume $\boldsymbol{x}^{\boldsymbol{\alpha}} = x_1^{\alpha_1} x_2^{\alpha_2} \cdots x_d^{\alpha_d}$ for $\boldsymbol{\alpha} \in \mathbb{N}^d$. For any $N, L \in \mathbb{N}^+$ such that $NL + 2^{\lfloor \log_2 N \rfloor} \geq |\boldsymbol{\alpha}|$, there exists a $\sigma_2$ neural network $\phi(\boldsymbol{x})$ with the width $4N + 2d$ and depth $L + \lceil \log_2 N \rceil$ such that*

$$\phi(\boldsymbol{x}) = \boldsymbol{x}^{\boldsymbol{\alpha}}$$

*for any $\boldsymbol{x} \in \mathbb{R}^d$.*

*(vi) Assume $P(\boldsymbol{x}) = \sum_{j=1}^{J} c_j \boldsymbol{x}^{\boldsymbol{\alpha}_j}$ for $\boldsymbol{\alpha}_j \in \mathbb{N}^d$. For any $N, L, a, b \in \mathbb{N}^+$ such that $ab \geq J$ and $(L - 2b - b\log_2 N)N \geq b\max_j |\boldsymbol{\alpha}_j|$, there exists a $\sigma_2$ neural network $\phi(\boldsymbol{x})$ with the width $4Na + 2d + 2$ and depth $L$ such that*

$$\phi(\boldsymbol{x}) = P(\boldsymbol{x}) \text{ for any } \boldsymbol{x} \in \mathbb{R}^d.$$

Next, we define a function which will be repeatedly used in the proof of Corollary 1 in this section.

**Definition 11.** *Define $s(x)$ from $\mathbb{R} \to [0,1]$ as*

$$s(x) := \begin{cases} 2x^2, & x \in \left[0, \frac{1}{2}\right] \\ -2(x-1)^2 + 1, & x \in \left[\frac{1}{2}, 1\right] \\ 1, & x \in [1,2] \\ -2(x-2)^2 + 1, & x \in \left[2, \frac{5}{2}\right] \\ 2(x-3)^2, & x \in \left[\frac{5}{2}, 3\right] \\ 0, & Otherwise. \end{cases} \tag{92}$$

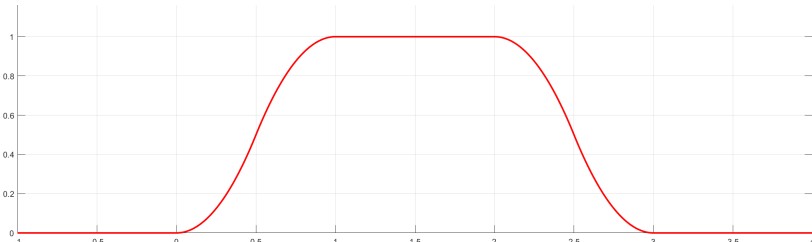

Figure 4: $s(x)$ in $\mathbb{R}$.

**Definition 12.** *Given $K \in \mathbb{N}_+$, then we define two functions in $\mathbb{R}$:*

$$s_1(x) = \sum_{i=0}^{K} s\left(4Kx + 1 - 4i\right), \; s_2(x) = s_1\left(x + \frac{1}{2K}\right). \tag{93}$$

*Then for any $\boldsymbol{m} = (m_1, m_2, \ldots, m_d) \in \{1,2\}^d$, we define*

$$s_{\boldsymbol{m}}(\boldsymbol{x}) := \prod_{j=1}^{d} s_{m_j}(x_j) \tag{94}$$

*for any $\boldsymbol{x} = (x_1, x_2, \ldots, x_d) \in \mathbb{R}^d$.*

**Proposition 8.** *Given $N, L, d \in \mathbb{N}_+$ with $NL + 2^{\lfloor \log_2 N \rfloor} \geq d$ and $L \geq \lceil \log_2 N \rceil$, and setting $K = \lfloor N^{1/d} \rfloor \lfloor L^{2/d} \rfloor$, $\{s_{\boldsymbol{m}}(\boldsymbol{x})\}_{\boldsymbol{m} \in \{1,2\}^d}$ defined in Definition 12 satisfies:*

*(i): $\|s_{\boldsymbol{m}}(\boldsymbol{x})\|_{L^\infty((0,1)^d)} \leq 1$, $\|s_{\boldsymbol{m}}(\boldsymbol{x})\|_{W^{1,\infty}((0,1)^d)} \leq 8K$ and $\|s_{\boldsymbol{m}}(\boldsymbol{x})\|_{W^{1,\infty}((0,1)^d)} \leq 64K^2$ for any $\boldsymbol{m} \in \{1,2\}^d$.*

*(ii): $\{s_{\boldsymbol{m}}(\boldsymbol{x})\}_{\boldsymbol{m} \in \{1,2\}^d}$ is a partition of the unity $[0,1]^d$ with $\operatorname{supp} s_{\boldsymbol{m}}(\boldsymbol{x}) \cap [0,1]^d = \Omega_{\boldsymbol{m}}$ defined in Definition 5.*

*(iii): For any $\boldsymbol{m} \in \{1,2\}^d$, there is a $\sigma_2$ neural network $\lambda_{\boldsymbol{m}}(\boldsymbol{x})$ with the width $16N + 2d$ and depth $4L + 5$ such as*

$$\lambda_{\boldsymbol{m}}(\boldsymbol{x}) = \prod_{j=1}^{d} s_{m_j}(x_j) = s_{\boldsymbol{m}}(\boldsymbol{x}), \boldsymbol{x} \in [0,1]^d.$$

*Proof.* (i) and (ii) are proved by direct calculation. The proof of (iii) follows:

First, we architect $s(x)$ by a $\sigma_2$ neural network. The is a $\sigma_1$ neural network $g(x)$ with 3 the width and one layer such that:

$$g(x) := \begin{cases} x, & x \in \left[0, \frac{1}{2}\right] \\ \frac{1}{2}, & x \in \left[\frac{1}{2}, +\infty\right) \\ 0, & Otherwise. \end{cases} \tag{95}$$

Based on (iii) in Lemma 10, $g^2(x)$ is a $\sigma_2$ neural network with 3 the width and two layers. Then by direct calculation, we notice that

$$s(x) = 2g^2(x) - 2g^2(-x+1) + 2g^2(3-x) - 2g^2(2+x) + \frac{1}{2}, \tag{96}$$

which is a $\sigma_2$ neural network with 12 the width and two layers. The $\widetilde{g}(x)$ defined as

$$\widetilde{g}(x) = \sum_{i=0}^{\lfloor N^{1/d}\rfloor - 1} s\left(4Kx - 4i - \frac{1}{2}\right) \tag{97}$$

is a $\sigma_2$ neural network with $12(\lfloor N^{1/d}\rfloor)$ the width and two layers.

Similar with Lemma 1, we know that

$$\hat{g} = \widetilde{g} \circ \psi_2 \circ \psi_3 \circ \psi_4(x)$$

is a $\sigma_2$ neural network with $12(\lfloor N^{1/d}\rfloor)$ the width and $5 + 2\lfloor L^{1/d}\rfloor$, and

$$s_1(x) = \hat{g}\left(x + \frac{1}{8K}\right), \ s_2(x) = s_1\left(x + \frac{1}{2K}\right), \ x \in [0,1]. \tag{98}$$

Based on (v) in Lemma 10, we have there is a $\sigma_2$ neural network $\lambda_{\boldsymbol{m}}(\boldsymbol{x})$ with the width $16N + 2d$ and depth $4L + 5$ such as

$$\lambda_{\boldsymbol{m}}(\boldsymbol{x}) = \prod_{j=1}^{d} s_{m_j}(x_j) = s_{\boldsymbol{m}}(\boldsymbol{x}), \boldsymbol{x} \in [0,1]^d.$$

$\square$

### 7.3.2  Proof of Corollaries 1 and 2

The proof is comprised of three parts, which include Theorem 8 and 9, followed by the combination of these results. Theorem 8 is to apply the Bramble–Hilbert Lemma 9 measured in the norm of $W^{2,\infty}$:

**Theorem 8.** *Let $K \in \mathbb{N}_+$ and $n \geq 2$. Then for any $f \in W^{n,\infty}((0,1)^d)$ with $\|f\|_{W^{n,\infty}((0,1)^d)} \leq 1$ and $\boldsymbol{m} \in \{1,2\}^d$, there exist piece-wise polynomials function $f_{K,\boldsymbol{m}} = \sum_{|\boldsymbol{\alpha}|\leq n-1} g_{f,\boldsymbol{\alpha},\boldsymbol{m}}(\boldsymbol{x})\boldsymbol{x}^{\boldsymbol{\alpha}}$ on $\Omega_{\boldsymbol{m}}$ (Definition 5) with the following properties:*

$$\|f - f_{K,\boldsymbol{m}}\|_{W^{2,\infty}(\Omega_{\boldsymbol{m}})} \leq C_1(n,d)K^{-(n-2)},$$
$$\|f - f_{K,\boldsymbol{m}}\|_{W^{1,\infty}(\Omega_{\boldsymbol{m}})} \leq C_1(n,d)K^{-(n-1)},$$
$$\|f - f_{K,\boldsymbol{m}}\|_{L^{\infty}(\Omega_{\boldsymbol{m}})} \leq C_1(n,d)K^{-n}. \tag{99}$$

*Furthermore, $g_{f,\boldsymbol{\alpha},\boldsymbol{m}}(\boldsymbol{x}) : \Omega_{\boldsymbol{m}} \to \mathbb{R}$ is a constant function with on each $\Omega_{\boldsymbol{m},\boldsymbol{i}}$ for $\boldsymbol{i} \in \{0,1\ldots,K\}^d$. And*

$$|g_{f,\boldsymbol{\alpha},\boldsymbol{m}}(\boldsymbol{x})| \leq C_2(n,d) \tag{100}$$

*for all $\boldsymbol{x} \in \Omega_{\boldsymbol{m}}$, where $C_1$ and $C_2$ are constants independent with $K$.*

The proof is the same as that of Theorem 6. Note that $\{f_{K,\boldsymbol{m}}\}_{\boldsymbol{m}\in\{1,2\}^d}$ will be same in two theorems if $f \in W^{n,\infty}((0,1)^d)$ in two theorem are same.

Theorem 9 is to establish $\sigma_2$ neural networks $\{\gamma_{\boldsymbol{m}}\}_{\{1,2\}^d}$, and each $\gamma_{\boldsymbol{m}}$ can approximate $f$ well on $\Omega_{\boldsymbol{m}}$.

**Theorem 9.** *For any $f \in W^{n,\infty}((0,1)^d)$ with $\|f\|_{W^{n,\infty}((0,1)^d)} \leq 1$, any $N, L \in \mathbb{N}_+$ with $NL + 2^{\lfloor \log_2 N\rfloor} \geq n$ and $L \geq \lceil \log_2 N\rceil$, and $\boldsymbol{m} = (m_1, m_2, \ldots, m_d) \in \{1,2\}^d$, there is a $\sigma_2$ neural network $\gamma_{\boldsymbol{m}}$ with the width $28n^{d+1}(N+d)\log_2(8N)$ and depth $11n^2(L+2)\log_2(4L)$ such that*

$$\|f(\boldsymbol{x}) - \gamma_{\boldsymbol{m}}(\boldsymbol{x})\|_{W^{2,\infty}(\Omega_{\boldsymbol{m}})} \leq C_{10}(n,d)N^{-2(n-2)/d}L^{-2(n-2)/d}$$
$$\|f(\boldsymbol{x}) - \gamma_{\boldsymbol{m}}(\boldsymbol{x})\|_{W^{1,\infty}(\Omega_{\boldsymbol{m}})} \leq C_{10}(n,d)N^{-2(n-1)/d}L^{-2(n-1)/d}$$
$$\|f(\boldsymbol{x}) - \gamma_{\boldsymbol{m}}(\boldsymbol{x})\|_{L^{\infty}(\Omega_{\boldsymbol{m}})} \leq C_{10}(n,d)N^{-2n/d}L^{-2n/d}, \tag{101}$$

*where $C_{10}$ is the constant independent with $N, L$.*

*Proof.* The proof is similar to that of Theorem 7; the difference is that $xy$ and $x^\alpha$ can be architected precisely by $\sigma_2$ neural networks.

Without loss of the generalization, we consider the case for $\boldsymbol{m}_* = (1, 1, \dots, 1)$. Due to Theorem 8 and setting $K = \lfloor N^{1/d} \rfloor^2 \lfloor L^{2/d} \rfloor$, we have

$$
\begin{aligned}
\|f - f_{K,\boldsymbol{m}_*}\|_{W^{2,\infty}(\Omega_{\boldsymbol{m}_*})} &\le C_1(n,d)K^{-(n-2)} \le C_1(n,d)N^{-2(n-2)/d}L^{-2(n-2)/d} \\
\|f - f_{K,\boldsymbol{m}_*}\|_{W^{1,\infty}(\Omega_{\boldsymbol{m}_*})} &\le C_1(n,d)K^{-(n-1)} \le C_1(n,d)N^{-2(n-1)/d}L^{-2(n-1)/d} \\
\|f - f_{K,\boldsymbol{m}_*}\|_{L^\infty(\Omega_{\boldsymbol{m}_*})} &\le C_1(n,d)K^{-n} \le C_1(n,d)N^{-2n/d}L^{-2n/d},
\end{aligned}
\tag{102}
$$

where $f_{K,\boldsymbol{m}_*} = \sum_{|\boldsymbol{\alpha}| \le n-1} g_{f,\boldsymbol{\alpha},\boldsymbol{m}_*}(\boldsymbol{x})\boldsymbol{x}^{\boldsymbol{\alpha}}$ for $x \in \Omega_{\boldsymbol{m}_*}$. Note that $g_{f,\boldsymbol{\alpha},\boldsymbol{m}_*}(\boldsymbol{x})$ is a constant function for $\boldsymbol{x} \in \prod_{j=1}^d \left[\frac{i_j}{K}, \frac{3+4i_j}{4K}\right]$ and $\boldsymbol{i} = (i_1, i_2, \dots, i_d) \in \{0, 1, \dots, K-1\}^d$. The remaining part is to approximate $f_{K,\boldsymbol{m}_*}$ by neural networks.

The way to approximate $g_{f,\boldsymbol{\alpha},\boldsymbol{m}_*}(\boldsymbol{x})$ is same with Theorem 7, and we have that

$$
\begin{aligned}
\|\phi_{\boldsymbol{\alpha}}(\boldsymbol{\phi}_2(\boldsymbol{x})) - g_{f,\boldsymbol{\alpha},\boldsymbol{m}_*}(\boldsymbol{x})\|_{W^{2,\infty}(\Omega_{\boldsymbol{m}_*})} &= \|\phi_{\boldsymbol{\alpha}}(\boldsymbol{\phi}_2(\boldsymbol{x})) - g_{f,\boldsymbol{\alpha},\boldsymbol{m}_*}(\boldsymbol{x})\|_{W^{1,\infty}(\Omega_{\boldsymbol{m}_*})} \\
&= \|\phi_{\boldsymbol{\alpha}}(\boldsymbol{\phi}_2(\boldsymbol{x})) - g_{f,\boldsymbol{\alpha},\boldsymbol{m}_*}(\boldsymbol{x})\|_{L^\infty(\Omega_{\boldsymbol{m}_*})} \\
&\le 2C_2(n,d)N^{-2n}L^{-2n}
\end{aligned}
\tag{103}
$$

which is due to $\phi_{\boldsymbol{\alpha}}(\boldsymbol{\phi}_2(\boldsymbol{x})) - g_{f,\boldsymbol{\alpha},\boldsymbol{m}_*}(\boldsymbol{x})$ is a step function, and the first order weak derivative is 0 in $\Omega_{\boldsymbol{m}_*}$.

Due to (v) in Lemma 10, there is a $\sigma_2$ neural network $\phi_{5,\boldsymbol{\alpha}}(\boldsymbol{x})$ with the width $4N + 2d$ and depth $L + \lceil \log_2 N \rceil$ such that

$$
\phi_{5,\boldsymbol{\alpha}}(\boldsymbol{x}) = \boldsymbol{x}^{\boldsymbol{\alpha}}, \ \boldsymbol{x} \in \mathbb{R}^d.
\tag{104}
$$

Due to (iv) in Lemma 10, there is a $\sigma_2$ neural network $\phi_6(\boldsymbol{x})$ with the width 4 and depth 1 such that

$$
\phi_6(x, y) = xy, \ x, y \in \mathbb{R}.
\tag{105}
$$

Now we define the neural network $\gamma_{\boldsymbol{m}_*}(\boldsymbol{x})$ to approximate $f_{K,\boldsymbol{m}_*}(\boldsymbol{x})$ in $\Omega_{\boldsymbol{m}_*}$:

$$
\gamma_{\boldsymbol{m}_*}(\boldsymbol{x}) = \sum_{|\boldsymbol{\alpha}| \le n-1} \phi_6\left[\phi_{\boldsymbol{\alpha}}(\boldsymbol{\phi}_2(\boldsymbol{x})), \phi_{5,\boldsymbol{\alpha}}(\boldsymbol{x})\right].
\tag{106}
$$

The remaining question is to find the error $\mathcal{E}$:

$$
\begin{aligned}
\widetilde{\mathcal{E}} &:= \left\| \sum_{|\boldsymbol{\alpha}| \le n-1} \phi_6\left[\phi_{\boldsymbol{\alpha}}(\boldsymbol{\phi}_2(\boldsymbol{x})), \phi_{5,\boldsymbol{\alpha}}(\boldsymbol{x})\right] - f_{K,\boldsymbol{m}_*}(\boldsymbol{x}) \right\|_{W^{2,\infty}(\Omega_{\boldsymbol{m}_*})} \\
&\le \sum_{|\boldsymbol{\alpha}| \le n-1} \left\| \phi_6\left[\phi_{\boldsymbol{\alpha}}(\boldsymbol{\phi}_2(\boldsymbol{x})), \phi_{5,\boldsymbol{\alpha}}(\boldsymbol{x})\right] - g_{f,\boldsymbol{\alpha},\boldsymbol{m}_*}(\boldsymbol{x})\boldsymbol{x}^{\boldsymbol{\alpha}} \right\|_{W^{2,\infty}(\Omega_{\boldsymbol{m}_*})} \\
&= \sum_{|\boldsymbol{\alpha}| \le n-1} \left\| \phi_{\boldsymbol{\alpha}}(\boldsymbol{\phi}_2(\boldsymbol{x}))\boldsymbol{x}^{\boldsymbol{\alpha}} - g_{f,\boldsymbol{\alpha},\boldsymbol{m}_*}(\boldsymbol{x})\boldsymbol{x}^{\boldsymbol{\alpha}} \right\|_{W^{2,\infty}(\Omega_{\boldsymbol{m}_*})} \\
&\le n^2 \sum_{|\boldsymbol{\alpha}| \le n-1} \left\| \phi_{\boldsymbol{\alpha}}(\boldsymbol{\phi}_2(\boldsymbol{x})) - g_{f,\boldsymbol{\alpha},\boldsymbol{m}_*}(\boldsymbol{x}) \right\|_{W^{2,\infty}(\Omega_{\boldsymbol{m}_*})} \\
&\le 2n^{d+2}C_2(n,d)N^{-2n}L^{-2n}.
\end{aligned}
\tag{107}
$$

At last, we finish the proof by estimating the network's the width and depth, implementing $\gamma_{\boldsymbol{m}_*}(\boldsymbol{x})$. From Eq. (106), we know that $\gamma_{\boldsymbol{m}_*}(\boldsymbol{x})$ consists of the following subnetworks:

1. $\phi_{5,\boldsymbol{\alpha}}(\boldsymbol{x})$ with the width $4N + 2d$ and depth $L + \lceil \log_2 N \rceil$.

2. $\boldsymbol{\phi}_2(\boldsymbol{x})$ with the width $4N + 5$ and depth $4L + 4$.

3. $\phi_{\boldsymbol{\alpha}}$ with the width $16n(N+1)\log_2(8N)$ and depth $(5L+2)\log_2(4L)$.

4. $\phi_6(x, y)$ with the width 4 and depth 1.

Therefore $\phi(\boldsymbol{x})$ is a neural network with the width $28n^{d+1}(N+d)\log_2(8N)$ and depth $11n^2(L+2)\log_2(4L)$.

Combining Eqs. (102) and (107), we have that there is a neural network $\gamma_{\boldsymbol{m}_*}$ with the width $28n^{d+1}(N+d)\log_2(8N)$ and depth $11n^2(L+2)\log_2(4L)$ such that

$$\|f(\boldsymbol{x}) - \psi_{\boldsymbol{m}_*}(\boldsymbol{x})\|_{W^{2,\infty}(\Omega_{\boldsymbol{m}_*})} \leq C_{10}(n,d)N^{-2(n-2)/d}L^{-2(n-2)/d}$$

$$\|f(\boldsymbol{x}) - \psi_{\boldsymbol{m}_*}(\boldsymbol{x})\|_{W^{1,\infty}(\Omega_{\boldsymbol{m}_*})} \leq C_{10}(n,d)N^{-2(n-1)/d}L^{-2(n-1)/d}$$

$$\|f(\boldsymbol{x}) - \psi_{\boldsymbol{m}_*}(\boldsymbol{x})\|_{L^\infty(\Omega_{\boldsymbol{m}_*})} \leq C_{10}(n,d)N^{-2n/d}L^{-2n/d}, \tag{108}$$

where $C_{10} = C_1 + 2n^{d+2}C_2$ is the constant independent with $N, L$.

Similarly, we can construct a neural network $\gamma_{\boldsymbol{m}}$ with the width $28n^{d+1}(N+d)\log_2(8N)$ and depth $11n^2(L+2)\log_2(4L)$ which can approximate $f$ on $\Omega_{\boldsymbol{m}}$ with same order of Eq. (108). $\qquad\square$

The last part is to combine $\{\lambda_{\boldsymbol{m}}\}_{\boldsymbol{m}\in\{1,2\}^d}$ and $\{\gamma_{\boldsymbol{m}}\}_{\boldsymbol{m}\in\{1,2\}^d}$ in $[0,1]^d$ and obtain a $\sigma_2$ neural network to approximate $f$ measured in the norm of $W^2$.

*Proof of Corollary 1.* Based on Theorem 9, there is a sequence of the neural network $\{\gamma_{\boldsymbol{m}}(\boldsymbol{x})\}_{\boldsymbol{m}\in\{1,2\}^d}$ such that

$$\|f(\boldsymbol{x}) - \gamma_{\boldsymbol{m}}(\boldsymbol{x})\|_{W^{2,\infty}(\Omega_{\boldsymbol{m}})} \leq C_{10}(n,d)N^{-2(n-2)/d}L^{-2(n-2)/d}$$

$$\|f(\boldsymbol{x}) - \gamma_{\boldsymbol{m}}(\boldsymbol{x})\|_{W^{1,\infty}(\Omega_{\boldsymbol{m}})} \leq C_{10}(n,d)N^{-2(n-1)/d}L^{-2(n-1)/d}$$

$$\|f(\boldsymbol{x}) - \gamma_{\boldsymbol{m}}(\boldsymbol{x})\|_{L^\infty(\Omega_{\boldsymbol{m}})} \leq C_{10}(n,d)N^{-2n/d}L^{-2n/d}, \tag{109}$$

where $C_{10}$ is the constant independent with $N, L$, and each $\gamma_{\boldsymbol{m}}$ is a neural network with the width $28n^{d+1}(N+d)\log_2(8N)$ and depth $11n^2(L+2)\log_2(4L)$. According to Proposition 8, there is a sequence of the neural network $\{s_{\boldsymbol{m}}(\boldsymbol{x})\}_{\boldsymbol{m}\in\{1,2\}^d}$ satisfies:

(i): $\|s_{\boldsymbol{m}}(\boldsymbol{x})\|_{L^\infty((0,1)^d)} \leq 1$, $\|s_{\boldsymbol{m}}(\boldsymbol{x})\|_{W^{1,\infty}((0,1)^d)} \leq 8K$ and $\|s_{\boldsymbol{m}}(\boldsymbol{x})\|_{W^{1,\infty}((0,1)^d)} \leq 64K^2$ for any $\boldsymbol{m} \in \{1,2\}^d$.

(ii): $\{s_{\boldsymbol{m}}(\boldsymbol{x})\}_{\boldsymbol{m}\in\{1,2\}^d}$ is a partition of the unity $[0,1]^d$ with supp $s_{\boldsymbol{m}}(\boldsymbol{x}) \cap [0,1]^d = \Omega_{\boldsymbol{m}}$ defined in Definition 5.

For each $s_{\boldsymbol{m}}$, it is a $\sigma_2$ neural network with the width $16N + 2d$ and depth $4L + 5$.

Due to (iv) in Lemma 10, there is a $\sigma_2$ neural network $\phi_6(\boldsymbol{x})$ with the width $4$ and depth $1$ such that

$$\phi_6(x,y) = xy, \ x,y \in \mathbb{R}. \tag{110}$$

Now we define

$$\gamma(\boldsymbol{x}) = \sum_{\boldsymbol{m}\in\{1,2\}^d} \phi_6(s_{\boldsymbol{m}}(\boldsymbol{x}), \gamma_{\boldsymbol{m}}(\boldsymbol{x})). \tag{111}$$

Note that

$$\widetilde{\mathcal{R}} := \|f(\boldsymbol{x}) - \gamma(\boldsymbol{x})\|_{W^{2,\infty}((0,1)^d)} \leq \sum_{\boldsymbol{m}\in\{1,2\}^d} \|s_{\boldsymbol{m}}(x) \cdot f(\boldsymbol{x}) - s_{\boldsymbol{m}}(\boldsymbol{x})\gamma_{\boldsymbol{m}}(\boldsymbol{x})\|_{W^{2,\infty}((0,1)^d)}$$

$$= \sum_{\boldsymbol{m}\in\{1,2\}^d} \|s_{\boldsymbol{m}}(x) \cdot f(\boldsymbol{x}) - s_{\boldsymbol{m}}(\boldsymbol{x})\gamma_{\boldsymbol{m}}(\boldsymbol{x})\|_{W^{2,\infty}(\Omega_{\boldsymbol{m}})}. \tag{112}$$

where the last equality is due to supp $s_{\boldsymbol{m}}(\boldsymbol{x}) \cap [0,1]^d = \Omega_{\boldsymbol{m}}$.

Then due to chain rule, for each $\boldsymbol{m} \in \{1,2\}^d$, we have

$$\|s_{\boldsymbol{m}}(x) \cdot f(\boldsymbol{x}) - s_{\boldsymbol{m}}(\boldsymbol{x})\gamma_{\boldsymbol{m}}(\boldsymbol{x})\|_{W^{2,\infty}(\Omega_{\boldsymbol{m}})}$$

$$\leq \|s_{\boldsymbol{m}}(x)\|_{W^{2,\infty}(\Omega_{\boldsymbol{m}})} \|f(\boldsymbol{x}) - \gamma_{\boldsymbol{m}}(\boldsymbol{x})\|_{L^\infty(\Omega_{\boldsymbol{m}})} + 2\|s_{\boldsymbol{m}}(x)\|_{W^{1,\infty}(\Omega_{\boldsymbol{m}})} \|f(\boldsymbol{x}) - \gamma_{\boldsymbol{m}}(\boldsymbol{x})\|_{W^{1,\infty}(\Omega_{\boldsymbol{m}})}$$

$$+ \|s_{\boldsymbol{m}}(x)\|_{L^\infty(\Omega_{\boldsymbol{m}})} \|f(\boldsymbol{x}) - \gamma_{\boldsymbol{m}}(\boldsymbol{x})\|_{W^{2,\infty}(\Omega_{\boldsymbol{m}})} + \|s_{\boldsymbol{m}}(x)\|_{W^{1,\infty}(\Omega_{\boldsymbol{m}})} \|f(\boldsymbol{x}) - \gamma_{\boldsymbol{m}}(\boldsymbol{x})\|_{L^\infty(\Omega_{\boldsymbol{m}})}$$

$$+ \|s_{\boldsymbol{m}}(x)\|_{L^\infty(\Omega_{\boldsymbol{m}})} \|f(\boldsymbol{x}) - \gamma_{\boldsymbol{m}}(\boldsymbol{x})\|_{W^{1,\infty}(\Omega_{\boldsymbol{m}})} + \|s_{\boldsymbol{m}}(x)\|_{L^\infty(\Omega_{\boldsymbol{m}})} \|f(\boldsymbol{x}) - \gamma_{\boldsymbol{m}}(\boldsymbol{x})\|_{L^\infty(\Omega_{\boldsymbol{m}})}$$

$$\leq 91 C_{10}(n,d)N^{-2(n-2)/d}L^{-2(n-2)/d}. \tag{113}$$

Hence

$$\widetilde{\mathcal{R}} \le 2^{d+7} C_{10}(n,d) N^{-2(n-2)/d} L^{-2(n-2)/d}.$$

At last, we finish the proof by estimating the network's width and depth, implementing $\gamma(\boldsymbol{x})$. From Eq. (111), we know that $\gamma(\boldsymbol{x})$ consists of the following subnetworks:

1. $\gamma_{\boldsymbol{m}}(\boldsymbol{x})$ with the width $28n^{d+1}(N+d) \log_2(8N)$ and depth $11n^2(L+2) \log_2(4L)$.

2. $s_{\boldsymbol{m}}(\boldsymbol{x})$ with the width $16N + 2d$ and depth $4L + 5$.

3. $\phi_6(x,y)$ with the width $4$ and depth $1$.

Therefore $\gamma(\boldsymbol{x})$ is a neural network with the width $2^{d+6} n^{d+1}(N+d) \log_2(8N)$ and depth $15n^2(L+2) \log_2(4L)$.

$\square$

Our method can easily extend to approximations measured by the norm of $W^{m,\infty}$. The primary difference in the proof lies in the need to establish a differential $\{s_{\boldsymbol{m}}(\boldsymbol{x})\}_{\{1,2\}^d}$, which can be achieved by constructing architected $s_{\boldsymbol{m}}(\boldsymbol{x})$ as piece-wise $m$-degree polynomial functions. By extanding this approach, we can obtain Corollary 2 using our method.

### 7.4 Proof of Theorem 4

*Proof.* The Theorem 4 will be proved by contradiction. The idea of the proof is inspired by Ref. [28].

**Claim 1.** *There exist $\rho, C_1, C_2, C_3, J_0 > 0$ and $s, d \in \mathbb{N}^+$ such that, for any $f \in \mathcal{F}_{n,d}$, we have*

$$\inf_{\phi \in \widehat{\Phi}} |\phi - f|_{W^{1,\infty}((0,1)^d)} \le C_3 L^{-2(n-1)/d - \rho} N^{-2(n-1)/d - \rho}. \tag{114}$$

*for all $NL \ge J_0$, where*

$$\widehat{\Phi} := \{\phi : \text{ReLU FNNs } \phi \text{ with the width} \le C_1 N \log N \text{ and depth} \le C_2 L \log L\}.$$

The remaining question is to show Claim 1 is invalid.

Denote

$$D\widehat{\Phi} := \{\psi : \psi = D_i \phi, \phi \in \widehat{\Phi}, i = 1, \ldots, d\},$$

Due to Theorem 1, we obtain

$$\text{VCDim}(D\widehat{\Phi}) \le C_4 N^2 L^2 \log_2 L \log_2 N =: b_u. \tag{115}$$

Now we will use Claim 1 to estimate a lower bound

$$b_l := \lfloor (NL)^{\frac{2}{d} + \frac{\rho}{2(n-1)}} \rfloor^d$$

of $\text{VCDim}(D\widehat{\Phi})$. In other words, we will construct $\{\psi_\beta(\boldsymbol{x}) : \psi_\beta(\boldsymbol{x}) \in D\widehat{\Phi}, \beta \in \mathcal{B}\}$ to scatter $b_l$ points. $\mathcal{B}$ will be defined later.

First, fix $i = 1, \ldots, d$, and there exists $\widetilde{g} \in C^\infty (0,1)^d$ such that $\frac{\partial \widetilde{g}(\boldsymbol{0})}{\partial x_i} = 1$ and $\widetilde{g}(\boldsymbol{x}) = 0$ for $\|\boldsymbol{x}\|_2 \ge 1/3$. And we can find a constant $C_5 > 0$ such that $g := \widetilde{g}/C_5 \in \mathcal{F}_{n,d}$.

Denote $M = \lfloor (NL)^{\frac{2}{d} + \frac{\rho}{2(n-1)}} \rfloor$. Divide $[0,1]^d$ into $M^d$ non-overlapping sub-cubes $\{Q_{\boldsymbol{\theta}}\}_{\theta}$ as follows:

$$Q_{\boldsymbol{\theta}} := \left\{ \boldsymbol{x} = [x_1, x_2, \cdots, x_d]^T \in [0,1]^d : x_i \in \left[ \frac{\theta_i - 1}{M}, \frac{\theta_i}{M} \right], i = 1, 2, \cdots, d \right\},$$

for any index vector $\boldsymbol{\theta} = [\theta_1, \theta_2, \cdots, \theta_d]^T \in \{1, 2, \cdots, M\}^d$. Denote the center of $Q_{\boldsymbol{\theta}}$ by $\boldsymbol{x_\theta}$ for all $\boldsymbol{\theta} \in \{1, 2, \cdots, M\}^d$. Define

$$\mathcal{B} := \{\beta : \beta \text{ is a map from } \{1, 2, \cdots, M\}^d \text{ to } \{-1, 1\}\}.$$

For each $\beta \in \mathcal{B}$, we define, for any $\boldsymbol{x} \in \mathbb{R}^d$,

$$h_\beta(\boldsymbol{x}) := \sum_{\boldsymbol{\theta} \in \{1,2,\cdots,M\}^d} M^{-n}\beta(\boldsymbol{\theta})g_{\boldsymbol{\theta}}(\boldsymbol{x}), \quad \text{where } g_{\boldsymbol{\theta}}(\boldsymbol{x}) = g\left(M \cdot (\boldsymbol{x} - \boldsymbol{x_\theta})\right).$$

Due to $|\text{supp}\widetilde{g}(\boldsymbol{x})| \leq \frac{2}{3}$ and $|D^{\boldsymbol{\alpha}}h_\beta(\boldsymbol{x})| \leq M^{-n+|\boldsymbol{\alpha}|}\|g\|_{W^{n,\infty}} \leq 1$, we obtain that

$$|D^{\boldsymbol{\alpha}}f_\beta(\boldsymbol{x})| \leq 1$$

for any $|\boldsymbol{\alpha}| \leq n$ Therefore, $f_\beta \in \mathcal{F}_{n,d}$. And it is easy to check $\{D_i h_\beta = h_\beta : \beta \in \mathcal{B}\}$ can scatters $b_l$ points since $\frac{\partial \widetilde{g}(\boldsymbol{0})}{\partial x_i} = 1$ and $\widetilde{g}(\boldsymbol{x}) = 0$ for $\|\boldsymbol{x}\|_2 \geq 1/3$.

Note that for any $h_\beta \in \mathcal{F}_{n,d}$, there is a $\phi_\beta \in \widehat{\Phi}$ such that $C_3(NL)^{\frac{-2(n-1)}{d}-\frac{\rho}{2}} \geq |D_i h_\beta(\boldsymbol{x_\theta}) - D_i\phi_\beta(\boldsymbol{x_\theta})|$ for any $J_\beta \leq NL$ due to Claim 1. Denote $J_1 = \max_{\beta \in \mathcal{B}}\{J_\beta\}$. There is a constant $J_2$ such that $\frac{M^{-n+1}}{C_5} \geq C_3(NL)^{\frac{-2(n-1)}{d}-\rho}$ for $J_2 \leq NL$. Define $J := \max\{J_1, J_2\}$, then for any $J \leq NL$, we have

$$|D_i h_\beta(\boldsymbol{x_\theta})| = \left|M^{-n+1}\frac{\partial g(\boldsymbol{x_\theta})}{\partial x_i}\right| = \frac{M^{-n+1}}{C_5} \geq C_3(NL)^{\frac{-2(n-1)}{d}-\rho} \geq |D_i h_\beta(\boldsymbol{x_\theta}) - D_i\phi_\beta(\boldsymbol{x_\theta})|.$$
(116)

In other words, for any $\beta \in \mathcal{B}$ and $\boldsymbol{\theta} \in \{1,2,\cdots,M\}^d$, $D_i f_\beta(\boldsymbol{x_\theta})$ and $D_i\phi_\beta(\boldsymbol{x_\theta})$ have the same sign. Then $\{D_i\phi_\beta : \beta \in \mathcal{B}\}$ shatters $\{\boldsymbol{x_\theta} : \boldsymbol{\theta} \in \{1,2,\cdots,M\}^d\}$ since $\{D_i h_\beta : \beta \in \mathcal{B}\}$ shatters $\{\boldsymbol{x_\theta} : \boldsymbol{\theta} \in \{1,2,\cdots,M\}^d\}$ as discussed above. Hence,

$$\text{VCDim}\left(\{\phi_\beta : \beta \in \mathcal{B}\}\right) \geq M^d = b_l,$$
(117)

for $N, L \in \mathbb{N}$ with $NL \geq J$.

By Eqs. (115,117), for any $N, L \in \mathbb{N}$ with $NL \geq J$, we have $b_l \leq \text{VCDim}\left(\{\phi_\beta : \beta \in \mathcal{B}\}\right) \leq \text{VCDim}(D\widehat{\Phi}) \leq b_u$, implying that

$$\lfloor (NL)^{\frac{2}{d}+\frac{\rho}{2(n-1)}}\rfloor^d \leq C_4 N^2 L^2 \log_2 L \log_2 N$$
(118)

which is a contradiction for sufficiently large $N, L \in \mathbb{N}$. So we finish the proof of Theorem 4. $\quad\square$

Based on the proof of Theorem 4, we can easily check that the estimation of VC-dimension of DNN derivatives (Theorem 1) is nearly optimal and prove Corollary 3. Assume $\text{VCDim}(D\widehat{\Phi}) \leq b_u = \boldsymbol{O}(N^{2-\varepsilon}L^{2-\varepsilon})$ in Eq. (118) for $\varepsilon > 0$, and $b_l$ must be larger than $\lfloor (NL)^{\frac{2}{d}}\rfloor^d$ according the construction in the proof of Theorem 4 and Theorem 3. Hence we still obtain a contradiction in Eq. (118), and the estimation in Theorem 1 is nearly optimal.

## 7.5 Proof of Theorem 5

### 7.5.1 Bounding generalization error by Rademacher complexity

**Definition 13** (Rademacher complexity [3]). *Given a sample set $S = \{z_1, z_2, \ldots, z_M\}$ on a domain $\mathcal{Z}$, and a class $\mathcal{F}$ of real-valued functions defined on $\mathcal{Z}$, the empirical Rademacher complexity of $\mathcal{F}$ in $S$ is defined as*

$$\mathbf{R}_S(\mathcal{F}) := \frac{1}{M}\mathbf{E}_{\Sigma_M}\left[\sup_{f \in \mathcal{F}}\sum_{i=1}^M \sigma_i f(z_i)\right],$$

*where $\Sigma_M := \{\sigma_1, \sigma_2, \ldots, \sigma_M\}$ are independent random variables drawn from the Rademacher distribution, i.e., $\mathbf{P}(\sigma_i = +1) = \mathbf{P}(\sigma_i = -1) = \frac{1}{2}$ for $i = 1, 2, \ldots, M$. For simplicity, if $S = \{z_1, z_2, \ldots, z_M\}$ is an independent random variable set with the uniform distribution, denote*

$$\mathbf{R}_M(\mathcal{F}) := \mathbf{E}_S\mathbf{R}_S(\mathcal{F}).$$

The following lemma will be used to bounded generalization error by Rademacher complexities:

**Lemma 11** ([47], Proposition 4.11). *Let $\mathcal{F}$ be a set of functions. Then*

$$\mathbf{E}_X \sup_{u \in \mathcal{F}} \left| \frac{1}{M} \sum_{i=1}^{M} u(x_j) - \mathbf{E}_{x \sim \mathcal{P}_\Omega} u(x) \right| \leq 2\mathbf{R}_M(\mathcal{F}),$$

*where $X := \{x_1, \ldots, x_M\}$ is an independent random variable set with the uniform distribution.*

Now we can show that generalization error can be bounded by Rademacher complexities of two function sets.

**Lemma 12.** *Let $d, N, L, M \in \mathbb{N}_+$, $B, C_1, C_2 \in \mathbb{R}_+$. For any $f \in W^{1,\infty}((0,1)^d)$ with $\|f\|_{W^{1,\infty}((0,1)^d)} \leq 1$, set*

$$\widetilde{\Phi} := \{\phi : \phi \text{ with the width} \leq C_1 N \log N \text{ and depth} \leq C_2 L \log L, \|\phi\|_{W^{1,\infty}((0,1)^d)} \leq B\}$$

$$D\widetilde{\Phi} := \{\psi : \psi = D_i \phi, i = 1, \ldots, d\}. \tag{119}$$

*We have*

$$2 \sup_{\boldsymbol{\theta}, \phi(\boldsymbol{x};\boldsymbol{\theta}) \in \widetilde{\Phi}} |\mathbf{E}(\mathcal{R}_S(\boldsymbol{\theta})) - \mathcal{R}_D(\boldsymbol{\theta})| \leq 4(B+1)(d\mathbf{R}_M(D\widetilde{\Phi}) + \mathbf{R}_M(\widetilde{\Phi})),$$

*where $\mathbf{E}$ is expected responding to $X$, and $X := \{\boldsymbol{x}_1, \ldots, \boldsymbol{x}_M\}$ is an independent random variables set uniformly distributed on $(0,1)^d$.*

*Proof.* For any $\phi(\boldsymbol{x}; \boldsymbol{\theta}) \in \widetilde{\Phi}$, we have

$$|\mathbf{E}(\mathcal{R}_S(\boldsymbol{\theta})) - \mathcal{R}_D(\boldsymbol{\theta})|$$

$$= \sum_{j=1}^{d} \left( \mathbf{E}\frac{1}{M} \sum_{i=1}^{M} \left| \frac{\partial(f(\boldsymbol{x}_i) - \phi(\boldsymbol{x}_i; \boldsymbol{\theta}))}{\partial x_j} \right|^2 - \int_{(0,1)^d} \left| \frac{\partial(f(\boldsymbol{x}) - \phi(\boldsymbol{x}; \boldsymbol{\theta}))}{\partial x_j} \right|^2 \mathrm{d}\boldsymbol{x} \right)$$

$$+ \mathbf{E}\frac{1}{M} \sum_{i=1}^{M} |(f(\boldsymbol{x}_i) - \phi(\boldsymbol{x}_i; \boldsymbol{\theta}))|^2 - \int_{(0,1)^d} |(f(\boldsymbol{x}) - \phi(\boldsymbol{x}; \boldsymbol{\theta}))|^2 \mathrm{d}\boldsymbol{x}$$

$$\leq (B+1) \sum_{j=1}^{d} \left( \mathbf{E} \left| \frac{1}{M} \sum_{i=1}^{M} \frac{\partial(f(\boldsymbol{x}_i) - \phi(\boldsymbol{x}_i; \boldsymbol{\theta}))}{\partial x_j} - \int_{(0,1)^d} \frac{\partial(f(\boldsymbol{x}) - \phi(\boldsymbol{x}; \boldsymbol{\theta}))}{\partial x_j} \mathrm{d}\boldsymbol{x} \right| \right)$$

$$+ (B+1)\mathbf{E} \left| \frac{1}{M} \sum_{i=1}^{M} (f(\boldsymbol{x}_i) - \phi(\boldsymbol{x}_i; \boldsymbol{\theta})) - \int_{(0,1)^d} (f(\boldsymbol{x}) - \phi(\boldsymbol{x}; \boldsymbol{\theta})) \mathrm{d}\boldsymbol{x} \right|$$

$$\leq 2(B+1)(d\mathbf{R}_M(D\widetilde{\Phi}) + \mathbf{R}_M(\widetilde{\Phi})) \tag{120}$$

where the last inequality is due to Lemma 12. $\qquad\square$

### 7.5.2 Bounding the Rademacher complexity and the proof of Theorem 5

In this subsection, we aim to estimate the Rademacher complexity using the covering number. We then estimate the covering number using the pseudo-dimension.

**Definition 14** (covering number [3]). *Let $(V, \|\cdot\|)$ be a normed space, and $\Theta \in V$. $\{V_1, V_2, \ldots, V_n\}$ is an $\varepsilon$-covering of $\Theta$ if $\Theta \subset \cup_{i=1}^{n} B_{\varepsilon, \|\cdot\|}(V_i)$. The covering number $\mathcal{N}(\varepsilon, \Theta, \|\cdot\|)$ is defined as*

$$\mathcal{N}(\varepsilon, \Theta, \|\cdot\|) := \min\{n : \exists \varepsilon\text{-covering over } \Theta \text{ of size } n\}.$$

**Definition 15** (Uniform covering number [3]). *Suppose the $\mathcal{F}$ is a class of functions from $\mathcal{F}$ to $\mathbb{R}$. Given $n$ samples $\boldsymbol{Z}_n = (z_1, \ldots, z_n) \in \mathcal{X}^n$, define*

$$\mathcal{F}|_{\boldsymbol{Z}_n} = \{(u(z_1), \ldots, u(z_n)) : u \in \mathcal{F}\}.$$

*The uniform covering number $\mathcal{N}(\varepsilon, \mathcal{F}, n)$ is defined as*

$$\mathcal{N}(\varepsilon, \mathcal{F}, n) = \max_{\boldsymbol{Z}_n \in \mathcal{X}^n} \mathcal{N}\left(\varepsilon, \mathcal{F}|_{\boldsymbol{Z}_n}, \|\cdot\|_\infty\right),$$

*where $\mathcal{N}\left(\varepsilon, \mathcal{F}|_{\boldsymbol{Z}_n}, \|\cdot\|_\infty\right)$ denotes the $\varepsilon$-covering number of $\mathcal{F}|_{\boldsymbol{Z}_n}$ w.r.t the $L_\infty$-norm.*

Then we use a lemma to estimate the Rademacher complexity using the covering number.

**Lemma 13** (Dudley's theorem [3]). *Let $\mathcal{F}$ be a function class such that $\sup_{f \in \mathcal{F}} \|f\|_\infty \leq B$. Then the Rademacher complexity $\mathbf{R}_n(\mathcal{F})$ satisfies that*

$$\mathbf{R}_n(\mathcal{F}) \leq \inf_{0 \leq \delta \leq B} \left\{ 4\delta + \frac{12}{\sqrt{n}} \int_\delta^B \sqrt{\log 2\mathcal{N}(\varepsilon, \mathcal{F}, n)} \, d\varepsilon \right\}$$

To bound the Rademacher complexity, we employ Lemma 13, which bounds it by the uniform covering number. We estimate the uniform covering number by the pseudo-dimension based on the following lemma.

**Lemma 14** ([3]). *Let $\mathcal{F}$ be a class of functions from $\mathcal{X}$ to $[-B, B]$. For any $\varepsilon > 0$, we have*

$$\mathcal{N}(\varepsilon, \mathcal{F}, n) \leq \left( \frac{2enB}{\varepsilon Pdim(\mathcal{F})} \right)^{Pdim(\mathcal{F})}$$

*for $n \geq Pdim(\mathcal{F})$.*

The remaining problem is to bound $\mathrm{Pdim}(\widetilde{\Phi})$ and $\mathrm{Pdim}(D\widetilde{\Phi})$. Based on [4], $\mathrm{Pdim}(\widetilde{\Phi}) = \boldsymbol{O}(L^2 N^2 \log_2 L \log_2 N)$. For the $\mathrm{Pdim}(D\widetilde{\Phi})$, we can estimate it by Theorem 2.

Now we can estimate generalization error based on Lemma 12.

*Proof of Theorem 5.* Let $J = \max\{\mathrm{Pdim}(D\widetilde{\Phi}), \mathrm{Pdim}(\widetilde{\Phi})\}$. Due to Lemma 13, 14 and Theorem 2, for any $M \geq J$, we have

$$\mathbf{R}_M(D\widetilde{\Phi}) \leq 4\delta + \frac{12}{\sqrt{M}} \int_\delta^B \sqrt{\log 2\mathcal{N}(\varepsilon, D\widetilde{\Phi}, M)} \, d\varepsilon$$

$$\leq 4\delta + \frac{12}{\sqrt{M}} \int_\delta^B \sqrt{\log 2 \left( \frac{2eMB}{\varepsilon \mathrm{Pdim}(D\widetilde{\Phi})} \right)^{\mathrm{Pdim}(D\widetilde{\Phi})}} \, d\varepsilon$$

$$\leq 4\delta + \frac{12B}{\sqrt{M}} + 12 \left( \frac{\mathrm{Pdim}(D\widetilde{\Phi})}{M} \right)^{\frac{1}{2}} \int_\delta^B \sqrt{\log \left( \frac{2eMB}{\varepsilon \mathrm{Pdim}(D\widetilde{\Phi})} \right)} \, d\varepsilon. \qquad (121)$$

By the direct calculation for the integral, we have

$$\int_\delta^B \sqrt{\log \left( \frac{2eMB}{\varepsilon \mathrm{Pdim}(D\widetilde{\Phi})} \right)} \, d\varepsilon \leq B \sqrt{\log \left( \frac{2eMB}{\delta \mathrm{Pdim}(D\widetilde{\Phi})} \right)}.$$

Then choosing $\delta = B \left( \frac{\mathrm{Pdim}(D\widetilde{\Phi})}{M} \right)^{\frac{1}{2}} \leq B$, we have

$$\mathbf{R}_M(D\widetilde{\Phi}) \leq 28B \left( \frac{\mathrm{Pdim}(D\widetilde{\Phi})}{M} \right)^{\frac{1}{2}} \sqrt{\log \left( \frac{2eM}{\mathrm{Pdim}(D\widetilde{\Phi})} \right)}. \qquad (122)$$

Therefore, due to Theorem 2, there is a constant $C_4$ independent with $L, N, M$ such as

$$\mathbf{R}_M(D\widetilde{\Phi}) \leq C_4 \frac{NL(\log_2 L \log_2 N)^{\frac{1}{2}}}{\sqrt{M}} \log M. \qquad (123)$$

$\mathbf{R}_M(\widetilde{\Phi})$ can be estimate in the similar way. Due to Lemma 12, we have that there is a constant $C_5 = C_5(B, d, C_1, C_2)$ such that

$$\mathbf{E}\mathcal{R}_S(\boldsymbol{\theta}_D) - \mathcal{R}_D(\boldsymbol{\theta}_D) + \mathbf{E}\mathcal{R}_D(\boldsymbol{\theta}_S) - \mathbf{E}\mathcal{R}_S(\boldsymbol{\theta}_S) \leq C_5 \frac{NL(\log_2 L \log_2 N)^{\frac{1}{2}}}{\sqrt{M}} \log M. \qquad (124)$$

$\square$

