}_*,i}), \ \gamma(\Omega_{\boldsymbol{m}_*,i}) = \frac{\mathrm{diam}(\Omega_{\boldsymbol{m}_*,i})}{r_{\max}^\star(\Omega_{\boldsymbol{m}_*,i})} = 2\sqrt{d}.$$

Therefore we can apply the Bramble–Hilbert Lemma 9 and have

$$\|\tilde{f} - p_{f,i}\|_{L^\infty(\Omega_{\boldsymbol{m}_*,i})} \leq C_{BH}(n,d) K^{-n}$$
$$|\tilde{f} - p_{f,i}|_{W^{1,\infty}(\Omega_{\boldsymbol{m}_*,i})} \leq C_{BH}(n,d) K^{-(n-1)} \tag{59}$$

where $C_{BH}(n,d) = |\{|\boldsymbol{\alpha}| = n\}| \frac{1}{d \int_0^1 x^{d-1} e^{-(1-x^2)^{-1}} \, \mathrm{d}x} \left(2 + 4\sqrt{d}\right)^d C_E$ by following the proof of Lemma [7, Lemma 4.3.8]. Therefore,

$$\|\tilde{f} - p_{f,i}\|_{W^{1,\infty}(\Omega_{\boldsymbol{m}_*,i})} \leq C_1(n,d) K^{-(n-1)}$$

where $C_1(n,d) = 2 C_{BH}(n,d)$.

Now we construct a partition of unity that we use in this theorem. First of all, given any integer $K$, define $\{h_i\}_{i=0}^K$ from $\mathbb{R} \to \mathbb{R}$:

$$h_i(x) := h\left(4K\left(x - \frac{8i+3}{8K}\right)\right), \ h(x) := \begin{cases} 1, & |x| < \frac{3}{2} \\ 0, & |x| > 2 \\ 4 - 2|x|, & \frac{3}{2} \leq |x| \leq 2. \end{cases} \tag{60}$$

It is easy to check that $\{h_i\}_{i=0}^K$ is a partition of unity of $[0,1]$ and $h_i(x) = 1$ for $x \in \left[\frac{i}{K}, \frac{3+4i}{4K}\right]$. Hence we can define $h_{\boldsymbol{i}}(\boldsymbol{x})$ for $\boldsymbol{i} = (i_1, i_2, \ldots, i_d) \in \{0, 1, \ldots, K\}^d$ and $\boldsymbol{x} = (x_1, x_2, \ldots, x_d) \in \mathbb{R}^d$:

$$h_{\boldsymbol{i}}(\boldsymbol{x}) = \prod_{j=1}^d h_{i_j}(x_j), \tag{61}$$

and $\{h_{\boldsymbol{i}} : \boldsymbol{i} \in \{0, 1, \ldots, K\}^d\}$ is a partition of unity of $[0,1]^d$ and $h_{\boldsymbol{i}}(\boldsymbol{x}) = 1$ for $\boldsymbol{x} \in \prod_{j=1}^d \left[\frac{i_j}{K}, \frac{3+4i_j}{4K}\right] = \Omega_{\boldsymbol{m}_*,i}$ and $\boldsymbol{i} = (i_1, i_2, \ldots, i_d) \in \{0, 1, \ldots, K\}^d$.

Furthermore,

$$\|h_{\boldsymbol{i}}(\tilde{f} - p_{f,i})\|_{L^\infty(\Omega_{\boldsymbol{m}_*,i})} \leq \|\tilde{f} - p_{f,i}\|_{L^\infty(\Omega_{\boldsymbol{m}_*,i})} \leq C_{BH}(n,d) K^{-n} \tag{62}$$

and

$$|h_{\boldsymbol{i}}(\tilde{f} - p_{f,i})|_{W^{1,\infty}(\Omega_{\boldsymbol{m}_*,i})} \leq |\tilde{f} - p_{f,i}|_{W^{1,\infty}(\Omega_{\boldsymbol{m}_*,i})} \leq C_{BH}(n,d) K^{-(n-1)} \tag{63}$$

which is due to $h_{\boldsymbol{i}} = 1$ on $\Omega_{\boldsymbol{m}_*,i}$.

Then

$$\|h_{\boldsymbol{i}}(\tilde{f} - p_{f,i})\|_{W^{1,\infty}(\Omega_{\boldsymbol{m}_*,i})} \leq C_1(n,d) K^{-(n-1)}.$$

Finally,

$$\left\| f - \sum_{i \in \{0,1,\dots,K\}^d} h_i p_{f,i} \right\|_{W^{1,\infty}(\Omega_{m_*})} \le \max_{i \in \{0,1,\dots,K\}^d} \| h_i(\tilde{f} - p_{f,i}) \|_{W^{1,\infty}(\Omega_{m_*,i})}$$
$$\le C_1(n,d) K^{-(n-1)}, \tag{64}$$

which is due to $\cup_{i \in \{0,1,\dots,K\}^d} \Omega_{m_*,i} = \Omega_{m_*}$ and $\operatorname{supp} h_i \cap \Omega_{m_*} = \Omega_{m_*,i}$.

Similarly,

$$\left\| f - \sum_{i \in \{0,1,\dots,K\}^d} h_i p_{f,i} \right\|_{L^\infty(\Omega_{1,d})} \le C_1(n,d) K^{-n}. \tag{65}$$

Last of all,

$$f_{k,m_*}(x) := \sum_{i \in \{0,1,\dots,K\}^d} h_i p_{f,i} = \sum_{i \in \{0,1,\dots,K\}^d} \sum_{|\alpha| \le n-1} h_i c_{f,i,\alpha} x^\alpha$$
$$= \sum_{|\alpha| \le n-1} \sum_{i \in \{0,1,\dots,K\}^d} h_i c_{f,i,\alpha} x^\alpha$$
$$=: \sum_{|\alpha| \le n-1} g_{f,\alpha,m_*}(x) x^\alpha \tag{66}$$

with $|g_{f,\alpha,m_*}(x)| \le C_2(n,d)$ for $x \in \Omega_{m_*}$. Note that $g_{f,\alpha,m_*}(x)$ is a step function from $\Omega_{m_*} \to \mathbb{R}$:

$$g_{f,\alpha,m_*}(x) = c_{f,i,\alpha} \tag{67}$$

for $x \in \prod_{j=1}^d \left[ \frac{i_j}{K}, \frac{3+4i_j}{4K} \right]$ and $i = (i_1, i_2, \dots, i_d)$ since $h_i(x) = 0$ for $x \in \Omega_{m_*} \setminus \prod_{j=1}^d \left[ \frac{i_j}{K}, \frac{3+4i_j}{4K} \right]$ and $h_i(x) = 1$ for $x \in \prod_{j=1}^d \left[ \frac{i_j}{K}, \frac{3+4i_j}{4K} \right]$. $\square$

### 7.2.3 Approximation of functions in $W^{n,\infty}$ with $W^{1,\infty}$ norm by ReLU neural networks in the whole space except a small set

**Theorem 7.** *For any $f \in W^{n,\infty}((0,1)^d)$ with $\|f\|_{W^{n,\infty}((0,1)^d)} \le 1$, any $N, L \in \mathbb{N}_+$, and $m = (m_1, m_2, \dots, m_d) \in \{1,2\}^d$, there is a neural network $\psi_m$ with the width $25 n^{d+1}(N+1) \log_2(8N)$ and depth $27 n^2(L+2) \log_2(4L)$ such that*

$$\|f(x) - \psi_m(x)\|_{W^{1,\infty}(\Omega_m)} \le C_6(n,d) N^{-2(n-1)/d} L^{-2(n-1)/d}$$
$$\|f(x) - \psi_m(x)\|_{L^\infty(\Omega_m)} \le C_6(n,d) N^{-2n/d} L^{-2n/d}, \tag{68}$$

*where $C_6$ is the constant independent with $N, L$.*

*Proof.* Without loss of the generalization, we consider the case for $m_* = (1,1,\dots,1)$. Due to Theorem 6 and setting $K = \lfloor N^{1/d} \rfloor^2 \lfloor L^{2/d} \rfloor$, we have

$$\|f - f_{K,m_*}\|_{W^{1,\infty}(\Omega_{m_*})} \le C_1(n,d) K^{-(n-1)} \le C_1(n,d) N^{-2(n-1)/d} L^{-2(n-1)/d}$$
$$\|f - f_{K,m_*}\|_{L^\infty(\Omega_{m_*})} \le C_1(n,d) K^{-n} \le C_1(n,d) N^{-2n/d} L^{-2n/d}, \tag{69}$$

where $f_{K,m_*} = \sum_{|\alpha| \le n-1} g_{f,\alpha,m_*}(x) x^\alpha$ for $x \in \Omega_{m_*}$. Note that $g_{f,\alpha,m_*}(x)$ is a constant function for $

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

$\square$