# OpenReview forum: "Nearly Optimal VC-Dimension and Pseudo-Dimension Bounds for Deep Neural Network Derivatives"
_NeurIPS.cc/2023/Conference — NeurIPS 2023 poster_

### Official Review · Reviewer_vGRi · 2023-07-03

**Soundness:** 3 good
**Presentation:** 4 excellent
**Contribution:** 3 good
**Rating:** 7
**Confidence:** 3

**Summary:**

The paper proposed a method to estimate the Vapnik-Chervonenkis (VC) dimension and pseudo-dimension of deep neural network (DNN) derivatives with the ReLU activation function, which have important applications such as characterizing the generalization error of machine learning methods and establishing the optimal approximation of DNNs in Sobolev spaces.

The authors provided theoretical analysis and proofs for their proposed method, which fills a gap in learning error estimations for many physics-informed machine learning models and applications, including solving partial differential equations, operator learning, network compression, and regularization.

Another contribution of the paper is the demonstration of how DNNs can be used to approximate functions in Sobolev spaces using ReLU activation functions in a deep feedforward neural network architecture, with a nearly-optimal approximation rate.

Overall, the study provides a framework for analyzing and optimizing DNNs for different applications while taking into account mathematical concepts such as VC-dimension and pseudo-dimension.

**Strengths:**

* The topic of the paper is highly relevant to the field of deep learning and offers an interesting approach to estimate VC-dimension and pseudo-dimension of derivatives of deep neural networks.
* The paper is well-written and clearly presents the mathematical language and definitions used in the study. The proofs provided are detailed and structured in a logical manner.
* The paper presents two important theorems that provide a solution to the approximation rate problem of DNNs in Sobolev spaces and the degree of generalization error in loss functions involving derivatives of DNNs.
* The proposed approach has the potential to be applied in different areas of physics-informed machine learning such as solving partial differential equations, operator learning, and generative models.

**Weaknesses:**

- Some aspects of the paper could be clearer and more thoroughly explained. The introduction, for instance, could better demonstrate the main contribution of the paper and provide a more detailed overview of the state-of-the-art and the limitations of existing research.

- The section on references could be more comprehensive, covering more related studies and presenting a more thorough overview of the existing literature.

**Questions:**

1. Pseudo dimension is a more general concept than VC dimension,  while the bounds in Theorem 1 and Theorem 2 seems similar to each other except two constants $\hat{C}$ and $\overline{C}$, is there any relationship between  $\hat{C}$ and $\overline{C}$ ?

2. Are those results in this paper holds only for ReLU networks?

---

> ### Author Rebuttal · Authors · 2023-08-10
>
> We are grateful to the reviewer for your thorough and diligent review, helpful feedback, positive remarks, and insightful summary.
>
> Reviewer's comment: " Some aspects of the paper could be clearer and more thoroughly explained. The introduction, for instance, could better demonstrate the main contribution of the paper and provide a more detailed overview of the state-of-the-art and the limitations of existing research."
>
> Response: Thanks for providing kindly suggestions. In the introduction (Page 2), we add a more comprehensive discussion of the state-of-the-art and the limitations of existing research on the VC-dimension and pseudo-dimension of DNNs. We observe that most existing research in this area does not consider the derivatives of DNNs, which are crucial in the error analysis of Sobolev training. Furthermore, we note that a recent study by Duan et al. [2021] analyzed the VC-dimension and pseudo-dimension of DNN derivatives, but their results were suboptimal due to a lack of consideration for the relationships between the multiplied terms in a DNN derivative. As a result, their findings cannot be used to determine the optimal approximation error of DNNs in Sobolev training, and may only provide a generalization error that is much larger than the actual error that may arise from Sobolev training.
>
> Reviewer's comment: " The section on references could be more comprehensive, covering more related studies and presenting a more thorough overview of the existing literature. "
>
>
>
> Response: We appreciate your feedback and have made additional revisions to the introduction and references. Specifically, we have added more references on Sobolev training and the estimation of VC-dimension and pseudo-dimension to the introduction and references sections.
>
>
>
> Reviewer's comment: " Pseudo-dimension is a more general concept than VC dimension, while the bounds in Theorem 1 and Theorem 2 seems similar to each other except two constants $\bar{C}$ and $\hat{C}$, is there any relationship between them?"
>
> Response: Yes. As you can see in the proof of Theorem 2, we establish that $$\bar{C}(N+1)^2(L+1)^2\log_2 (L+1)\log_2 (N+1)\le 64\bar{C}N^2L^2\log_2 L\log_2 N.$$ Since the pseudo-dimension of derivatives of DNNs with $N$ width and $L$ depth can be controlled by the VC-dimension of derivatives of DNNs with $N+1$ width and $L+1$ depth. Therefore, we conclude that $64\bar{C}\ge\hat{C}$. We add this discussion after the proof of Theorem 2 in appendix.
>
> Reviewer's comment: " Are those results in this paper holds only for ReLU networks?"
>
> Response: We do not only consider ReLU networks. In Corollaries 1 and 2, we present the approximation results of DNNs with ReLU and square of ReLU activation functions, respectively. Unlike piece-wise polynomial activation functions such as ReLU and square of ReLU, developing the method based on this paper for DNNs with other activation functions can be challenging. Especially, difficulties arise from not only building DNNs that can approximate functions in Sobolev spaces but also estimating the VC-dimension of these DNNs. This is an interesting question that requires further exploration, and we refer to it as an area for future research.

---

> > ### Comment · Reviewer_vGRi · 2023-08-15
> >
> > Thank you for addressing some of my comments. I am maintaining my score.

---

> > > ### Author Response · Authors · 2023-08-15
> > >
> > > Thank you again for your help and suggestions.

---

### Official Review · Reviewer_MY91 · 2023-07-04

**Soundness:** 3 good
**Presentation:** 3 good
**Contribution:** 3 good
**Rating:** 6
**Confidence:** 2

**Summary:**

This paper facilitates the understandings of Sobolev training and performances of DNNs in Sobolev spaces by providing the near optimal VC-dimension and pseudo-dimension of DNN derivatives.

**Strengths:**

Technically, they improve the bounds on VC and pseudo-dimensions of DNN derivatives in reference [10].

**Weaknesses:**

Though I think the theoretical contributions of this paper is great, but in terms of readability of the paper, there are some rooms for the improvements. \\
First, this paper assumes that the readers are very familiar with the notions of Sobolev training. In my opinion, authors should defer some technical proofs in the appendix, and introduce the notion briefly in the main paper, and motivate readers why the Sobolev training is interesting problem to consider. \\
Second, some notations should be introduced first, before being stated. I realized the Sobolev Spaces $W^{n,\infty}([0,1]^{d})$ first appeared in line 59, then introduced in line 114, formally.\\
Third, some sentences are repeated quite often. For instance, line numbers 125-127 are same with line numbers 167-169.


**Questions:**

1.	To my knowledge, VC-dimension and Pseudo-dimension are essentially same notion. (Reference [4].) I am wondering why the results in Theorem 1 and Theorem 2 are surprising in a sense that they have the same bound. Is there any intuitive reason on why it is non-trivial to expect they should be same for the DNN derivatives?  \\
2.	What is the meaning of approximating functions in $W^{n, \infty}([0,1]^{d})$ with Sobolev norm $W^{1,\infty}([0,1]^{d})$? Why this is interesting?  \\
3.	How do we know the bound is optimal? To my knowledge, we commonly refer that we have an optimal bound when we have the matching orders of lower bound and upper bounds. But the author only provides the upper bounds in the paper.


**Limitations:**

This work has no negative societal impact.

---

> ### Author Rebuttal · Authors · 2023-08-10
>
> We would like to express our gratitude to the reviewer for your thorough review and positive feedback. We would like to clarify that our contribution is not limited to improving the bounds on VC and pseudo-dimensions of DNN derivatives. Beyond that, our contributions also include: 1) establishes DNNs as effective approximators of functions in Sobolev spaces through the use of Sobolev norms, resulting in lower error rates compared to previous works. 2) demonstrate the optimality of our approach through the estimation of VC-dimension. 3) utilize pseudo-dimension to obtain the generalization error of Sobolev training in supervised learning of DNNs. All are novel results not found in other works.
>
> Reviewer's comment:" Though I think the theoretical contributions of this paper is great, but in terms of readability of the paper, there are some rooms for the improvements. First, this paper assumes that the readers are very familiar with the notions of Sobolev training. In my opinion, authors should defer some technical proofs in the appendix, and introduce the notion briefly in the main paper, and motivate readers why the Sobolev training is interesting problem to consider. Second, some notations should be introduced first, before being stated. I realized the Sobolev Spaces $W^{n,\infty}([0,1]^d)$
>  first appeared in line 59, then introduced in line 114, formally.Third, some sentences are repeated quite often. For instance, line numbers 125-127 are same with line numbers 167-169. "
>
> Response: Thank you for your valuable suggestions! Based on your input, we have moved the complete proofs of Theorems 1 and 2 in the appendix, while providing proof sketches in Section 5. Furthermore, we have enriched the introductions by including additional references and discussing the significance of Sobolev training. We have also added a befily review of a specific task focused on solving partial differential equations to provide a practical context for the application of Sobolev training.
>
> Secondly, we have included the definitions of Sobolev spaces in the introduction section to enhance readability. Finally, we have also removed the duplicated parts. Thanks for your help and suggestions again!
>
> Reviewer's comment: "To my knowledge, VC-dimension and Pseudo-dimension are essentially same notion. (Reference [4].) I am wondering why the results in Theorem 1 and Theorem 2 are surprising in a sense that they have the same bound. Is there any intuitive reason on why it is non-trivial to expect they should be same for the DNN derivatives?"
>
> Response:  Please note that the claim in the arXiv version of Bartlett et al. [2019], stating that the VC-dimension and pseudo-dimension of DNNs are the same, dose not always hold. The authors correct this claim in the published form in Journal of Machine Learning Research (See the reference of their paper in our reference list). Roughly speaking,  the pseudo-dimension can be bounded by the VC-dimension of a larger set. The most challenging aspect of our work revolved around proving the VC-dimension of DNN derivatives. The introduction of the pseudo-dimension was necessary because the VC-dimension alone is insufficient for obtaining a generalization error estimate. We rely on the notion of pseudo-dimension to achieve this.
>
> Secondly, we were not surprised to find that the order of the VC-dimension and pseudo-dimension are the same. What surprised us is the order of VC-dimension between DNNs and their derivatives are same, considering the differences in the complexity of their respective structures.
>
> Reviewer's comment:" What is the meaning of approximating functions in $W^{n,\infty}$ with Sobolev norm $W^{1,\infty}$? Why this is interesting? "
>
> Response: When utilizing ReLU-based DNNs to approximate functions in the Sobolev space $W^{n,\infty}$, the goal is to capture both the magnitude and derivative of the functions.  This is the essence of approximating functions in $W^{n,\infty}$ with the Sobolev norm $W^{1,\infty}$. Although it is desired to consider approximating functions in $W^{n,\infty}$ with Sobolev norms $W^{m,\infty}$ for $m\geq2$, one has to recognize that ReLU DNNs lack higher-order derivatives, making them impossible to approximate functions in Sobolev norms $W^{m,\infty}$ with $m\geq2$. As a result, when employing ReLU DNNs, the primary focus is on approximations measured within the $W^{1,\infty}$ framework e.g. Gühring et al. [2020].
>
> The consideration of approximating functions in $W^{n,\infty}$ with the Sobolev norm $W^{1,\infty}$ sufficiently explains the success of DNNs in Sobolev training, particularly when dealing with loss functions that only involve first-order derivatives of both the DNNs and the target functions, as mentioned in the introduction, such as solving second-order partial differential equations (PDEs) in a weak sense and penalizing function gradients in the loss functions to control the Lipschitz constant of DNNs. Therefore, this scenario is both valuable and interesting.
>
> Reviewer's comment:" How do we know the bound is optimal? To my knowledge, we commonly refer that we have an optimal bound when we have the matching orders of lower bound and upper bounds. But the author only provides the upper bounds in the paper. "
>
> Response: The paper's Corollaries 3 and 4 provide lower bounds for the VC-dimension and Pseudo-dimension, respectively. These lower bounds are proven to be on the order of $O(N^{2-\epsilon}L^{2-\epsilon})$ for any small $\epsilon>0$. The upper bounds in Theorems 1 and 2 are $O(N^2L^2\log_2 N\log _2L$). This means that the paper's results are considered nearly optimal because the order of polynomials in the upper bounds cannot be further reduced without contradicting Corollaries 3 and 4. Note that there still exists a small gap between the upper bound and the lower bound, which is the meaning of nearly optimal,

---

> > ### Comment · Reviewer_MY91 · 2023-08-13
> > **Thank you for your rebuttal.**
> >
> > I have no further questions. I will raise the score to 6.

---

> > > ### Author Response · Authors · 2023-08-15
> > >
> > > Thank you for your appreciation of our work.

---

### Official Review · Reviewer_2REw · 2023-07-05

**Soundness:** 3 good
**Presentation:** 4 excellent
**Contribution:** 4 excellent
**Rating:** 7
**Confidence:** 3

**Summary:**

The authors provide estimates on two measures of statistical complexity, the VC-dimension and the pseudo-dimension, of derivatives of deep neural networks. The estimate of the VC-dimension is shown to be optimal up to logarithmic factors. They also propose a constructive method for approximating functions in Sobolev spaces by deep neural networks. The VC-dimension bound is used to show that the obtained approximation rate is optimal as a function of the width and depth of the network. Finally, they prove a generalization bound in terms of Sobolev norm by leveraging the pseudo-dimension upper bound.

**Strengths:**

The paper is globally well-written and pleasant to read. It brings several new results regarding the statistical and approximation properties of deep neural networks in terms of Sobolev norms, which could be of broad use, in particular in the community of deep learning for PDEs. I like that most of the presented bounds have matching lower bounds. I have not checked the proofs in details, so I cannot provide evidence on their soundness, but the mathematical statements presented in the main paper are easy to understand and unambiguous.

**Weaknesses:**

I do not have any strong reservation, a few questions are list below. The only part of the paper that I found hard to follow is the proof of Theorem 1. I suggest that authors take advantage of the additional page to expand a bit on the proof. Perhaps a drawing would help?

**Questions:**

+ Line 77: you claim that the estimate of the pseudo-dimension is nearly optimal, but I do not see a lower bound in the paper. Could you provide a lower-bound or at least an argument on how to obtain one, or otherwise change the phrasing of this sentence (and similar ones elsewhere in the paper)?
+ Line 101: I don’t understand the \leq sign. Shouldn’t it be an equal sign? Otherwise, I feel the argument of lines 184-187 breaks down, since \sigma_2 networks include ReLU networks.
+ Line 129: the dependence of the width on the dimension d is exponential. Is this expected? Do you think that you could get a matching dependence in the lower boud?
+ Line 224 and Theorem 5: I think it would be clearer to upper bound your generalization error term by 2 sup_{\theta} |\Esp(R_S(\theta)) - R_D(\theta)|. Otherwise it is a bit confusing since, without further clarification, the expectation applies both to the estimator \theta_S and to the random function R_S. Similarly, in the proof of Lemma 12, I don’t think that the proof is correct as it is if you apply it for \theta_S, since \theta_S depends on the random sample. However, it is correct if you write it for any (deterministic) \theta, thereby getting the sup over \theta as in the LHS of Lemma 11, which you can then apply to get the same upper bound that you get with your proof.


**Limitations:**

See weaknesses.

---

> ### Author Rebuttal · Authors · 2023-08-10
>
> We extend our heartfelt appreciation to the reviewer for your comprehensive and diligent review, invaluable feedback, positive remarks, and insightful summary.
>
> Reviewer's comment: "I do not have any strong reservation, a few questions are list below. The only part of the paper that I found hard to follow is the proof of Theorem 1. I suggest that authors take advantage of the additional page to expand a bit on the proof. Perhaps a drawing would help?"
>
> Response: We greatly appreciate your kind suggestions and insights. We agree that the proof of Theorem 1 in our paper is lengthy. We have found that providing a sketch of the proof can effectively convey the main ideas to a wider range of readers. The most challenging and extensive aspect lies in the refinements necessary to obtain the partitions of parameter spaces, which are crucial for our analysis. For this purpose, we have included a concise sketch of the proof in the main paper. This will allow readers to grasp the key concepts and logical flow without being overwhelmed by excessive details. We have moved the detailed proof of refinements required to obtain the partitions to the appendix, for those who desire a more thorough understanding. By adopting this approach, we hope to enhance the overall readability of our paper.
>
> Reviewer's comment: "Line 77: you claim that the estimate of the pseudo-dimension is nearly optimal, but I do not see a lower bound in the paper. Could you provide a lower-bound or at least an argument on how to obtain one, or otherwise change the phrasing of this sentence (and similar ones elsewhere in the paper)? "
>
> Response: We would like to express our sincere appreciation for your attentive review of our paper. We have included Corollary 4 for the lower bound of the pseudo-dimension, which demonstrates the near optimality of the pseudo-dimension estimate presented in Theorem 2.
>
> Reviewer's comment: "Line 101: I don’t understand the $\leq$ sign. Shouldn’t it be an equal sign? Otherwise, I feel the argument of lines 184-187 breaks down, since $\sigma_2$ networks include ReLU networks."
>
>  Response:  The sign $\leq$ is correct. We understand the concern regarding the presence of higher-order derivatives in DNNs that employ ReLU activation functions. While it is true that not all DNNs utilizing ReLU or the square of ReLU activation functions have higher-order derivatives, it is worth noting that some DNNs do have higher-order derivatives. For instance, the expression $\sigma_2\circ\sigma_2\circ (\sigma_1(x)-\sigma_1(-x))$ has higher-order derivatives.
>
> In the proofs of Corollaries 1 and 2, we construct $\sigma_2$ networks that include both ReLU and square of ReLU networks. These constructed networks are designed to approximate functions measured by $W^{m,\infty}$ norms, which implies that they can effectively capture higher-order derivatives.
>
> Reviewer's comment: " Line 129: the dependence of the width on the dimension $d$ is exponential. Is this expected? Do you think that you could get a matching dependence in the lower bound? "
>
> Response: In this paper, we focus on the optimality of approximation rate with respect to width $N$ and depth $L$ of DNNs. The dimensionality $d$ is not the focus that we consider in our research. Addressing your question about mitigating the exponential dependence of width on dimensionality, we have observed that this arises from the utilization of methods like Taylor's expansion or average Taylor polynomials in our approximation techniques. It remains an open question for future research to explore alternative approaches to address the challenge of getting the dependence of $d$ in the lower bounds.
>
> Reviewer's comment: "Line 224 and Theorem 5: I think it would be clearer to upper bound your generalization error term by $2 \sup_{\theta} |E(R_S(\theta)) - R_D(\theta)|$. Otherwise it is a bit confusing since, without further clarification, the expectation applies both to the estimator $\theta_S$ and to the random function $R_S$. Similarly, in the proof of Lemma 12, I don’t think that the proof is correct as it is if you apply it for $\theta_S$, since $\theta_S$ depends on the random sample. However, it is correct if you write it for any (deterministic) $\theta$, thereby getting the sup over $\theta$ as in the LHS of Lemma 11, which you can then apply to get the same upper bound that you get with your proof. "
>
> Response: Thank you for your feedback. Following the suggestion, we have revised Theorem 5 and Lemma 12 to ensure clarity and avoid confusion.

---

> > ### Comment · Reviewer_2REw · 2023-08-11
> >
> > I thank the authors for taking the time to write the rebuttal. All my questions are addressed thoroughly. My rating is unchanged.
> >
> > > The sign $\leq$  is correct (...)
> >
> > Thank you for the clarification. Then I understand that the word “alone” in line 184 is crucial? I suggest expanding a bit the explanation in this paragraph to clarify why it is still acceptable to have ReLU activations appearing in the network.
> >
> > > In this paper, we focus on the optimality of approximation rate with respect to width $N$ and depth $L$ of DNNs. The dimensionality $d$ is not the focus (...)
> >
> > Thank you for the clarification. I suggest adding this discussion to the paper.

---

> > > ### Author Response · Authors · 2023-08-12
> > >
> > > Thank you for your valuable suggestions. We genuinely appreciate your input, and we will make the necessary additions to our paper as per your recommendations in the final version. Specifically, we will include an explanation in Line 184 regarding our utilization of the smooth partition of the unit to ensure the elimination of parts of ReLU-based DNNs that lack high-order derivatives in the final presentation. Furthermore, we will incorporate a discussion on the dependency of the width on the dimension $d$ after presenting Theorem 3.

---

### Official Review · Reviewer_dSYn · 2023-07-09

**Soundness:** 3 good
**Presentation:** 3 good
**Contribution:** 3 good
**Rating:** 6
**Confidence:** 4

**Summary:**

The main contribution of the paper are new VC dimension and pseudo-dimension bounds for derivatives of functions implemented by deep neural networks. The utility of these bounds is demonstrated by proving the tightness of approximation error bounds in the Sobolev norm for networks with ReLU and ReLU squared activations, and by giving an improved generalization bound in a similar setting.

**Strengths:**

**Contribution.** This is a good technical paper that improves state of the art in the theoretical studies of VC/pseudo-dimensions and approximation and generalization rates of DNNs. The focus of the paper is the setting where VC/pseudo-dimensions are estimated for model derivatives, and approximation/generalization is considered with respect to Sobolev norms. This setting is not so well-explored as the more common setting where the fitted functions are assumed to belong to a Sobolev space, but the error of fitting does not involve the derivatives. The main results claimed in the paper are new VC/pseudo-dimensions bounds. The other results are new approximation and generalization bounds. The VC/pseudo-dimension bounds naturally help to obtain the generalization bounds and show the tightness of approximation bounds. It appears that all the results established in the paper improve previous analogous results in terms of giving more accurate/tight rates.

**Quality and clarity.** The paper is fairly well written. All the results are precisely stated, sketches of proofs are provided where appropriate (full proofs provided in the appendix), connections between the results are well-explained, previous work is duly mentioned.

**Weaknesses:**

I don't see any major issues in the paper, but my overall impression is that it is fairly technical and lacks significant new insights. Virtually all results in the paper rely very heavily on previous research, and most of them look like being assembled from ideas scattered across many previous publications. I find it hard to name new ideas that never appeared before. I would say that this paper is more suitable for a journal.

I think that the claims of achievement in this paper are exaggerated. The main claim is connected with the new VC-dimension bound: "obtaining such bounds for DNN derivatives is much more difficult", "DNN derivatives consist of a series of interdependent parts...rendering existing methods for estimating bounds inapplicable". In fact, the bound for VC-dimension of derivatives given in Theorem 1 is very close to the bound for the original network function given in Theorem 7 of Bartlett et al (2019), and the proof of Theorem 1 is just a slight modification of the proof of Theorem 7 in Bartlett et al (2019). This is not surprising because the chain rule expression (13) for the derivatives of the network function is only slightly more complicated than the original function for the purpose of partitioning the parameter domain into piecewise polynomial components and estimating the degrees of the resulting polynomials, as required for the proof. The authors claim "we propose a method to achieve nearly optimal estimations of the VC-dimension and pseudo-dimension of DNN derivatives", but I don't see here any new method.

**Questions:**

What are the important non-technical takeaways from this paper?

**Limitations:**

See above

---

> ### Author Rebuttal · Authors · 2023-08-10
>
> We sincerely appreciate the reviewer's careful reading of our paper, as well as your positive feedback and helpful suggestions.
>
> The problem we addressed is important and open since the nearly tight VC dimension bound of neural networks (not their derivatives) by Bartlett. Even though our analysis tools are not fundamentally new, we have addressed an open problem with wide applications in the theoretical analysis of deep neural networks. We hope that the reviewer understands that it may not be necessary to address an open problem with completely novel analysis tools. The main contributions of our research are establishing nearly optimal DNN structures in Sobolev spaces and deriving nearly optimal bounds for VC-dimension and pseudo-dimension. To the best of our knowledge, no existing work has achieved these results, despite the long-standing consideration of derivatives in DNN training, such as solving partial differential equations (PDEs) using DNNs. Our findings confirm the effectiveness of DNNs in Sobolev training and substantially improve approximation and generalization errors compared to methodologies described in the existing literature, such as those proposed by Duan et al. (2021), De Ryck and Mishra (2022), and Jiao et al. (2023). We believe that our results are of great importance and interest to the NeurIPS community. Presentation of our paper at the conference will enable the community to quickly access our findings and to use our results in their further analyses of DNN algorithms, ultimately leading to a broader impact. This aligns with the purpose of a conference paper on NeurIPS.
>
> Moreover, to achieve our theory, there are several technical difficulties to be addressed with new ideas.
>
> Firstly, one particular difficulty arises when constructing ReLU-based DNNs in Theorem 1. In the work of Lu et al. [2021], the target functions are approximated in the trifling region, the entire domain except for a small subset, and techniques such as integration and shifting, employing a middle-value function are utilized to control the error in the small subset. However, when it comes to approximating functions in Sobolev spaces, these methods prove to be ineffective since the derivatives of DNNs tend to deteriorate in such small subset.
>
> Secondly, we agree that while there are similarities between our method and Bartlett et al. [2019] in proving the upper bound of VC-dimension for DNN derivatives, there are notable differences. Applying the chain rule necessitates considering the correlations between different parts of the DNNs, rather than treating them as independent components multiplied together. This correlation-based partitioning of parameter spaces is a specific and challenging aspect of our estimation process, setting it apart from previous approaches. The difficulties associated with this aspect also contribute to the suboptimality observed in the results of Duan et al. [2021].
>
> Thirdly, our approach to proving the optimality of VC-dimension differs from Bartlett et al. [2019] (Theorem 3). The proof of optimality for DNN derivatives is not easily generalizable using their approach. Instead, we establish the optimality of VC-dimension estimation (Corollary 3) based on the DNN approximation results we derived within the Sobolev space (Theorem 3). Specifically, we demonstrate that if the degree of polynomials in the upper bounds of VC-dimension for DNN derivatives in Theorem 1 can be reduced, it becomes impossible to find DNNs that achieve the established approximation rate. This approach distinguishes it from Bartlett et al. [2019] (Theorem 3).
>
> While this paper primarily focuses on technical aspects, it also offers valuable non-technical insights.
>
> 1. DNNs outperform traditional methods: According to Theorem 1 in our paper, DNNs with $O(N^2L\log L(\log N)^2)$ parameters can achieve an error rate of $O(N^{\frac{-2(n-1)}{d}}L^{\frac{-2(n-1)}{d}})$ when measured by the norm in $W^{1,\infty}$, while approximating functions in $W^{n,\infty}$ in Sobolev spaces. In comparison, traditional methods like finite elements require $O(N^2L^2)$ parameters to achieve the same approximation error. This shows that DNNs have a clear advantage over traditional methods in terms of approximation in Sobolev spaces, particularly in terms of the freedom of the depth parameter $L$. Notably, this result is not found in other papers that focus on DNN approximation in Sobolev spaces measured by Sobolev norms, such as Gühring et al. [2020] and Gühring and Raslan [2021], as their results are suboptimal.
>
> 2. Replacing ReLU with squared ReLU: Although ReLU-based DNNs cannot approximate functions measured by $W^{m,\infty}$ for $m \geq 2$ due to their lack of smoothness, our paper suggests a solution by replacing some ReLU activations with squared ReLU activations. This modification allows for the approximation of functions measured by higher-order Sobolev norms. Furthermore, the number of squared ReLU activations required can be very few, as shown in the proof of Corollary 1.2.
>
> 3. Generalization error and sample points: Based on Theorem 5, our findings indicate that learning target functions with loss functions defined by Sobolev norms does not require substantially more sample points compared to those defined by $L_2$-norms. The generalization error orders of these two types of loss functions are equivalent with respect to the width $N$ and depth $L$ of DNNs.
>
> 4. Implement of solving PDEs by DNNs: Our findings serve as confirmation that DNNs are indeed capable of effectively solving partial differential equations (PDEs) within frameworks like Deep Ritz, Wasserstein GAN (WGAN), and Physics-Informed Neural Networks (PINN). Furthermore, our research contributes significant advancements by substantially improving the approximation and generalization errors presented in the methodologies in this field proposed by Duan et al. [2021], etc.

---

> > ### Comment · Reviewer_dSYn · 2023-08-16
> > **Thank you**
> >
> > Thank you for your replies, I find them generally reasonable.
> >
> > I still think, however, that it is not quite fair for you to write "*there are similarities between our method and Bartlett et al. [2019] in proving the upper bound of VC-dimension*". In fact, your proof very closely follows the structure and specific elements of the original proof. Your contribution is, indeed, in extending it to the more complex scenario involving derivatives. It might be reasonable to add a comment to the paper explaining in more detail the relation of your proof to the original proof, and the associated challenges.
> >
> > Anyway, I'm increasing my score.

---

> > > ### Author Response · Authors · 2023-08-16
> > > **Thank you**
> > >
> > > Thank you for your valuable suggestion. We will incorporate a comment in the final version of the paper, providing a more detailed explanation of the relationship between our proof and the original proof, as well as discussing the associated challenges.

---

### Author Rebuttal · Authors · 2023-08-10

We would like to express our gratitude for all the reviewers' valuable suggestions and careful reading. Based on your advice, we have made several improvements to our paper. Additionally, we have added an example for solving Partial Differential Equations by DNNs with Sobolev training in the introduction section, as well as a corollary discussing lower bounds of pseudo-dimension. We have provided further details in the attached file.

---

### Comment · Area_Chair_TxqU · 2023-08-11

Hi all,

Thanks for serving as the reviewers for this submission. As the authors have already turned in their responses. It is our turn to start the further discussion. Here is a to-do list:

(1) Please acknowledge the authors when you finish reading their responses.
(2) Please indicate whether you have any further questions for the authors such that they can continue to response.
(3) Please indicate whether you are willing to change the ratings.

Best

AC

---

### Decision · Program_Chairs · 2023-09-21

**Decision:**

Accept (poster)

**Comment:**

This paper presents novel theoretical results bounding the VC/pseudo-dimensions and approximation/generalization properties of derivatives of deep neural networks. The technical contributions improve upon prior work and fill an important gap in the theoretical deep learning literature.

The reviewers agree that the core theoretical results are solid and represent meaningful advances to the state of the art. While constructive feedback is offered to further strengthen the manuscript, the reviewers concur that the central contributions warrant acceptance.

In particular, the proofs of the main VC/dimension and approximation/generalization rate bounds are viewed as technically strong. These theoretical findings can enable progress in diverse applications including physics-informed deep learning.

I recommend accepting this paper based on its significant theoretical contributions. The authors should address the constructive feedback through revisions that expand the framing, provide more intuition, consolidate repetition, and further situate the work in the literature. These revisions will help broaden the impact of this theoretically grounded work.